# 30-year lidar observations of the stratospheric aerosol layer state over Tomsk (Western Siberia, Russia)

Vladimir V. Zuev[1,2,3], Vladimir D. Burlakov[4], Aleksei V. Nevzorov[4], Vladimir L. Pravdin[1], Ekaterina S. Savelieva[1], and Vladislav V. Gerasimov[1,2]

[1]Institute of Monitoring of Climatic and Ecological Systems SB RAS, Tomsk, 634055, Russia
[2]Tomsk State University, Tomsk, 634050, Russia
[3]Tomsk Polytechnic University, Tomsk, 634050, Russia
[4]V.E. Zuev Institute of Atmospheric Optics SB RAS, Tomsk, 634055, Russia

*Correspondence to*: Vladislav V. Gerasimov (gvvsnake@mail.ru)

**Abstract.** There are only four lidar stations in the world which have almost continuously performed observations of the stratospheric aerosol layer (SAL) state for over the last 30 years. The longest time series of the SAL lidar measurements have been accumulated at the Mauna Loa Observatory (Hawaii) since 1973, the NASA Langley Research Center (Hampton, Virginia) since 1974, and Garmisch-Partenkirchen (Germany) since 1976. The fourth lidar station we present started to perform routine observations of the SAL parameters in Tomsk (56.48° N, 85.05° E, Western Siberia, Russia) in 1986. In this

paper, we mainly focus on and discuss the stratospheric background period from 2000 to 2005 and the causes of the SAL perturbations over Tomsk in the 2006–2015 period. During the last decade, volcanic aerosol plumes from tropical Mt. Manam, Soufriere Hills, Rabaul, Merapi, Nabro, and Kelut, and extratropical (northern) Mt. Okmok, Kasatochi, Redoubt, Sarychev Peak, Eyjafjallajökull, and Grimsvötn were detected in the stratosphere over Tomsk. When it was possible, we used the NOAA HYSPLIT trajectory model to assign aerosol layers observed over Tomsk to the corresponding volcanic

eruptions. The trajectory analysis highlighted some surprising results. For example, in cases of the Okmok, Kasatochi, and Eyjafjallajökull eruptions, the HYSPLIT air-mass backward trajectories, started from altitudes of aerosol layers detected over Tomsk with a lidar, passed over these volcanoes on their eruption days at altitudes higher than the maximum plume altitudes given by the Smithsonian Institution Global Volcanism Program. An explanation of these facts is suggested. The role of both tropical and northern volcanoes eruptions in volcanogenic aerosol loading of the mid-latitude stratosphere is also

discussed. In addition to volcanoes, we considered other possible causes of the SAL perturbations over Tomsk, i.e. the polar stratospheric cloud (PSC) events and smoke plumes from strong forest fires. At least two PSC events were detected in 1995 and 2007. We also make an assumption that the Kelut volcano eruption (Indonesia, February 2014) could be the cause of the SAL perturbations over Tomsk during the first quarter of 2015.

# 1 Introduction

Long-term studies show that the presence of various types of aerosol in the stratosphere is mainly caused by powerful volcanic eruptions (Robock, 2000; Robock and Oppenheimer, 2003). Volcanic eruptions are ranked in the volcanic explosivity index (VEI) category from 0 to 8 (Newhall and Self, 1982; Siebert et al., 2010). During Plinian or, more rarely, Vulcanian explosive eruptions with VEI ≥ 3, volcanic ejecta and gases can directly reach the stratospheric altitudes, where the volcanogenic aerosol stays for a long time. Then this aerosol spreads throughout the global stratosphere in the form of clouds. The volcanogenic aerosol perturbs the radiation-heat balance of the atmosphere, and thus, significantly affects the atmospheric dynamics and climate (Timmreck, 2012; Driscoll et al., 2012; Kremser et al., 2016). The injection of volcanogenic aerosol particles into the stratosphere leads to a considerable increase of their specific surface area and, therefore, to activation of heterogeneous chemical reactions on the surface of these particles. The reactions can result in, e.g., stratospheric ozone depletion (Hofmann and Solomon, 1989; Prather, 1992; Randel et al., 1995). Moreover, the long-term presence of volcanogenic aerosol clouds in the stratosphere also leads to cooling of the underlying surface and near-surface atmosphere due to the aerosol scattering and extinction of the direct solar radiation (Stenchikov et al., 2002). The latter effect is the basis for several geoengineering projects on artificial climate control (Crutzen, 2006; Robock et al., 2009; Kravitz and Robock, 2011; Laakso et al., 2016). These projects require information on aerosol cloud transport in the stratosphere.

Among various techniques for stratospheric aerosol measurements, the lidar remote sensing techniques are the most sensitive and have high spatial and temporal resolution. The number of lidar stations for stratospheric aerosol monitoring significantly increased throughout the world soon after the large volcanic eruption of Mt. Pinatubo (Philippines, 15 June 1991; VEI = 6), the most powerful volcanic eruption of the 20th century after the Novarupta volcano eruption (the Alaska Peninsula, 6 June 1912; VEI = 6; Fierstein and Hildreth, 1992). Some of these lidar stations formed continuous lidar observation networks, such as the Network for the Detection of Stratospheric Change (NDSC; now: NDACC, Network for the Detection of Atmospheric Composition Change; http://www.ndsc.ncep.noaa.gov), the European Aerosol Research Lidar Network (EARLINET; Bösenberg et al., 2003), and the Asian Dust and aerosol lidar observation Network (AD-Net; Murayama et al., 2000). However, before the Pinatubo eruption, only several individual lidars provided continuous monitoring of the stratosphere. The longest time series of the stratospheric aerosol layer (SAL) lidar measurements have been accumulated at the Mauna Loa Observatory (Hawaii) since 1973 (Barnes and Hofmann, 1997; Barnes and Hofmann, 2001), the NASA Langley Research Center (Hampton, Virginia) since 1974 (Woods and Osborn, 2001), and Garmisch-Partenkirchen (Germany) since 1976 (Trickl et al., 2013).

The first lidar observations of the SAL parameters in the USSR were performed at the Institute of Atmospheric Optics (IAO) of the Siberian Branch of the USSR Academy of Sciences (now: V.E. Zuev Institute of Atmospheric Optics of the Siberian Branch of the Russian Academy of Sciences), located in Tomsk, in 1975 (Zuev, 1982). A layer near 19 km altitude with increased stratospheric aerosol concentration due to the sub-Plinian eruption of Fuego volcano (Guatemala, 14 October 1974; VEI = 4) was detected at that time.

Tomsk (56.48° N, 85.05° E, Western Siberia, Russia) is located in the central part of the Eurasian continent. The information on the atmosphere over the vast area of Siberia is poorly presented in various databases. Therefore, the lidar measurements time series accumulated in Tomsk are definitely unique and can be useful, e.g., in studying climate change (Mills et al., 2016). A new lidar station was designed and implemented at the IAO in 1985 for continuous monitoring of the

SAL volcanogenic perturbations and other stratospheric parameters over Tomsk. The monitoring started at the end of 1985 and is ongoing at the present time (i.e. more than 30 years). In 2004, the lidar station in Tomsk was integrated into the Lidar Network for atmospheric monitoring in the Commonwealth of Independent States (CIS-LiNet; Chaykovskii et al., 2005; Zuev et al., 2009). The CIS-LiNet has been established by six lidar teams from Belarus, Russia, and the Kyrgyz Republic. Note that the CIS-LiNet station located in Minsk, Belarus, is also integrated into the European Aerosol Research Lidar

Network (EARLINET; Wandinger et al., 2016).

The detection of high aerosol concentration in the stratosphere over Tomsk after the Nevado del Ruiz volcano eruption (Colombia, 13 November 1985; VEI = 3) marked the beginning of routine lidar observations in 1986 (El'nikov et al., 1988). Definitely, the detection and subsequent monitoring of strong SAL perturbations by volcanogenic aerosol after the Pinatubo eruption were the major events during the first decade of lidar observations in Tomsk. The data of lidar measurements made

in Tomsk over the 1986–2000 period were summarized and analyzed by Zuev et al. (1998) and Zuev et al. (2001).

In this paper, we mainly focus on and discuss: 1) the stratospheric background period from 2000 to 2006; 2) the SAL volcanogenic perturbations during the last decade (2006–2015); and 3) the potential detection of polar stratospheric clouds over Tomsk. The role of strong forest fires in the SAL perturbations is discussed. A brief review of previous lidar observations in Tomsk during the 1986–1999 period is also given.

**2 Lidar instruments and methods**

Regular monitoring of the SAL parameters over Tomsk was started at the IAO with a single-wavelength aerosol lidar in January 1986. A pulsed mode Nd:YAG laser LTI-701 operating at a wavelength of 532 nm with 1 W average power at a pulse repetition rate of 3 kHz was used as the lidar transmitter (El'nikov et al., 1988). The lidar backscattered signals were collected by a Newtonian receiving telescope with a mirror of 1 m diameter and a 2 m focal length. The signals were

registered with a vertical resolution of 374 m by a photomultiplier tube (PMT) FEU-130 (USSR, Moscow Elecro-Lamp Plant) operating in the photon counting mode. The first results of stratospheric ozone measurements were obtained with a modified version of the IAO lidar in 1989 (El'nikov et al., 1989). In 1991, the IAO lidar system was updated with a receiving telescope with a mirror of 2.2 m diameter and a 10 m focal length. Note that this 2.2 m telescope can be used both as Newtonian and prime-focus depending on the remotely sensed object and selected lidar transmitter wavelength. Since 1994,

the IAO lidar system has been named the Siberian Lidar Station (SLS; Zuev, 2000). Now the SLS represents a multichannel station for regular measurements of aerosol parameters, ozone content and vertical distribution, and for temperature

retrievals in the troposphere and stratosphere. The receiving telescopes with the main mirror diameters of 2.2, 1, 0.5, and 0.3 m and lasers operating in the wavelength range 271–1064 nm are used at the SLS for these purposes.

The SLS aerosol channel we consider uses a Nd:YAG laser as the channel transmitter and a Newtonian telescope with a mirror diameter of 0.3 m and a focal length of 1 m as the channel receiver. The laser (LS-2132T-LBO model, LOTIS TII Co., the Republic of Belarus) can operate at wavelengths of 1064, 532, and 355 nm with 200, 100, and 40 mJ pulse energies, respectively, at a pulse repetition rate of 20 Hz. The backscattered signals from altitudes up to the stratopause (~50 km) are registered with a vertical resolution of 100 m by R7206-01 and R7207-01 PMTs (Hamamatsu Photonics, Japan) at used wavelengths of 532 and 355 nm, respectively. The PMTs operate in the photon counting mode. Two shutdown periods of the SLS aerosol channel from July 1997 to May 1999 and from February to September 2014 were due to the maintenance of the channel laser, and the rearrangement and improvement of the SLS. A more detailed technical description of the SLS aerosol channel and its data acquisition electronics can be found, e.g., in (Burlakov et al., 2010).

We use the scattering ratio $R(H)$ to describe the stratospheric aerosol vertical distribution, i.e.

$$R(H) = \frac{\beta_\pi^m(H) + \beta_\pi^a(H)}{\beta_\pi^m(H)} = 1 + \frac{\beta_\pi^a(H)}{\beta_\pi^m(H)}, \tag{1}$$

where $\beta_\pi^m(H)$ and $\beta_\pi^a(H)$ are the molecular (Rayleigh) and aerosol (Mie) backscatter coefficients, respectively; $\pi$ denotes an angle of $\pi$ radian, i.e. the angle of the backscatter lidar signal propagation. The SLS aerosol channel makes it possible to receive almost undisturbed backscattered signals from altitudes of ~40–45 km. At higher altitudes, the signal-to-noise ratio is too low. Therefore, altitudes of ~30–35 km, where the stratosphere is considered to be aerosol-free, were used as the calibration altitudes $H_0$. Thus, the detected lidar signals were calibrated by normalizing them to the molecular backscatter signal from altitudes $H_0 \geq 30$ km. The calibration method of lidar signals against the molecular backscatter coefficient $\beta_\pi^m(H)$ is described in detail by, e.g., Measures (1984).

We use the integrated aerosol backscatter coefficient $B_\pi^a$ to describe the temporal dynamics (time series) of stratospheric aerosol loading over Tomsk. The coefficient is calculated for a certain range of stratospheric altitudes $(H_1; H_2)$

$$B_\pi^a = \int_{H_1}^{H_2} \beta_\pi^a(H)dH. \tag{2}$$

Here $H_1$ is the local tropopause altitude or slightly above, where upper-tropospheric aerosol does not contribute to the value of $B_\pi^a$, and $H_2$ corresponds to the calibration altitude $H_0 = 30$ km. Tomsk is located near the southern boundary of subarctic latitudes, where the tropopause altitude can significantly vary, e.g., due to migration of the Arctic stratospheric jet stream within the Tomsk region. Sometimes one can observe a double (or even multiple) tropopause. For this reason, we consciously removed the interval of the tropopause altitude variations to observe the stratospheric perturbations only. As the tropopause altitude over Tomsk varies from ~11 to 13 km, depending on season, we set $H_1 = 15$ km.

Various data on volcanic eruptions were taken from the Smithsonian Institution Global Volcanism Program (GVP; http://volcano.si.edu/; Section: Reports; Subsections: Smithsonian/USGS Weekly Volcanic Activity Report and Bulletin of the Global Volcanism Network). To study the SAL volcanogenic perturbations, we also analyze air-mass backward trajectories started from aerosol layers observed over Tomsk. All the trajectories were calculated by using the NOAA's Hybrid Single-Particle Lagrangian Integrated Trajectory model (HYSPLIT; Stein et al., 2015; http://ready.arl.noaa.gov/HYSPLIT.php) and the HYSPLIT-compatible NOAA meteorological data from the Global Data Assimilation System (GDAS) one-degree archive.

## 3 Results of the SAL lidar observations over Tomsk

### 3.1 Time series of the integrated stratospheric backscatter coefficient (1986–2015)

The 30-year time series of the integrated stratospheric backscatter coefficient $B_\pi^a$, obtained from the SAL lidar observations performed at $\lambda = 532$ nm in Tomsk from 1986 to 2015, is presented in Fig. 1. The backscatter coefficients are integrated over the 15–30 km stratospheric layer described above.

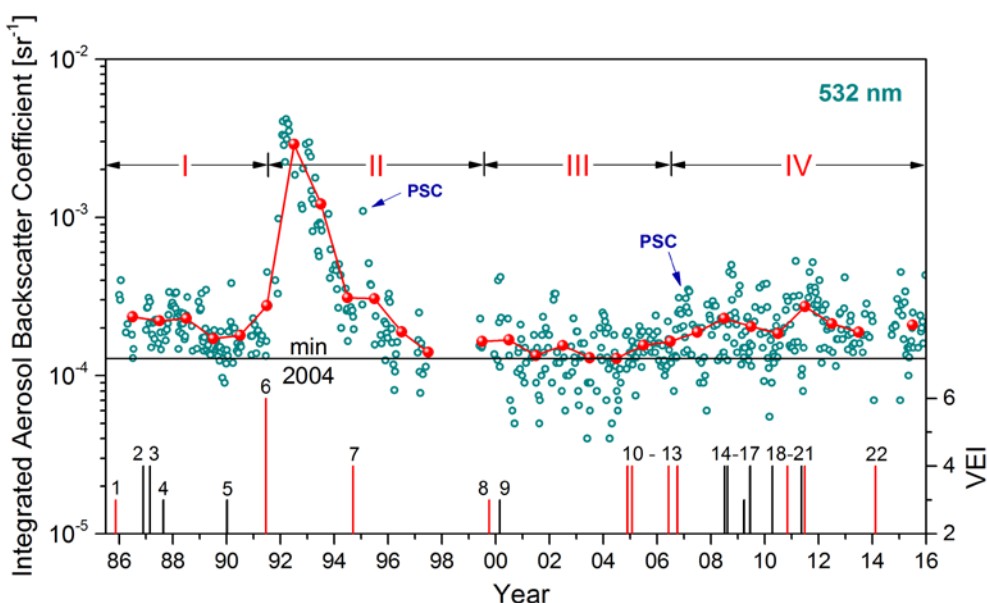

**Figure 1.** 30-year time series of the integrated stratospheric backscatter coefficient at $\lambda = 532$ nm over Tomsk between 15 and 30 km. Open dark-green circles denote the 10-day average $B_\pi^a$ values. Solid red circles show the annual average $B_\pi^a$ values assigned to 1 July of each year. Black and red vertical bars at the bottom of the figure indicate volcanic eruptions (ranked on VEI) which caused the SAL volcanogenic perturbations over Tomsk from 1986 to the present day (see also Table 1). The red bars correspond to tropical volcanic eruptions, whereas the black ones correspond to eruptions of extratropical volcanoes located in the Northern Hemisphere. The thin horizontal line in Fig. 1 indicates the minimum value of the annual average $B_\pi^a$ reached in 2004. PSC: polar stratospheric clouds.

We divided the time series into the following four intervals. The 1986–1991 period (I) reflects the final SAL relaxation after the explosive eruption of El Chichon volcano (Mexico, 29 March 1982, VEI = 5) together with small SAL perturbations after several less powerful volcanic eruptions during the period (see Table 1). The next 1991–1999 period (II) is mainly determined by the strong perturbation and subsequent long-term relaxation of the SAL after the Pinatubo eruption.

**Table 1.** List of volcanic eruptions that have caused the SAL volcanogenic perturbations detected over Tomsk from 1986 to the present day. The list was retrieved from the GVP data. $H_{MPA}$: maximum plume altitude.

| N | Date/Period | Volcano | Location | $H_{MPA}$, km | VEI |
|---|---|---|---|---|---|
| 1 | 13 Nov. 1985 | Nevado del Ruiz | Colombia (4.9° N, 75.3° W) | 31 | 3 |
| 2 | 20 Nov. 1986 | Chikurachki | Kuril Islands (50.3° N, 155.5° E) | 14 | 4 |
| 3 | 23 Feb. 1987 | Kliuchevskoi | Kamchatka (56.0° N, 160.6° E) | 13.7 | 4 |
| 4 | 28 Aug. 1987 | Cleveland | Alaska (52.8° N, 169.9° W) | 10.6 | 3 |
| 5 | 2 Jan. 1990 | Redoubt | Alaska (60. 5° N, 152.7° W) | 13.5 | 3 |
| 6 | 15 Jun. 1991 | Pinatubo | Philippines (15.1° N, 120.3° E) | 35 | **6** |
| 7 | 19 Sep. 1994 | Rabaul | Papua New Guinea (4.3° S, 152.2° E) | 21 | 4 |
| 8 | 5 Oct. 1999 | Guagua Pichincha | Ecuador (0.2° S, 78.6° W) | 20 | 3 |
| 9 | 26 Feb. 2000 | Hekla | Iceland (64.0° N, 19.7° W) | 15 | 3 |
| 10 | 24 Nov. 2004 | Manam | Papua New Guinea (4.1° S, 145.0° E) | 18 | 4 |
| 11 | 27 Jan. 2005 | Manam | Papua New Guinea (4.1° S, 145.0° E) | 24 | 4 |
| 12 | 20 May 2006 | Soufriere Hills | West Indies (16.7° N, 62.2° W) | 17 | 4 |
| 13 | 7 Oct. 2006 | Rabaul | Papua New Guinea (4.3° S, 152.2° E) | 18 | 4 |
| 14 | 12 Jul. 2008 | Okmok | Aleutian Islands (53.4° N, 168.1° W) | 15 | 4 |
| 15 | 7 Aug. 2008 | Kasatochi | Aleutian Islands (52.2° N, 175.5° W) | 14 | 4 |
| 16 | 22 Mar. 2009 | Redoubt | Alaska (60. 5° N, 152.7° W) | 20 | 3 |
| 17 | 11–16 Jun. 2009 | Sarychev Peak | Kuril Islands (48.1° N, 153.2° E) | 21 | 4 |
| 18 | 14–17 Apr. 2010 | Eyjafjallajökull | Iceland (63.6° N, 19.6° W) | 9 | 4 |
| 19 | 4–5 Nov. 2010 | Merapi | Indonesia (7.5° S, 110.4° E) | 18.3 | 4 |
| 20 | 21 May 2011 | Grimsvötn | Iceland (64.4° N, 17.3° W) | 20 | 4 |
| 21 | 13 Jun. 2011 | Nabro | Eritrea (13.4° N, 41.7° E) | 13.7 | 4 |
| 22 | 13 Feb. 2014 | Kelut | Indonesia (7.9° S, 112.3° W) | 17 | 4 |

The 1999–2006 period (III) is marked by reaching the background level of $B_\pi^a$ under comparatively small SAL volcanogenic perturbations. The last 2006–2015 period (IV) reflects an increase in $B_\pi^a$ (i.e. in stratospheric aerosol loading) due to an increase in volcanic activity. Table 1 contains all volcanic eruptions that have caused the SAL perturbations detected over Tomsk since 1986.

As noted above, the results of aerosol lidar observations at the SLS during the periods I and II were described by Zuev et al. (1998) and Zuev et al. (2001). Next, we consider the temporal dynamics of stratospheric aerosol loading over Tomsk during the periods III and IV.

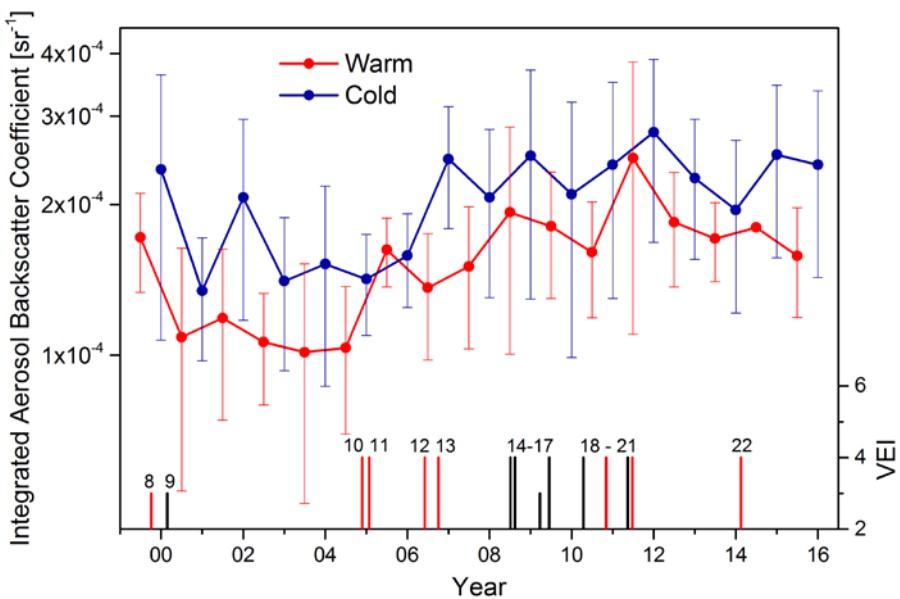

**Figure 2.** Inter-annual variations of $B_\pi^a$ values (in the stratosphere over Tomsk) separately averaged over the warm and cold half-years. The "warm" and "cold" average points are assigned to 1 June of the current year and 1 January of the next year, respectively. Black and red vertical bars at the bottom of the figure indicate volcanic eruptions as in Fig. 1 (see also Table 1). All error bars represent the standard deviation.

Low explosive volcanic activity during the comparatively long post-Pinatubo period led to a gradual reduction in volcanogenic aerosol loading of the stratosphere down to the background level of $B_\pi^a$ reached after 1998. Only the after-effect of the Rabaul volcano eruption (Papua New Guinea, 19 September 1994; VEI = 4) was definitely detected over Tomsk in the post-Pinatubo period (II). The minimum annual average $B_\pi^a = 1.29 \times 10^{-4}$ sr$^{-1}$ was reached in 2004. Thus, we can consider the state of the SAL over Tomsk as background during the period III, when the annual average $B_\pi^a$ values were less than those in the pre-Pinatubo period (1989–1991). Note that taking into account the spectral dependence of $B_\pi^a$, its

minimum annual average value observed in Tomsk at λ = 532 nm in 2004 was close to that determined for Garmisch-Partenkirchen at λ = 694 nm in 1979 and considered as the background by Trickl et al. (2013).

Both inter- and intra-annual variations of $B_{\pi}^{a}$ values in the stratosphere over Tomsk during the periods III and IV are presented in Figs. 2 and 3. Figure 2 shows the inter-annual $B_{\pi}^{a}$ variations separately averaged over the warm (April to September) and cold (October to March) half-years. The $B_{\pi}^{a}$ values are mostly higher in the cold half-year than those in the warm one. Furthermore, these "cold" and "warm" average $B_{\pi}^{a}$ values are modulated by the quasi-biennial oscillations (QBO; http://www.geo.fu-berlin.de/en/met/ag/strat/produkte/qbo/). The behavior of both $B_{\pi}^{a}$ curves is seen in Fig. 2 to clearly demonstrate the influence of the Brewer-Dobson circulation on the aerosol state of the mid-latitude stratosphere. Stratospheric aerosol loading is minimal in the warm half-year, when the zonal air mass transport dominates. On the other hand, the meridional air mass transport from tropical into extratropical (middle) latitudes intensifies in the cold half-year and, therefore, it provides the mid-latitude stratosphere with additional aerosol mass from the stratospheric tropical aerosol reservoir (Hitchman et al., 1994). Note that the minimum "warm" average $B_{\pi}^{a} = 1.01 \times 10^{-4}$ sr$^{-1}$ was reached in 2003 (Fig. 2).

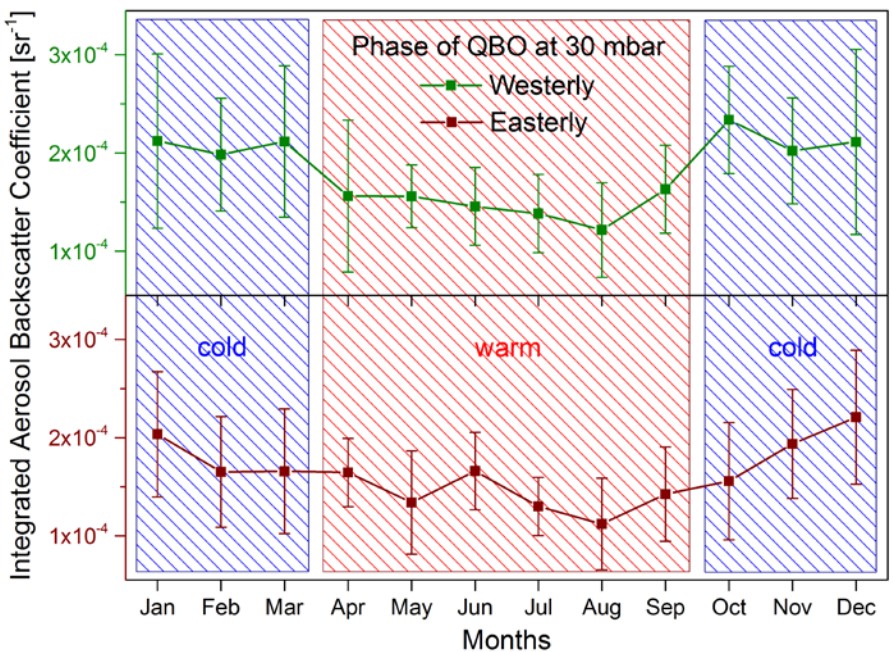

**Figure 3.** Intra-annual variation of the background monthly average $B_{\pi}^{a}$ values averaged over sixteen years (1999–2015) of the SAL lidar observations, excluding after-effects of volcanic eruptions during the period. The $B_{\pi}^{a}$ values were averaged separately for the westerly and easterly phases of the QBO characterized by zonal winds in the equatorial region at 30 mbar. All error bars represent the standard deviation.

The influence of the Brewer-Dobson circulation on background aerosol loading in the stratosphere over Tomsk can also be discovered by analyzing the intra-annual variations of the monthly average $B_\pi^a$ values. For example, Fig. 3 shows the $B_\pi^a$ values averaged over the period 1999–2015, separately for the westerly and easterly phases of the QBO (excluding after-effects of volcanic eruptions during the period). The monthly average $B_\pi^a$ data for March–June 2000 (after the Hekla eruption), August–November 2008 (after the Okmok and Kasatochi eruptions), August–October 2009 (after the Sarychev Peak eruption), and also April and August–October 2011 (after the Merapi, Grimsvötn, and Nabro eruptions) were not taken into account. The exclusion of these perturbed data allowed us to extend the analyzed period of the background aerosol loading variations up to 16 years and, therefore, to improve the statistical reliability of the $B_\pi^a$ data series. As seen in Fig. 3, aerosol loading of the mid-latitude stratosphere is maximal in the cold half-year, when the meridional air mass transport dominates (especially during the westerly phase of the QBO). Thus, both types of $B_\pi^a$ variation (Figs. 2 and 3) lead us to the same conclusion.

Turning to Fig. 1, one can see that there is a positive trend in stratospheric aerosol loading over Tomsk caused by an increase in the number of explosive volcanic eruptions with VEI = 4 during the last decade. A small increase in the $B_\pi^a$ value started in 2005 due to two Manam volcano eruptions occurred in Papua New Guinea closely spaced in time (Table 1). Soon after, in 2006, two relatively strong eruptions of the Soufriere Hills and Rabaul tropical volcanoes (Table 1) additionally enriched the stratospheric tropical aerosol reservoir. As a result, two corresponding volcanic aerosol peaks were observed in the stratosphere over Tomsk in October–December 2006 and January–March 2007 due to the meridional transport intensified in the cold period (Fig. 4). These peaks determined the increase of the annual average $B_\pi^a$ values in 2006 and 2007.

The further increase of the annual average $B_\pi^a$ value in 2008 was due to explosive eruptions of two northern volcanoes located in the Aleutian Islands: Okmok and Kasatochi (Table 1; Schmale et al., 2010). In the following two years, 2009–2010, there were only two eruptions of northern volcanoes with VEI = 4, namely Sarychev Peak (the Kuril Islands, 11 June 2009) and Eyjafjallajökull (Iceland, 14 April 2010). Note that the eruption plumes of Eyjafjallajökull mostly did not exceed the tropopause altitude over the volcano. This can explain a gradual decrease in stratospheric aerosol loading from 2008 to 2010 (see Fig. 1). However, a new increase in the annual average $B_\pi^a$ value was observed in 2011. This increase resulted from aerosol perturbations of the northern mid-latitude stratosphere after the explosive eruptions of Merapi, Grimsvötn, and Nabro volcanoes (all VEI = 4, Table 1). In the next sections we consider contributions of plumes from the volcanoes erupted in the period IV to the SAL volcanogenic perturbations over Tomsk, and also discuss other possible sources of the SAL perturbations.

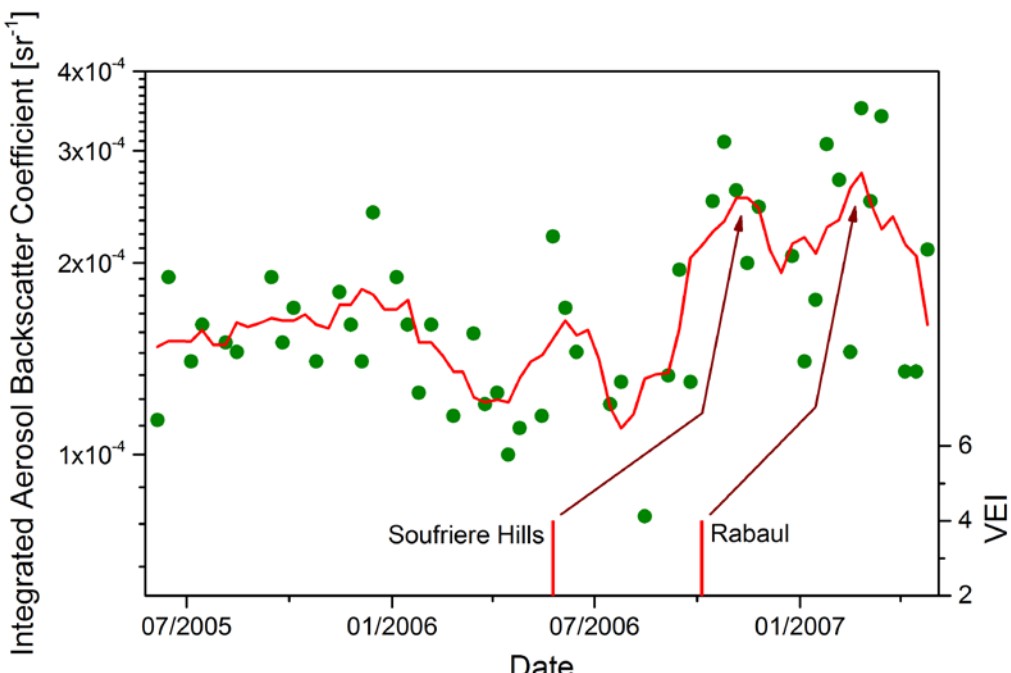

**Figure 4.** Two $B_\pi^a$ value peaks observed in the stratosphere over Tomsk in October–December 2006 and January–March 2007 after the Soufriere Hills and Rabaul eruptions, respectively (Table 1). Solid green circles denote the 10-day average $B_\pi^a$ values. The red curve denotes the $B_\pi^a$ values smoothed by five-point averaging.

**3.2 Detection of plumes from northern volcanoes in the stratosphere over Tomsk in 2008–2010**

Detection of volcanic plumes over Tomsk is based on: 1) the use of the scattering ratio $R(H)$ profiles retrieved from the lidar measurements between 12.5 and 30 km and 2) the assignment of observed aerosol layers to volcanic eruptions via the HYSPLIT model trajectory analysis, when possible.

**3.2.1 Okmok and Kasatochi**

In summer 2008, two Aleutian volcanoes Okmok and Kasatochi started to erupt at 19:43 UTC on 12 July and between 23:00 UTC on 7 August and 05:35 UTC on 8 August, respectively (both VEI = 4). The plumes from these volcanoes considerably perturbed the SAL over Tomsk from August to October 2008. Vertical profiles of the scattering ratio $R(H)$, showing the detection of the Okmok and Kasatochi plumes over Tomsk during these months, are presented in Fig. 5a as an example. Contemporaneous stratospheric aerosol observations at the Minsk CIS-LiNet station at λ = 532 nm revealed the similar SAL

perturbations over Minsk from July to October (Fig. 5b; Zuev et al., 2009). The after-effects of both volcano eruptions were detected in the stratosphere over Minsk and Tomsk up to December 2008.

Figure 6a shows the HYSPLIT air-mass backward trajectory started from the altitude of the $R(H)$ profile maximum (~15.1 km a.s.l.) over Tomsk on 8 August at 02:00 LT (or on 7 August at 19:00 UTC). The trajectory passed over Okmok volcano on the eruption day, 12 July, at the altitude $H_{\text{traj.}}^{\text{back.}} \approx 16.0$ km that is 1 km higher than the maximum plume altitude (MPA; Table 1) $H_{\text{MPA}}$ determined by the GVP. Figure 6b shows the backward trajectory started from the altitude of the $R(H)$ maximum (~16.3 km a.s.l.) over Tomsk on 2 September at 00:00 LT (1 September, 17:00 UTC). The trajectory passed over Kasatochi volcano on the eruption day, 7 August, at the altitude $H_{\text{traj.}}^{\text{back.}} \approx 16.4$ km that is 2.4 km higher than the GVP $H_{\text{MPA}}$ (Table 1). Our conclusion (based on the HYSPLIT trajectories), that the plumes from both volcanoes reached altitudes of ≥16 km, is consistent with different satellite observation data (Yang et al., 2010; Kristiansen et al., 2010; Prata et al., 2010). The inconsistency between the HYSPLIT $H_{\text{traj.}}^{\text{back.}}$ and GVP $H_{\text{MPA}}$ altitudes ($H_{\text{traj.}}^{\text{back.}}$ should normally be equal to or lower than $H_{\text{MPA}}$) is discussed in Sect. 4.

The HYSPLIT trajectory analysis also showed that both Okmok (Fig. 6a) and Kasatochi (Fig. 6b) plumes passed close to the Minsk lidar station. This explains the similarity of the $R(H)$ profiles presented in Fig. 5. Owing to the westerly transport of air masses, the volcanic plumes passed over Minsk three days earlier than over Tomsk. Figure 6c shows the backward trajectories which allowed us to find the connection between two aerosol layers (thick red lines in Fig. 5) detected over Minsk and Tomsk on 1 and 4 September, respectively. The more general and detailed analysis of the Okmok and Kasatochi plumes influence on the SAL state was made by Zuev et al. (2009) and later, e.g., by Bourassa et al. (2010) and Andersson et al. (2015).

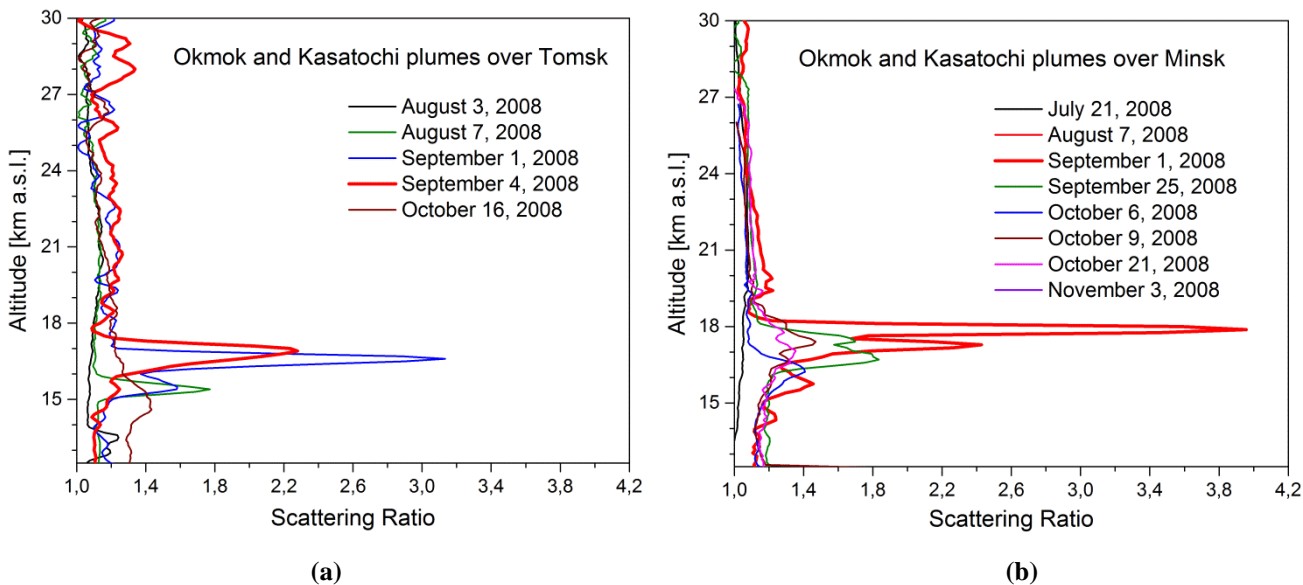

**Figure 5.** Detection of the Okmok and Kasatochi volcanic plumes in the stratosphere over **(a)** Tomsk (Russia) and **(b)** Minsk (Belarus). The volcanoes started to erupt in the Aleutian Islands on 12 July and 7 August 2008, respectively.

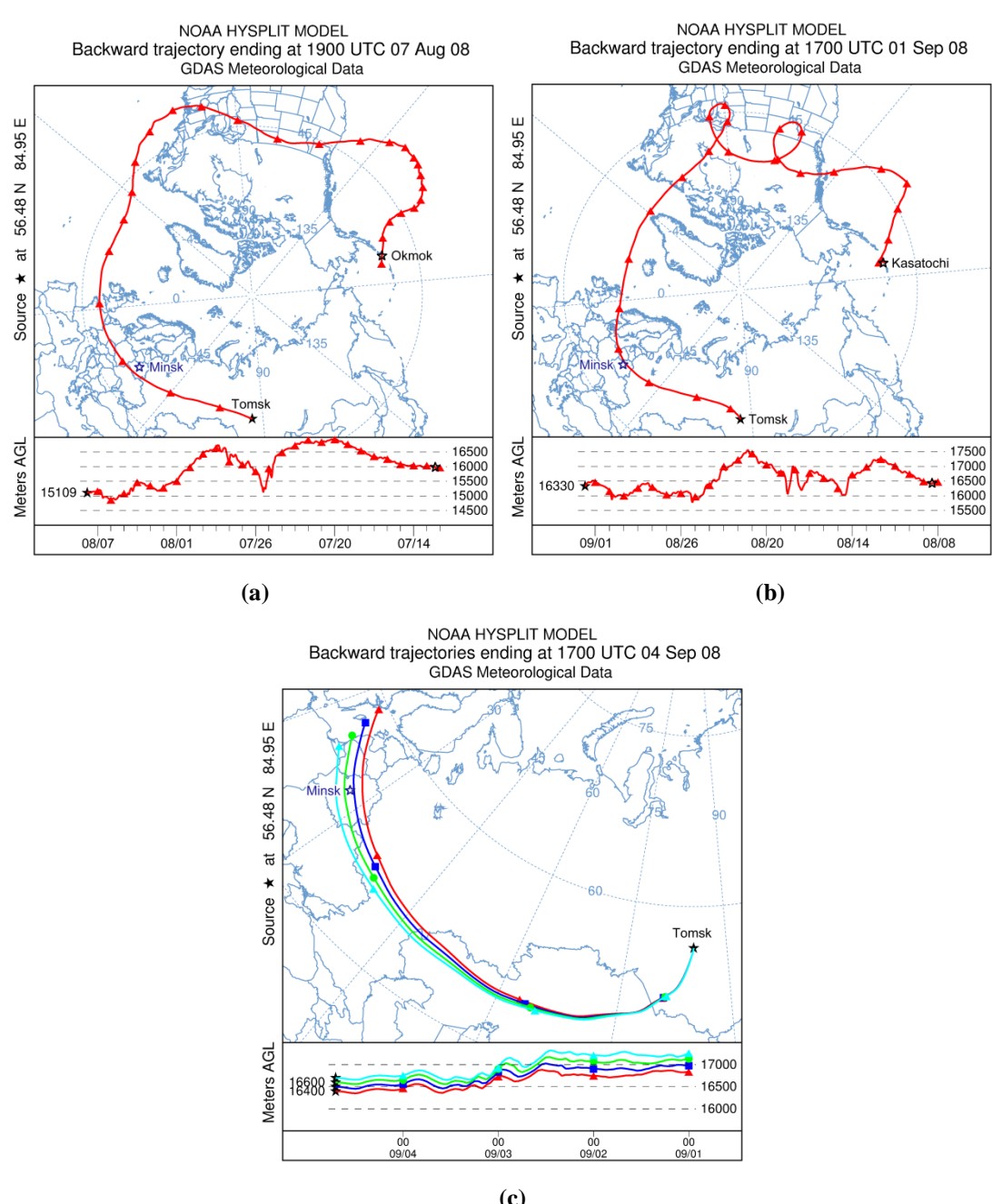

**Figure 6. (a)** Air-mass backward trajectory started from an altitude of ~15.1 km a.s.l. over Tomsk on 8 August 2008 at 02:00 LT (7 August, 19:00 UTC) and passed over Okmok volcano. **(b)** Air-mass backward trajectory started from an altitude of ~16.3 km a.s.l. over Tomsk on 2 September 2008 at 00:00 LT (1 September, 17:00 UTC) and passed over Kasatochi volcano. **(c)** Air-mass backward ensemble trajectories started from altitudes of 16.4–16.7 km a.s.l. over Tomsk on 5 September 2008 at 00:00 LT (4 September, 17:00 UTC) and passed close to Minsk.

It should be noted that due to the westerly zonal transport of air masses in the Northern Hemisphere lower stratosphere during summer seasons and vast geographical distance between Tomsk and the Aleutian Islands, both backward trajectories (Figs. 6a and 6b) could hardly be expected to be equal to or shorter than two weeks. Therefore, these trajectories are slightly longer than those usually used in the HYSPLIT model and, thus, can be considered only as probable ones. Nevertheless, we
made the trajectory analysis to assign the observed aerosol layers to the corresponding volcanic eruptions.

### 3.2.2 Redoubt and Sarychev Peak

The SAL perturbations over Tomsk in 2009 were caused by the eruptions of two northern volcanoes Redoubt (Alaska, 15 March to 4 April; VEI = 3) and Sarychev Peak (the Kuril Islands, 11–16 June; VEI = 4). The Redoubt plumes caused insignificant SAL perturbations over Tomsk during the first two weeks of May 2009 (Fig. 7). Stronger and longer-lasting
SAL perturbations were related to the Sarychev Peak volcano eruption. According to the GVP data, the MPA was within the range of 8–16 km or even reached 21 km (GVP, 2009). The Sarychev Peak plumes were reliably detected in the stratosphere over Tomsk during July and August (Fig. 8), and weakly observed up to November 2009. For a trajectory analysis, we considered an aerosol layer observed over Tomsk at an altitude of ~13.1 km on 7 July at 02:30 LT (6 July, 19:30 UTC). This layer is seen in Fig. 9 to be: 1) associated with the backward trajectory passed over Sarychev Peak volcano at an altitude of
~13.8 km on 15 June at the time of the eruption, 17:30 UTC; and 2) not associated with the after-effect of the Redoubt eruption.

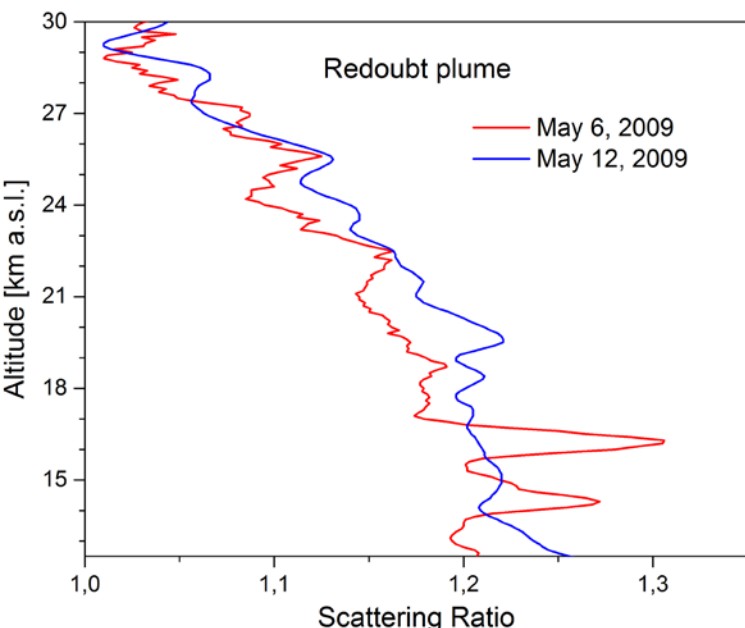

**Figure 7.** Detection of the Redoubt volcanic plumes in the stratosphere over Tomsk. The volcano erupted in Alaska from 15 March to 4 April 2009.

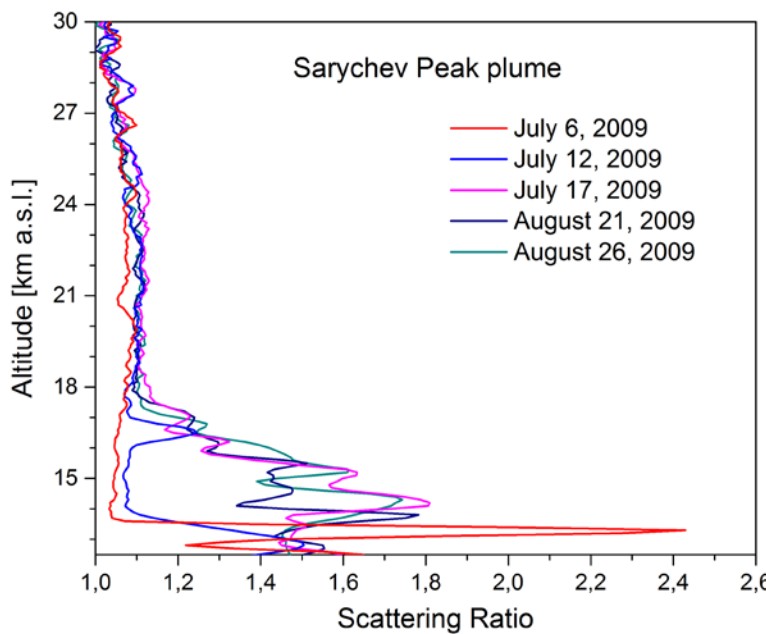

**Figure 8.** Detection of the Sarychev Peak volcanic plumes in the stratosphere over Tomsk. The volcano erupted in the Kuril Islands from 11 to 16 June 2009.

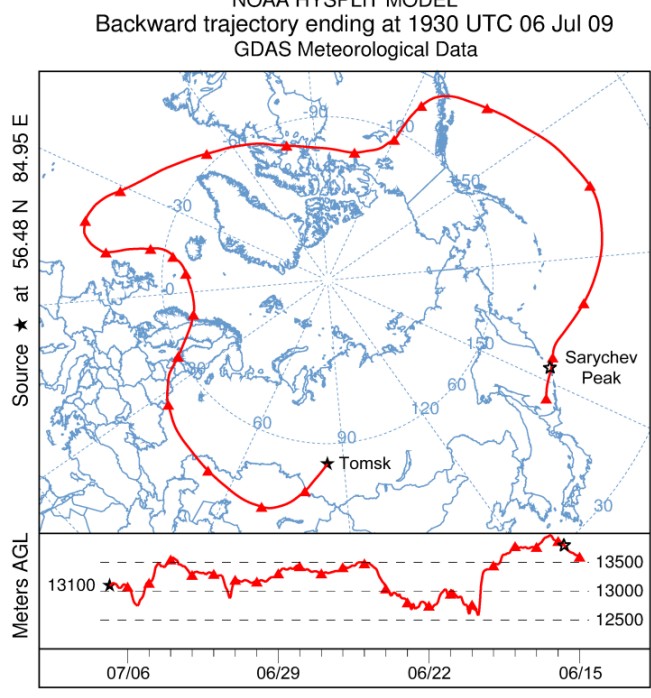

**Figure 9.** Air-mass backward trajectory started from an altitude of ~13.1 km a.s.l. over Tomsk on 7 July at 02:30 LT (6 July, 19:30 UTC).

### 3.2.3 Eyjafjallajökull

During April–May 2010, there was a series of explosive eruptions of the Icelandic volcano Eyjafjallajökull. These eruptions are noted for the subsequent extensive air travel disruption across large parts of Western Europe. According to the GVP data, the MPA occasionally reached 9 km (GVP, 2010), but did not exceed the local tropopause (GVP, 2010). However, lidar observations, performed in Tomsk on 20 and 26 April 2010, detected the presence of aerosol layers in the troposphere and lower stratosphere at altitudes up to 15 km (Fig. 10). As a comparison, aerosol lidar measurements at Garmisch-Partenkirchen revealed that the upper boundary of the observed aerosol layers from the Eyjafjallajökull volcanic plumes was ~14.3 km on 20 April, whereas the average altitude of the local tropopause was of ~10.2 km (Trickl et al., 2013).

Figure 11 shows the HYSPLIT air-mass backward ensemble trajectories started from altitudes of the detected aerosol layers (~11.1–14.6 km a.s.l.) over Tomsk on 21 April at 00:00 LT (20 April, 17:00 UTC). Only one trajectory (started from an altitude of ~11.6 km) directly passed over Eyjafjallajökull volcano on one of the eruption days, 16 April at 13:00 UTC, at the altitude $H_{\text{traj.}}^{\text{back.}} \approx 10.7$ km that is clearly higher than $H_{\text{MPA}} \leq 9$ km. The inconsistency between the HYSPLIT $H_{\text{traj.}}^{\text{back.}}$ and GVP $H_{\text{MPA}}$ altitudes is discussed in Sect. 4. The other trajectories passed south of the volcano. Note also that, according to the Icelandic meteorological station Keflavik, the local tropopause altitude went down to ~7 km on 16 April after 12:00 UTC (Trickl et al., 2013). Hence, the Eyjafjallajökull volcanic plumes reached altitudes of 8–9 km on that day and directly entered the local lower stratosphere.

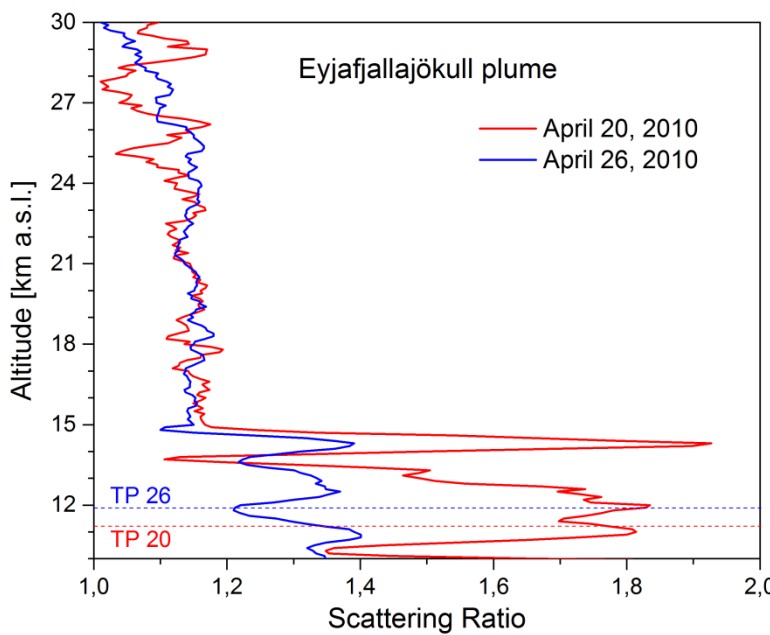

**Figure 10.** Detection of the Eyjafjallajökull volcanic plumes in the upper troposphere and lower stratosphere over Tomsk. The volcano erupted in Iceland from 14 to 17 April 2010. The tropopause altitude over Tomsk was of 11.2 km on 20 April and 11.9 km on 26 April.

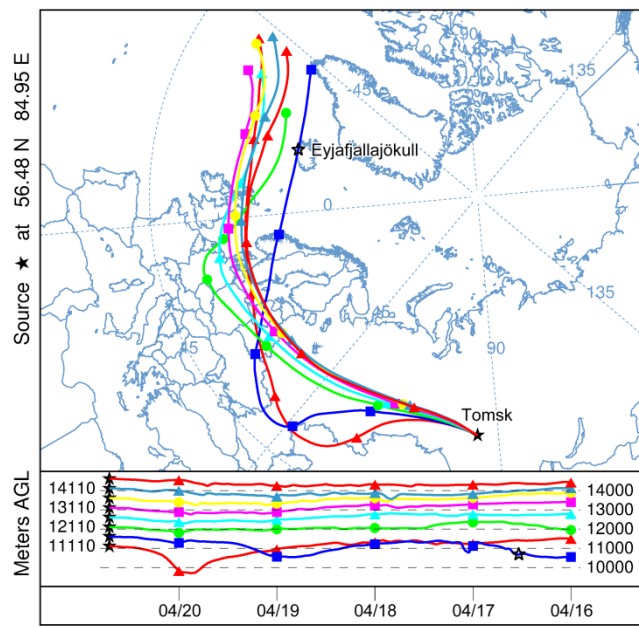

**Figure 11.** Air-mass backward ensemble trajectories started from altitudes of ~11.1–14.6 km a.s.l. over Tomsk on 21 April 2010 at 00:00 LT (20 April, 17:00 UTC) and passed over or south of Eyjafjallajökull volcano.

### 3.3 Detection of volcanic plumes in the stratosphere over Tomsk in 2011

High values of $B_\pi^a$ were detected during the SAL lidar observations in Tomsk from February to April and from August to December 2011. The "first" wave of the SAL perturbations in the winter-spring period was caused by the Merapi volcano eruption (Indonesia, 4–5 November 2010; VEI = 4), whereas the "second" wave was due to the eruptions of the northern volcano Grimsvötn (Iceland, 21 May 2011; VEI = 4) and the tropical volcano Nabro (Eritrea, 13 June 2011; VEI = 4).

### 3.3.1 Merapi

High values of $B_\pi^a$ were detected in the stratosphere over Tomsk from February to April 2011, i.e. 3–5 months after the Merapi volcano eruption. Figure 12 presents the observed after-effect of the Merapi eruption, i.e. several perturbed scattering ratio profiles retrieved from the SLS aerosol lidar measurements between 28 February and 18 April 2011. The Merapi plume (Table 1) supplied the stratospheric tropical reservoir with long-lived volcanic aerosol. The SAL perturbations, reflected by increased $B_\pi^a$ and $R(H)$ values during the winter and spring of 2011, were due to the meridional air mass transport from the

tropics into northern mid-latitudes in this cold period (see Sect. 3.1).

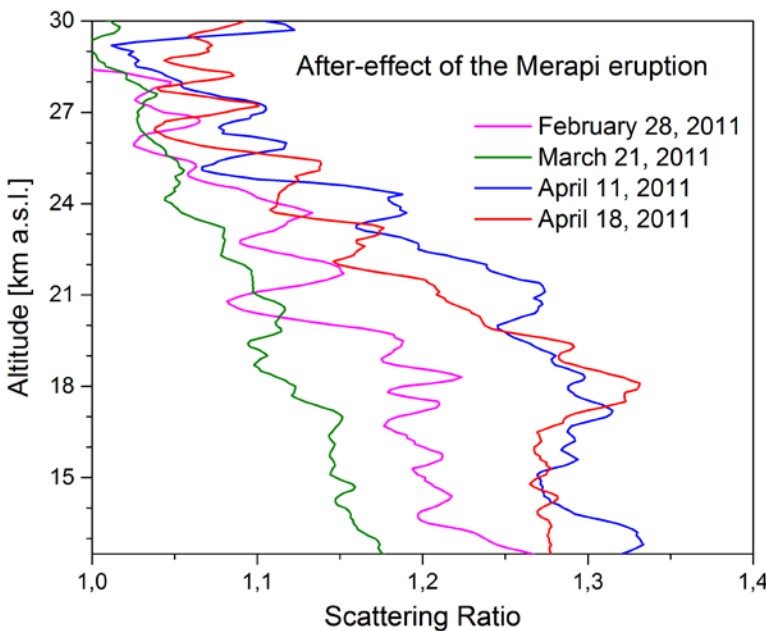

**Figure 12.** Perturbed scattering ratio profiles retrieved from the SLS aerosol lidar measurements in the winter-spring period of 2011.

### 3.3.2 Grimsvötn and Nabro

In 2011, two volcanoes with VEI = 4 Grimsvötn and Nabro started to erupt on 21 May at 19:25 UTC and 13 June after 22:00
5  UTC, respectively. According to the GVP data, Grimsvötn volcano erupted ash clouds and gases directly into the stratosphere at an altitude of 20 km, whereas the Nabro volcanic plume did not exceed the local tropopause altitude. Bourassa et al. (2012) showed that a considerable part of the Nabro volcanic aerosol and gases, erupted into the upper troposphere, was able to enter the mid-latitude stratosphere due to deep convection and vertical air transport associated with the strong Asian summer monsoon anticyclone. On the other hand, Vernier et al. (2013), Fromm et al. (2013), Fairlie et al.
10  (2014), Clarisse et al. (2014), and Penning de Vries et al. (2014) showed that the initial Nabro plume was directly injected into the lower stratosphere at altitudes up to 18 km (Fromm eat al., 2014). The SAL perturbations by volcanogenic aerosol after the eruptions of both volcanoes were observed in the lower stratosphere over Tomsk from August to November 2011 (Fig. 13). All the scattering ratio profiles shown in Fig. 13, with equal probability, represent superpositions of plumes from both Grimsvötn and Nabro volcanoes.

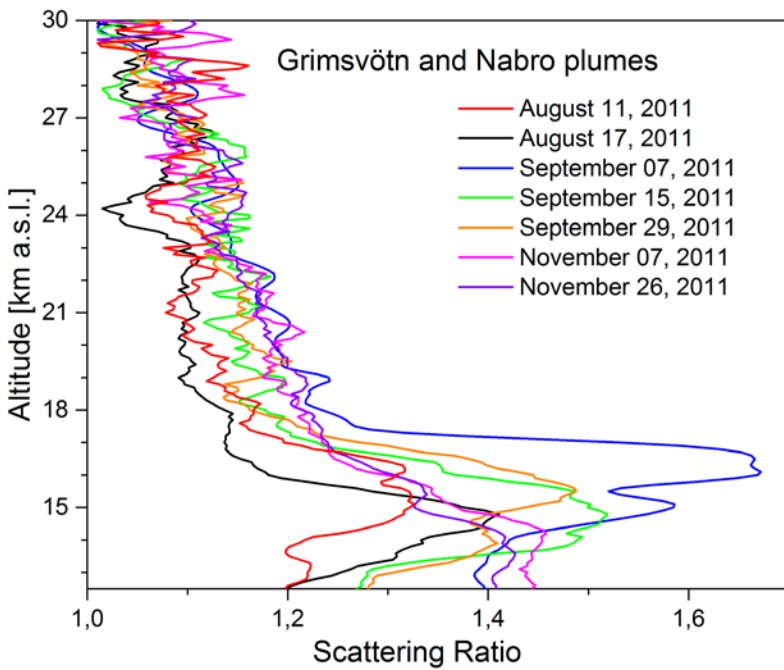

**Figure 13.** Detection of the Grimsvötn (Iceland) and Nabro (Eritrea) volcanic plumes in the stratosphere over Tomsk. The volcanoes started to erupt on 21 May and 13 June 2011, respectively.

### 3.4 Polar stratospheric clouds and the after-effect of the 2006 Rabaul eruption

Occasional perturbations of the mid-latitude SAL can also be related to the occurrence of polar stratospheric clouds (PSCs) in winter periods. PSCs are known to form at extremely low temperatures (lower than –78 °C) mainly on sulfuric acid ($H_2SO_4$) aerosols, acting as condensation nuclei and formed from sulfur dioxide ($SO_2$; Finlayson-Pitts and Pitts, 2000). Therefore, injections of volcanogenic $H_2SO_4$ aerosols or/and $SO_2$ into the stratosphere can lead to PSC formation, if the air temperature $< –78$ °C. The direct positive correlation between PSC formation and volcanogenic nitric and sulfur acid
aerosols loading was shown, e.g., by Rose et al. (2006). However, it should be noted that, in contrast to Rose et al. (2006), Fromm et al. (2003) showed little (or even negative) correlation between PSC events and ambient aerosol loading.

    The Northern Hemisphere stratosphere is usually cooled to the required low temperatures inside the Arctic stratospheric polar vortex in cold seasons (Newman, 2010). The Arctic polar vortex sometimes deforms and stretches to mid-latitudes including Siberian regions. Hence, the stratospheric temperature over Tomsk can occasionally be cooled lower than –78 °C,
when Tomsk is inside the polar vortex. Thus, the detection of aerosol layers in the stratosphere at extremely low temperatures can be indicative of the presence of PSCs.

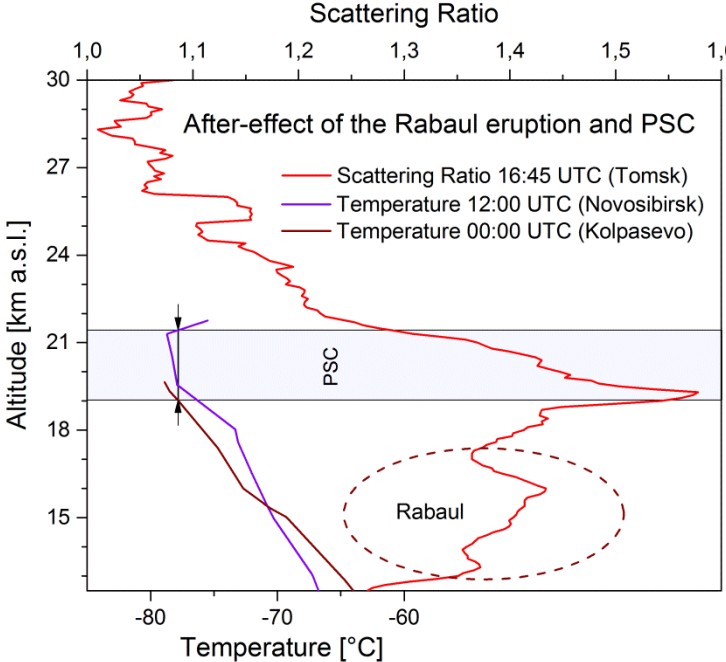

**Figure 14.** Detection of PSCs formed at extremely low temperatures (< –78 °C) in the stratosphere over Tomsk. Temperature profiles were obtained from radiosondes launched on 27 January 2007 in Kolpashevo (station 29231) at 00:00 UTC and in Novosibirsk (station 29634) at 12:00 UTC (WWW, 2007). The dashed ellipse denotes the after-effect of the Rabaul volcanic eruption occurred in Papua New Guinea on 7 October 2006.

The first lidar PSC observations over Tomsk were made at $\lambda$ = 1064 nm in January 1995 (Zuev and Smirnov, 1997). More precisely, some dense aerosol layers were detected at altitudes in the range of 15 to 19 km on 24 and 26 January. The maximum scattering ratio $R(H)$ was more than 14 at an altitude of 18.1 km. The stratospheric temperature was lower than –80 °C. The cold pool presence and PSC events near the Tomsk longitude during the northern winter of 1994/95 were also reported by Fromm et al. (1999). The formation of these dense PSCs was caused by high concentrations of residual post-Pinatubo aerosols.

Another event of PSCs over Tomsk was observed at $\lambda$ = 532 nm on 27 January 2007 (Fig. 14). As seen in Fig. 14, the maximum scattering ratio $R(H)$ was more than 1.55 at an altitude of 19.3 km. According to the data of the two nearest to Tomsk meteorological stations, launching radiosondes twice a day and situated in Novosibirsk (55.02° N, 82.92° E) and Kolpashevo (58.32° N, 82.92° E), the stratospheric temperature was lower than –78 °C at altitudes between 19 and 21.5 km (WWW, 2007) during the lidar measurements. High $R(H)$ values at altitudes in the range of 13 to 17 km were probably due to the winter aerosol supplying of the SAL from the stratospheric tropical aerosol reservoir enriched by the 2006 Rabaul eruption plume (Table 1, Fig. 14). Thus, PSCs were detected at least twice (in 1995 and 2007) during 30 years of stratospheric aerosol lidar measurements in Tomsk.

### 3.5 The latest SAL perturbations over Tomsk (2012–2015)

In summer 2011, the annual average $B_\pi^a$ value started to decrease and the SAL state over Tomsk started to relax to its background one (Fig. 1). However, a marked increase in $B_\pi^a$ value was observed in the winter of 2015. Figure 15 shows several perturbed scattering ratio profiles retrieved from the SLS aerosol lidar measurements between 29 January and 30 March, 2015. During that period of time, the Kelut volcano eruption could probably be a source of the SAL perturbations over Tomsk.

An explosive eruption of the tropical volcano Kelut occurred in East Java, Indonesia, on 13 February 2014 (Table 1). The MPA $H_{MPA}$ value for this eruption was initially estimated by both ground and space monitoring systems to be ~17 km. On the other hand, according to the data from the space-borne lidar CALIOP (Cloud-Aerosol LIdar with Orthogonal Polarization) onboard the CALIPSO (Cloud-Aerosol Lidar and Infrared Pathfinder Satellite Observation) satellite (http://www.nasa.gov/mission_pages/calipso/main/index.html), a rapidly rising portion of the Kelut plume ejected material up to an altitude exceeding ~26 km, i.e. directly into the tropical stratosphere. Most of the less rapidly rising plume portions remained lower, at altitudes of 19–20 km (GVP, 2014). The Kelut plume passed over the Indian Ocean to the West, toward the African continent, with a small deviation to the South. Sandhya et al. (2015) showed that a part of this plume could turn back and pass over the South end of Hindustan. Thus, the Kelut plume enriched the stratospheric tropical aerosol reservoir at least over the Indian Ocean. This led to the increasing annual average $B_\pi^a$ value in the northern mid-latitudes, including Tomsk, in 2015 (Fig. 1) due to the meridional aerosol transport.

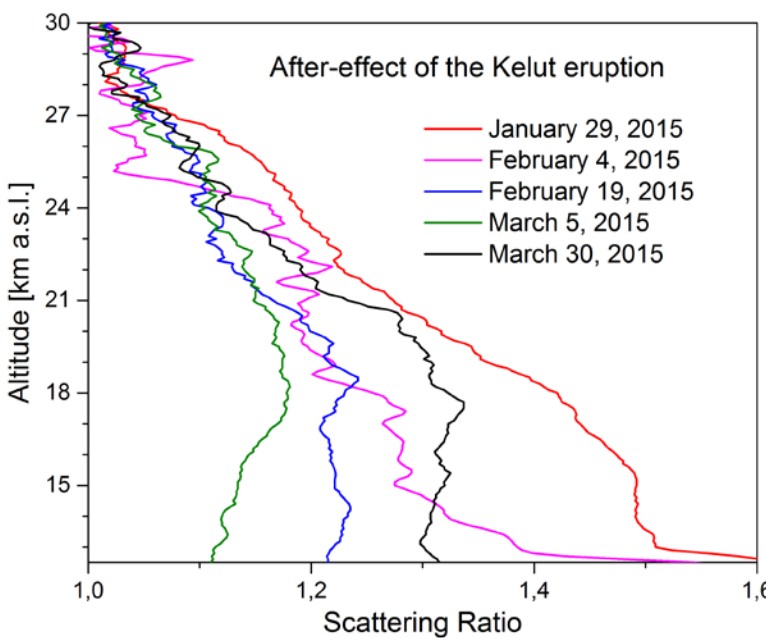

**Figure 15.** Perturbed scattering ratio profiles retrieved from the SLS aerosol lidar measurements between 29 January and 30 March 2015.

# 4 Discussion and conclusion

Thirty years (1986–2015) of lidar monitoring of the SAL state over Tomsk definitely showed that explosive eruptions with VEI ≥ 3 of both tropical and extratropical (northern) volcanoes represent the main cause of the northern mid-latitude SAL perturbations. Moreover, the tropical volcanoes, rather than the northern ones, have a dominant role in volcanogenic aerosol

loading of the mid-latitude stratosphere. Indeed, major explosive eruptions of tropical volcanoes are able to enrich the stratospheric tropical reservoir with volcanogenic aerosol. Additional aerosol loading of the tropical reservoir can usually lead to an increase in the annual average $B_\pi^a$ value in the Northern Hemisphere mid-latitude stratosphere via the meridional transport in the cold seasons (October to March; Hitchman et al., 1994). For example, plumes from both Merapi and Kelut volcanoes additionally supplied the stratospheric tropical reservoir with volcanic aerosol and gases (Table 1). As a result, the

increased annual average $B_\pi^a$ values (i.e. the SAL perturbations) were detected over Tomsk in 2011 and 2015, respectively (see Sects. 3.3.1 and 3.5). On the other hand, by contrast to tropical volcanoes, the narrow volcanic gas, aerosol, and ash plumes from northern volcanoes can either pass over a lidar station or pass it by. Owing to this, a certain part of northern volcanoes eruptions into the stratosphere did not perturb the SAL over Tomsk and, therefore, was not detected there. It is clear that an extensive network of lidar stations in the territory of the Russian Federation is required to obtain objective data

on the mid-latitude stratospheric aerosol loading.

In cases of the Eyjafjallajökull and probably Okmok and Kasatochi eruptions, the HYSPLIT air-mass backward trajectories, started from the altitudes of aerosol layers detected over Tomsk with the SLS aerosol lidar, passed over these volcanoes at altitudes $H_{\text{traj.}}^{\text{back.}}$ higher than their GVP MPAs (Sects. 3.2.1 and 3.2.3). On the other hand, the initial value $H_{\text{MPA}}$ for the Kelut volcano eruption was determined as about 17 km, but the measurements, made by the CALIOP space-borne

lidar onboard the CALIPSO satellite, clearly revealed that the rapidly rising portion of the Kelut plume reached an altitude of ~26 km that is 9 km higher than $H_{\text{MPA}}$ (GVP, 2014; Sect. 3.5). Based on these facts, we can offer the following explanation of the inconsistencies between the altitudes $H_{\text{traj.}}^{\text{back.}}$ and $H_{\text{MPA}}$. During Plinian explosive eruptions, solid and liquid ejecta, ash, and gas-vapor emissions intermix with each other, heat, and ascend inside the "convective thrust region" of an eruption column. Then the heated air together with erupted materials is known to expand, cool, and form the "umbrella region" of the

eruption column (Woods, 1988; Scase, 2009). The most heated fraction of gas-vapor emissions from the "convective thrust region" has the highest speed and, therefore, can penetrate through the higher-density "umbrella region" of the eruption column and reach altitudes higher than $H_{\text{MPA}}$ due to the cumulative (jet) effect (Raible et al., 2016). The secondary atmospheric $H_2SO_4$ aerosols are formed via oxidation of $SO_2$ contained in volcanic gas-vapor emissions. The currently available visual and radar methods for determining volcanic plume altitudes can detect only the large-sized volcanic ash

particles. At the same time, these methods are not sensitive to the small-sized atmospheric $H_2SO_4$ aerosols. Nevertheless, the submicron $H_2SO_4$ aerosol particles can be easily detected by lidars.

In addition to volcanoes, PSCs also represent a cause of significant SAL perturbations. However, the temperature condition required for PSC formation (air temperature should be < –78 °C) rarely holds in the mid-latitude stratosphere. Only two PSC events in January 1995 and January 2007 were observed over Tomsk during the 30-year period of lidar observations in Tomsk.

Extensive forest (bush) fires could be another cause of occasional increases of the $B_\pi^a$ value. Combustion products (gases and aerosol particles) can reach the stratospheric altitudes via convective ascent within pyro-cumulonimbus (pyroCb) clouds (see, e.g., Fromm et al., 2006). For example, the smoke plumes from the strong bush fire, occurred near the Australian city of Melbourne on 7 February 2009, were observed in the local stratosphere at an altitude of ~18 km (Siddaway and Petelina, 2011). Due to the climate warming, the number and intensity of massive forest fires have considerably increased in the last

few years (Wotton et al., 2010). For example, about 137 strong forest fires were registered in the Northwest Territories of Canada in July 2014 (CBC News, 2014). The smoke-filled air masses frequently enter the stratosphere over the South of Western Siberia from North America, where extensive forest fires occur. Their smoke plumes are most likely to be detected as the SAL perturbations over Tomsk. However, more detailed information about the pyroCb events is required for their correct identification. It is quite possible that some after-effects of strong forest fires occurred, e.g., in North America could

be detected over Tomsk, but not identified during lidar observations in Tomsk (1986–2015).

**Acknowledgements**

We thank V. N. Marichev and A. V. El'nikov for their arrangements for continuous monitoring of the stratospheric aerosol layer parameters over Tomsk and development of the stratospheric aerosol remote sensing methods. This work was performed with financial support from the Russian Science Foundation (Grant # 14-27.00022).

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
