# Peer review of "30-year lidar observations of the stratospheric aerosol layer state over Tomsk (Western Siberia, Russia)"

_Atmospheric Chemistry and Physics, 2016_

## Referee Comment (RC1) · Anonymous Referee #1 · 8 Nov 2016

GENERAL COMMENTS

This paper uses an extensive lidar dataset to characterize multiple eruptions over a northern mid-latitude site. The Russian site is quite far from other lidar sites so is unique is that respect. The analysis is straight forward and described well. I recommend that the paper be published with minor corrections. The content of the paper is well written although the English grammar should be edited.

SPECIFIC COMMENTS

Abstract: no comments.

1 Introduction: no comments

2 Lidar instruments and methods I would like to see a short description of the lidar photon-counting data acquisition electronics. Is it from a commercial vendor or built just for the lidar? Is signal induced noise, and counting saturation taken into account?

I would like to see more about the normalization. Is only a single altitude used? Is it 30 km? Many of the Scattering Ratio profiles shown in the paper haven't decreased to 1.0 at 30 km, so higher altitude data must have been used. Is there an objective way to do this?

Choosing H1 (lower attitude of the SAL) as 15 km seems reasonable, but other lidar groups have used the actual tropopause or tropopause + 1 km. Choosing this altitude can be complicated since there can be multiple tropopauses sometimes. It can also be complicated when there has been an eruption since the upper troposphere can have much more aerosol. But perhaps you can comment on how much of a difference it would make to lower the H1 altitude during background conditions.

3 Results of the SAL lidar observations over Tomsk

3.1 Time series of the integrated stratospheric backscatter coefficient (1986–2015) In Table 1 the maximum plume height is listed. How are these measured? The initial plume heights are not very accurate if done by naked-eye observations. Are these measured later with lidars?

There has been an ongoing discussion in the community about whether there is an annual cycle in SAL. Your Figure 2 shows a winter/summer ratio of about 1.35. Figure 3 is similar. This would be influenced by the choice of the H1 altitude. It would be interesting to calculate a ratio and error bar as your best estimate of an annual cycle.

In Figure 2 what do the error bars represent, one sigma of the spread of the data?

3.2 Detection of plumes from northern volcanoes in the stratosphere over Tomsk in 2008–2010

3.2.1 Okmok and Kasatochi

In Figure 5 there is a peak on September 4, 2000 at 27-30 km. Is that real? The altitude axis might be clearer with 1 km tic marks instead of 1.5 km.

3.2.2 Redoubt and Sarychev Peak No comments.

3.2.3 Eyjafjallajökull No comments.

3.3 Detection of volcanic plumes in the stratosphere over Tomsk in 2011

3.3.1 Merapi No comments.

3.3.2 Grimsvötn and Nabro No comments.

3.4 Polar stratospheric clouds No comments.

3.5 The latest SAL perturbations over Tomsk (2012–2015) No comments.

4 Discussion and conclusion "The Happy Camp Complex fire consumed more than 134 acres ($\sim$543 km2)...". Acres are much smaller than km2, something is wrong with the areas.

Figures 6 (a), 6 (b), 9, and 11. I am amazed that all four trajectories almost exactly cross over the volcanoes after so many days. Did you find an optimum altitude or something that gave you the best trajectory? Were the trajectories sensitive to the initial conditions?
* * *

---

## Referee Comment (RC2) · M. Fromm (Referee) · 16 Nov 2016

Review of Zuev et al., "30-year lidar observations of the stratospheric aerosol layer state over Tomsk (Western Siberia, Russia)"

Reviewer: Mike Fromm

This review consists of this review document and the manuscript annotated with comment bubbles. The comments in this document reflect both major and minor concerns. Technical suggestions are identified in the comment-bubble-annotated manuscript.

This manuscript gives a wide-ranging analysis of the aerosol lidar data collected at a western Siberia location. The long temporal span of this data set, for a location far
removed from other ground-based lidars, is a welcome addition to the globe's sparse stratospheric aerosol archive. Considering that these data are presumably still being collected adds value for continued monitoring of the upper troposphere and lower stratosphere. Consequently this is a potentially critical work that may offer significant value to our understanding. The authors' approach is similar to many prior works that examine a particular lidar's data set in the context of other data sets and meteorological analyses. They present some individual lidar profiles, long-term time series, and annual cycle analyses. This is a very appropriate study for Atmospheric Chemistry and Physics.

Along with these assets, the paper has several major weaknesses. My assessment is that the weaknesses substantially limit the value of the manuscript in its present form. For the paper to merit publication, attention to these major concerns must be paid.

One major concern is a significant absence of recognition of work that has direct bearing on the analyses and conclusions presented herein. The effect goes beyond that of showing insufficient background work. It extends to the realization that critical scientific aspects of stratospheric aerosol and cloud have not been taken into account in the interpretation of the Tomsk lidar data. Detailed examples are given below. Another major concern is that the Tomsk data may be biased with respect to other similar lidar data sets, as called out specifically in a comment below. The manuscript offers no direct or even indirect comparisons of the Tomsk lidar data with any other aerosol data sets; hence it is not possible to assess the accuracy of the aerosol data in the full range of stratospheric aerosol loading. Another concern is that the authors utilize a local measurement data set of total ozone abundance, but without any qualification. The reader is left with great uncertainty as to the robustness of the results and certain interpretations the authors give for certain phenomena.

Below is a list of questions and concerns, in page and line number order.

Somewhere in this paper the recent review paper by Kremser et al. should be men-
tioned and cited. Kremser, S., et al. (2016), Stratospheric aerosol—Observations, processes, and impact on climate, Rev. Geophys., 54,doi:10.1002/2015RG000511.

There is no citation of Vernier et al (GRL, 2011), Ridley et al. (GRL, 2014), Santer et al. (Nature Geo., 2014) and some other very relevant recent papers on volcanoes and recent stratospheric aerosol layer trends.

P2, L10. Consider citing Jäger, H. and Wege, K.: Stratospheric Ozone Depletion at Northern Midlatitudes after Major Volcanic Eruptions, J. Atmos. Chem., 10, 273-287, 1990.

P2, L5. "vulcanian" refers to a specific style of eruption, the characteristics of which are not associated with stratospheric injection. Is the term even needed to make the point of this sentence?

P3, L19. How about measurement frequency? What was the lidar operation frequency and regularity? It would be important for the reader to know what is typical for the number of profiles that go into the 10-day average.

P3, L23. Consider giving the name of the PMT manufacturer.

P4, L4. This statement arouses my curiosity to know if this lidar made any observations of the Chelyabinsk bolide plume in 2013. It was observed by satellite aerosol profilers above 30 km. Gorkavyi, N., D. F. Rault, P. A. Newman,A. M. da Silva, and A. E. Dudorov (2013), New stratosphericdust belt due to the Chelyabinsk bolide, Geophys. Res. Lett., 40,4728–4733, doi:10.1002/grl.50788.

P4, L10. What does the pi character refer to? Please consider stating its meaning.

P4, L20. This is understandable and defensible. But the median tropopause at Tomsk is such that a lot of the lowermost stratosphere is not sampled with the fixed lower limit of 15 km. For instance, many volcanic plumes just above the tropopause have occurred in recent years (Kasatochi is one example). And pyrocumulonimbus smoke plumes are routine in summer, but usually between the tropopause and ∼15 km. It

[Figure]

might be worthwhile to try another lower-boundary, for instance based on potential temperature or a tropopause-relative height offset.

Figure 2. The error/uncertainty bars are not defined or even mentioned. Moreover, however they are defined, the majority of them extend farther than the winter/summer range. This suggests to me that this pattern of monthly averages is not particularly repeatable. Auth should consider discussion of the robustness of this pattern in light of the relatively large uncertainty. The feature that stands out here is the March average and uncertainty. Can the authors explain why this month's aerosol amount is so large and variable?

P6, L4. The minimum integrated backscatter coefficient value in Figure 1 and Trickl et al.'s background value are not equal, in contrast to the authors' claim. Figure 1's value is between 1e-4 and 2e-4. Trickl et al.'s 1979 background value is 5e-5, smaller by roughly a factor of two. Zuev et al.'s Pinatubo peak matches well with Trickl et al., but the comparable quiescent-period values are off by about a factor of 2. This sentence should be corrected. The differences in the background values should be acknowledged and discussed. Trickl integrated from a tropopause-relative altitude that probably includes more of the lowermost stratosphere than Zuev et al. yet their integral is much smaller. The wavelength difference is seemingly too small to account for the difference.

P7, L14-19. Discussion of the Brewer Dobson circulation and its impact on the extra-tropical SAL. Presumably the point here is that the pattern shown by Fig. 2 is consistent with the generalizations here such that a citation is needed for work(s) that show this intra-annual tendency.

Figure 3. What is the averaging period of the data points? Since error bars were shown for Figure 2, they should also be shown here, and the implications discussed.

P8, L17-18. This statement does not reconcile with VEI=4. I recognize that VEI=4 is in the GVP database. So this sentence probably should call out the apparent discrepancy.

P10, L3. The trajectory analysis is largely unnecessary. There is no need to run trajectories for the 8 Aug observation; it's obvious that layer can't be from Kasatochi (and there is no other plausible source). For the September observation (and latter ones), the stratosphere at that time had Okmok and Kasatochi aerosols distributed all over. If the Tomsk back trajectory intersected with any of the Okmok layers prior to its pass over the Kasatochi volcano, that is a more plausible attribution than a 3.5 week trajectory connection.

P10. The impression I got from reading this analysis was that Zuev et al. are attempting to draw general conclusions about the Kasatochi and Okmok plumes and the SAL. If they are instead limiting their assessment of the SAL to just where the Tomsk observations are, they should make that explicit. The more general SAL analysis of these plumes is already published but not cited in this manuscript. E.g. Bourassa et al., Anderssen et al., Kravitz et al. These make clear that the Kasatochi and Okmok plumes were observed at all altitudes from the tropopause to nearly 19 km. Andersson et al. (2015), Significant radiative impact of volcanic aerosol in the lowermost stratosphere, DOI: 10.1038/ncomms8692. Bourassa, A., D. Degenstein, B. Elash, and E. Llewellyn (2010), Evolution of the stratospheric aerosol enhancement following the eruptions of Okmok and Kasatochi: Odin-OSIRIS measurements, J. Geophys. Res., 115, D00L03, doi:10.1029/2009JD013274. Kravitz, B., A. Robock, and A. Bourassa (2010), Negligible climatic effects from the 2008 Okmok and Kasatochi volcanic eruptions, J. Geophys. Res., 115, D00L05, doi:10.1029/2009JD013525.

P10, L12. My impression is that Zuev et al. are attempting to assess the accuracy of the injection height as reported by the GVP. The point made here, and below, regarding the accuracy of the Hmpa for the two eruptions, is of little consequence. I also get the impression that they are using the altitude of the back trajectory endpoint over the volcano as a point of comparison with Hmpa. If I have the wrong impression, perhaps other readers will be similarly affected. So I ask the authors to clarify the wording here. Otherwise, giving the precise altitude of a 3-4 week old trajectory much weight

asks more of the trajectory model and the weather analyses than they can promise. Secondly, the Hmpa data in the GVP database is neither tightly constrained. Thirdly, it can be shown easily with CALIOP data that the Hmpa for these two eruptions is even farther off the mark than this analysis suggests. CALIOP data within a few days of each eruption shows that the injection altitude was at least 17 km (Okmok) and 18 km (Kasatochi). These data would offer a much more compelling analysis for this argument than the Tomsk data.

P11, L15. This is way too precise and exclusive. The Sarychev Peak eruption spanned several days, as shown in their table. The time of a 3+ week back trajectory is by its nature too uncertain to permit such a definitive connection.

Figure 10. These strong stratospheric layers at 15 km and their source are not subjected to a back trajectory test. Given that they are a much shorter time post eruption than the prior examples, it would critical to show if they connect to the volcano. If Eyja did not inject material this far above the tropopause, where did this material come from?

P15, L13. Regarding the connection between the weak layer in Figure 12, localized reduced total ozone, and Merapi, a stronger candidate would be Arctic O3 depletion. See Manney et al., (Nature, 2011) "Unprecedented Arctic Ozone Loss in 2011" doi:10.1038/nature10556

P16, L4. This conclusion regarding Nabro's injection height has been exhaustively disputed. There are no citations here of the papers demonstrating the classic, direct injection of Nabro to the stratosphere. http://science.sciencemag.org/content/339/6120/647.4 http://science.sciencemag.org/content/339/6120/647.3 The Bourassa et al. scenario was shown to be improbable by Fairlie et al.(2014). Clarisse et al. (2014), Fairlie et al. (2014), and Penning de Vries et al. (2014) showed, using a variety of satellite data, that direct injection above the tropopause was more consistent with these data

than the indirect path via the Monsoon. Fromm et al. (2014) established a root cause for the misattribution to the Asian Monsoon pathway and made connections with prior papers on other volcanic stratospheric aerosol discrepancies. Hence the full weight of Nabro-related papers gives a very different perspective than what is documented here. I would ask the authors to more fully capture these various works in their presentation on Nabro. http://www.atmos-chem-phys.net/14/7045/2014/ http://www.atmos-chem-phys.net/14/3095/2014/ http://www.atmos-chem-phys.net/14/8149/2014/ http://onlinelibrary.wiley.com/doi/10.1002/2014JD021507/full

P16, L18. Often? I'm not aware of any literature showing any direct positive correlation between PSC formation and volcanic sulfur loading. Hence a citation is needed. Fromm et al. (2003) actually showed little (or even negative) correlation between PSC frequency and ambient aerosol loading. Hence it would be important for the authors to substantiate the claim they make here. Fromm, M., J. Alfred, and M. Pitts, A unified, long-term, high-latitude stratospheric aerosol and cloud database using SAM II, SAGE II, and POAM II/III data: Algorithm description, database definition, and climatology, J. Geophys. Res., 108(D12), 4366, doi:10.1029/2002JD002772, 2003.

P17, L6. Not necessarily. In fact, spatial correlations between total-O3 minima (so-called ozone mini-holes) and PSCs are routinely attributable to synoptic-scale dynamics. See Hood et al. and Teitelbaum et al. Exploring polar stratospheric cloud and ozone minihole formation: The primary importance of synoptic scale flow perturbations, Teitelbaum, H., M. Fromm, & M. Moustaoui, J. Geophys. Res., 106, 28,173-28,188, 2001. Origin of extreme ozone minima at middle to high northern latitudes, Hood, L. L., B.E. Soukharev, M. Fromm, and J.P. McCormack, J. Geophys. Res., 106, 20,925-20,940, 2001.

P17, L17. This PSC is consistent with the findings of Fromm et al. (1999) who showed that the cold pool and PSC frequency in the northern winter of 1994/95 was located near the Tomsk longitude. http://onlinelibrary.wiley.com/doi/10.1029/1999JD900273/epdf

P17, L19. Please see the prior comment regarding mini-holes.

P17, L20. This seems to be quite unlikely considering that PSCs are formed inside the vortex, which represents air isolated from the extratropics. Rabaul aerosols, introduced just a few months before the northern 1994/95 vortex season, would not likely have been meridionally transported that far north in time to be in place before the vortex formed and isolation was in place.

P18, L2. It seems highly unlikely (or at least hard to prove) to argue for a PSC in the northern polar vortex one season after a southern hemisphere volcanic eruption. It would be best to state this as speculative, if the statement is to remain.

P18, L6. See the prior comment about mini-holes. At this time in Jan 2007, a localized o3 minimum was at Tomsk's longitude. The ozone signature was likely an artifact of local dynamics. http://exp-studies.tor.ec.gc.ca/cgi-bin/selectMap?lang=e&type1=du&day1=27&month1=01&year1=2007&howmany1=1&interval1=1&intervalunit1=d&hem1=

P18, L26. By postulating that pyroCb smoke could increase "annual average" stratospheric aerosol, it would imply that prior research has come to similar conclusions. I'm not aware of any such finding. Please modify this statement appropriately.

P19, L13. "always" was not proved or demonstrated herein. Hence a citation is needed, or this point should be restated or removed.

P19, L18. It's not clear what is meant here. All volcanic plumes are represented as an initial point source. Please clarify.

P20, L15. What is meant by "thermal speed?" Please clarify.

P20, L15. This is confusing. A plume is buoyant because it is less dense than the surrounding air. This makes it sound as if the surrounding umbrella region is less dense.

P20, L24. "drifting" needs to be clarified. It implies to me that the ozone feature is

drifting with the wind, whereas the papers previously offered in these comments show that the minihole is tied to dynamics and thus move with the speed of the synoptic-scale wave, not the wind.

P20, L29. The impact of pyroCb smoke on annual average stratospheric aerosol has not been shown in the literature to my knowledge. Please cite a paper or modify this statement accordingly.

P20, L32. The measures in acres and square km are not equivalent. Please correct this.

P21, L2. What is the reason to call out the Happy Camp fire?

P21, L4. Regarding pyroCb smoke, I believe that there would be many occasions through the years for stratospheric observations at Tomsk. But of course the frequency of such observations would decrease rapidly with altitude above the tropopause. The highest smoke layers in the northern hemisphere to my knowledge are ∼19 km. They are much more likely to be at 12-15 km. Perhaps it would be to your advantage to directly employ the local tropopause height in the search for pyroCb smoke instead of a fixed altitude (e.g. 15 km) that is generally ∼4 km above the tropopause.

Technical Comments See the accompanying pdf of the manuscript.

Please also note the supplement to this comment:
http://www.atmos-chem-phys-discuss.net/acp-2016-792/acp-2016-792-RC2-supplement.pdf

**Supplement:**

[revised manuscript text omitted]

---

## Author Comment (AC1) · 20 Jan 2017

**Manuscript Number:** acp-2016-792
**Manuscript Type:** research article
**Title:** 30-year lidar observations of the stratospheric aerosol layer state over Tomsk (Western Siberia, Russia)

We thank Referee 1 for valuable comments, questions and suggestions which will allow us to improve our manuscript.

**Point-by-point response to Referee 1**

**2 Lidar instruments and methods**

**Comment:** I would like to see a short description of the lidar photon-counting data acquisition electronics. Is it from a commercial vendor or built just for the lidar?

**Our response:** The SLS aerosol channel data acquisition electronics we used is of in-house design and manufacture. A more detailed technical description of the SLS aerosol channel and its data acquisition electronics can be found, e.g., in (Burlakov et al., 2010). We will include the reference in the final version of our manuscript.

Burlakov, V. D., Dolgii, S. I., and Nevzorov, A. V.: A three-frequency Lidar for sensing microstructure characteristics of stratospheric aerosols, Instrum. Exp. Tech., 53, 890–894, doi:10.1134/S0020441210060230, 2010.

**We added the following sentence to the revised manuscript:**

"A more detailed technical description of the SLS aerosol channel and its data acquisition electronics can be found, e.g., in (Burlakov et al., 2010)."
[**Page 4, lines 10 and 11, revised manuscript**]

**Comment:** Is signal induced noise, and counting saturation taken into account?

**Response:** Yes. We take into account both photomultiplier tube (PMT) afterpulses and photon-counting saturation. Furthermore, current pulses from a PMT are fed to a broadband amplifier and a differential amplitude discriminator. The latter allows controlling the lower and upper discrimination thresholds of dark-current pulses of a particular PMT specimen under the conditions of real background illumination, i.e., selection of the optimal discrimination thresholds for increasing the signal-to-noise ratio (see, please, Sect. "Technical description of the lidar" in the mentioned paper Burlakov et al., 2010).

**Comment:** I would like to see more about the normalization. Is only a single altitude used? Is it 30 km? Many of the Scattering Ratio profiles shown in the paper haven't decreased to 1.0 at 30 km, so higher altitude data must have been used. Is there an objective way to do this?

**Response:** The SLS aerosol channel makes it possible to receive almost undisturbed backscattered signals from altitudes of ~ 40–45 km. At higher altitudes, the signal-to-noise ratio is too low. Therefore, altitudes of ~30–35 km, where the stratosphere is considered to be aerosol-free, were used as the calibration altitudes.

**Comment:** Choosing H1 (lower attitude of the SAL) as 15 km seems reasonable, but other lidar groups have used the actual tropopause or tropopause + 1 km. Choosing this altitude can be complicated since there can be multiple tropopauses sometimes. It can also be complicated when there has been an eruption since the upper troposphere can have much more aerosol. But perhaps you can comment on how much of a difference it would make to lower the H1 altitude during background conditions.

**Response:** Tomsk is located near the southern boundary of subarctic latitudes, where the tropopause altitude can vary significantly (from ~11 to 13 km, depending on season), e.g., due to migration of the Arctic stratospheric jet stream within our (Tomsk) region. Sometimes one can observe a double (or even multiple) tropopause. Therefore, we consciously removed the interval of the tropopause altitude variations to observe the stratospheric perturbations only. Probably, some information about the lowermost stratospheric aerosol perturbations could be missed.

**Instead of**
"In our case, the tropopause altitude over Tomsk varies from ~11 to 13 km, depending on season, and therefore, we set $H_1 = 15$ km."

"Tomsk is located near the southern boundary of subarctic latitudes, where the tropopause altitude can significantly vary, e.g., due to migration of the Arctic stratospheric jet stream within the Tomsk region. Sometimes one can observe a double (or even multiple) tropopause. For this reason, we consciously removed the interval of the tropopause altitude variations to observe the stratospheric perturbations only. As the tropopause altitude over Tomsk varies from ~11 to 13 km, depending on season, we set $H_1 = 15$ km."
[**Page 4, lines 23–26, revised manuscript**]

**3 Results of the SAL lidar observations over Tomsk**

**3.1 Time series of the integrated stratospheric backscatter coefficient (1986–2015)**

**Comment:** In Table 1 the maximum plume height is listed. How are these measured? The initial plume heights are not very accurate if done by naked-eye observations. Are these measured later with lidars?

**Response:** The maximum plume altitudes (MPAs) presented in Table 1 were taken from the Smithsonian Institution Global Volcanism Program. The Smithsonian MPA values were determined from the pooled analysis of visual and radar observations. When it was possible, the MPAs could also be determined with space-borne lidars more accurately compared to the mentioned observation methods. Considering in Sect. 3.5 the 2014 Kelut eruption as an example, we discussed the difference between MPA values determined via the space-borne lidar CALIOP and visual/radar observations.

**Comment:** There has been an ongoing discussion in the community about whether there is an annual cycle in SAL. Your Figure 2 shows a winter/summer ratio of about 1.35. Figure 3 is similar. This would be influenced by the choice of the H1 altitude. It would be interesting to calculate a ratio and error bar as your best estimate of an annual cycle. In Figure 2 what do the error bars represent, one sigma of the spread of the data?

**Response:** First of all, Figure 2 is now Figure 3 and, conversely, Figure 3 is now Figure 2 in the revised manuscript. We have extended the analyzed period of the background aerosol loading variations over Tomsk up to 16 years (1999–2015), instead of the period 2000–2006 in the old version of our manuscript. The monthly average $B_\pi^a$ data for March–June 2000 (after the Hekla eruption), August–November 2008 (after the Okmok and Kasatochi eruptions), August–October 2009 (after the Sarychev Peak eruption), and also April and August–October 2011 (after the Merapi, Grimsvötn, and Nabro eruptions) were not taken into account. The exclusion of these perturbed data allowed us to extend the analyzed period of the background aerosol loading variations and, therefore, to improve the statistical reliability of the $B_\pi^a$ data series. The $B_\pi^a$ values were averaged separately for the westerly and easterly phases of the quasi-biennial oscillations (QBO) characterized by zonal winds in the equatorial region at 30 mbar (Fig. 3, ex-Fig. 2). Based on these 16-year averaged values of the monthly average $B_\pi^a$, we show that an annual cycle in SAL exists. Moreover, based on the inter-annual $B_\pi^a$ variations (Fig. 2, ex-Fig. 3) separately averaged over the warm (April to September) and cold (October to March) half-years, we also show that aerosol loading of the mid-latitude stratosphere is maximal in the cold half-year, when the meridional air mass transport dominates (especially during the westerly phase of the QBO), and it is minimal in the warm half-year, when the zonal transport dominates. All error bars in Figs. 2 and 3 represent the standard (1–σ) deviation. As a whole, we considerably rewrote Sect 3.1. See, please, the colored version of our revised manuscript for details.

**3.2.1 Okmok and Kasatochi**

**Comment:** In Figure 5 there is a peak on September 4, 2000 at 27-30 km. Is that real?

**Response:** Yes, you are right. There are two small peaks of the scattering ratio profile (4 September 2000) at altitudes of 27 to 30 km in Figure 5. However, we believe that these peaks are not related to any aerosol events in the stratosphere and, therefore, they represent measurement uncertainties due to the low signal-to-noise ratio.

**Comment:** The altitude axis might be clearer with 1 km tic marks instead of 1.5 km.

**Response:** We agree. Done.

**4 Discussion and conclusion**

**Comment:** "The Happy Camp Complex fire consumed more than 134 acres (~543 km2)...". Acres are much smaller than km2, something is wrong with the areas.

**Response:** Thank you. We corrected the mistake: 134 acres ≈ 0.543 km$^2$. However, we removed the information about the Happy Camp fire from the text of our manuscript.

**Comment:** Figures 6 (a), 6 (b), 9, and 11. I am amazed that all four trajectories almost exactly cross over the volcanoes after so many days. Did you find an optimum altitude or something that gave you the best trajectory? Were the trajectories sensitive to the initial conditions?

**Response:** We usually calculate a set of the HYSPLIT backward trajectories which start from different aerosol layers detected in the stratosphere over Tomsk after a volcano eruption (See, please, Fig. 11 in our revised manuscript). A signal integration period of 20–30 min yields uncertainties in the start time and altitude (i.e. in initial conditions) of a backward trajectory. Therefore, we usually present the best trajectory to illustrate a possible way of how aerosol from an erupted volcanic plume could pass and be detected over Tomsk.

Sincerely,

Authors

---

## Author Comment (AC2) · 20 Jan 2017

**Manuscript Number:** acp-2016-792
**Manuscript Type:** Research article
**Title:** 30-year lidar observations of the stratospheric aerosol layer state over Tomsk (Western Siberia, Russia)

**Point-by-point response to Dr. Fromm**

**General comments**

**Comment:** This manuscript gives a wide-ranging analysis of the aerosol lidar data collected at a western Siberia location. The long temporal span of this data set, for a location far removed from other ground-based lidars, is a welcome addition to the globe's sparse stratospheric aerosol archive. Considering that these data are presumably still being collected adds value for continued monitoring of the upper troposphere and lower stratosphere. Consequently this is a potentially critical work that may offer significant value to our understanding. The authors' approach is similar to many prior works that examine a particular lidar's data set in the context of other data sets and meteorological analyses. They present some individual lidar profiles, long-term time series, and annual cycle analyses. This is a very appropriate study for Atmospheric Chemistry and Physics.

**Our response:** We thank Dr. Fromm for his great interest in our work and comprehensive criticism, comments, questions, and suggestions which will definitely allow us to improve our manuscript.

**Comment:** Along with these assets, the paper has several major weaknesses. My assessment is that the weaknesses substantially limit the value of the manuscript in its present form. For the paper to merit publication, attention to these major concerns must be paid. One major concern is a significant absence of recognition of work that has direct bearing on the analyses and conclusions presented herein. The effect goes beyond that of showing insufficient background work. It extends to the realization that critical scientific aspects of stratospheric aerosol and cloud have not been taken into account in the interpretation of the Tomsk lidar data. Detailed examples are given below. Another major concern is that the Tomsk data may be biased with respect to other similar lidar data sets, as called out specifically in a comment below.

**Response:** We have tried to take into account all weaknesses specified by Dr. Fromm.

**Comment:** The manuscript offers no direct or even indirect comparisons of the Tomsk lidar data with any other aerosol data sets; hence it is not possible to assess the accuracy of the aerosol data in the full range of stratospheric aerosol loading. Another concern is that the authors utilize a local measurement data set of total ozone abundance, but without any qualification. The reader is left with great uncertainty as to the robustness of the results and certain interpretations the authors give for certain phenomena.

**Response:** Such data comparisons exist and were made with, e.g., data of the Minsk lidar station (Belarus) which is simultaneously incorporated in 1) the CIS-LiNet together with the Tomsk lidar station (Chaykovskii et al., 2005; Zuev et al., 2009); and 2) the EARLINET (Wandinger et al., 2016). Moreover, the Tomsk station measurement data were widely used, e.g., by Ridley et al. (2014).

Chaykovskii, A. P., Ivanov, A. P., Balin, Yu. S., El'nikov, A. V., Tulinov, G. F., Plusnin, I. I., Bukin, O. A., and Chen, B. B.: CIS-LiNet lidar network for monitoring aerosol and ozone: methodology and instrumentation, Atmos. Ocean. Opt., 18, 958–964, 2005.

Zuev, V. V., Balin, Yu. S., Bukin, O. A., Burlakov, V. D., Dolgii, S. I., Kabashnikov, V. P., Nevzorov, A. V., Osipenko, F. P., Pavlov, A. N., Penner, I. E., Samoilova, S. V., Stolyarchuk, S. Yu., Chaikovskii, A. P., and Shmirko, K. A.: Results of joint observations of aerosol perturbations of the stratosphere at the CIS-LiNet network in 2008, Atmos. Ocean. Opt., 22, 295–301, 2009.

Wandinger, U., Freudenthaler, V., Baars, H., Amodeo, A., Engelmann, R., Mattis, I., Groß, S., Pappalardo, G., Giunta, A., D'Amico, G., Chaikovsky, A., Osipenko, F., Slesar, A., Nicolae, D., Belegante, L., Talianu, C., Serikov, I., Linné, H., Jansen, F., Apituley, A., Wilson, K. M., de Graaf, M., Trickl, T., Giehl, H., Adam, M., Comerón, A., Muñoz-Porcar, C., Rocadenbosch, F., Sicard, M., Tomás, S., Lange, D., Kumar, D., Pujadas, M., Molero, F., Fernández, A. J., Alados-Arboledas, L., Bravo-Aranda, J. A., Navas-Guzmán, F., Guerrero-Rascado, J. L., Granados-Muñoz, M. J., Preißler, J., Wagner, F., Gausa, M., Grigorov, I., Stoyanov, D., Iarlori, M., Rizi, V., Spinelli, N., Boselli, A., Wang, X., Lo Feudo, T., Perrone, M. R., De Tomasi, F., and Burlizzi, P.: EARLINET instrument intercomparison campaigns: overview on strategy and results, Atmos. Meas. Tech., 9, 1001-1023, doi:10.5194/amt-9-1001-2016, 2016.

Ridley, D. A., et al. (2014), Total volcanic stratospheric aerosol optical depths and implications for global climate change, Geophys. Res. Lett., 41, 7763–7769, doi:10.1002/2014GL061541.

**Specific comments**

**Comment:** Somewhere in this paper the recent review paper by Kremser et al. should be mentioned and cited.

Kremser, S., et al. (2016), Stratospheric aerosol – Observations, processes, and impact on climate, Rev. Geophys., 54, doi:10.1002/2015RG000511.

**Response:** OK. We added this reference to our manuscript.

**Instead of**
"The volcanogenic aerosol perturbs the radiation-heat balance of the atmosphere, and thus, significantly affects the atmospheric dynamics (Timmreck, 2012; Driscoll et al., 2012)."

**we wrote**
"The volcanogenic aerosol perturbs the radiation-heat balance of the atmosphere, and thus, significantly affects the atmospheric dynamics and climate (Timmreck, 2012; Driscoll et al., 2012; Kremser et al., 2016)."
[**Page 2, lines 7 and 8, revised manuscript**]

Kremser, S., Thomason, L. W., von Hobe, M., Hermann, M., Deshler, T., Timmreck, C., Toohey, M., Stenke, A., Schwarz, J. P., Weigel, R., Fueglistaler, S., Prata, F. J., Vernier, J.-P., Schlager, H., Barnes, J. E., Antuña-Marrero, J.-C., Fairlie, D., Palm, M., Mahieu, E., Notholt, J., Rex, M., Bingen, C., Vanhellemont, F., Bourassa, A., Plane, J. M. C., Klocke, D., Carn, S. A., Clarisse, L., Trickl, T., Neely, R., James, A. D., Rieger, L., Wilson, J. C., and Meland, B.: Stratospheric aerosol – Observations, processes, and impact on climate, Rev. Geophys., 54, 278–335, doi:10.1002/2015RG000511, 2016.

**Comment:** There is no citation of Vernier et al (GRL, 2011), Ridley et al. (GRL, 2014), Santer et al. (Nature Geo., 2014) and some other very relevant recent papers on volcanoes and recent stratospheric aerosol layer trends.

**Response:** Thank you for these references. However, these and other very relevant recent papers were considered in Kremser et al. (2016), which we already cited.

**Comment:** P2, L10. Consider citing

Jäger, H. and Wege, K.: Stratospheric Ozone Depletion at Northern Midlatitudes after Major Volcanic Eruptions, J. Atmos. Chem., 10, 273-287, 1990.

**Response:** Thank you for the reference, but the data and results of this paper were mentioned in Trickl et al. (2013), which we also cited in our manuscript.

**Comment:** P2, L5. "vulcanian" refers to a specific style of eruption, the characteristics of which are not associated with stratospheric injection. Is the term even needed to make the point of this sentence?

**Response:** According to the Smithsonian Institution Global Volcanism Program, gas and ash plumes from some volcanic eruptions (with VEI ≥ 3) of the Vulcanian type can directly reach the stratospheric altitudes:

https://web.archive.org/web/20111110173623/http://www.volcano.si.edu/world/eruptioncriteria.cfm

(See there, please, Section "VEI (Volcanic Explosivity Index)" and Table containing the Eruption Type and Cloud Column Height).

**Comment:** P3, L19. How about measurement frequency? What was the lidar operation frequency and regularity? It would be important for the reader to know what is typical for the number of profiles that go into the 10-day average.

**Response:** The lidar operation regularity depends on the weather conditions, of course. The absence of clouds is required for stratospheric aerosol measurements. As seen in Fig. 1, the measurement frequency allows us to retrieve the 10-day average values of the integrated stratospheric backscatter coefficient, when it is possible. See, please, also the "10-day average values" file in **the Supplement** containing data for Fig. 1.

**Comment:** P3, L23. Consider giving the name of the PMT manufacturer.

**Response:** The manufacturer of PMTs "FEU-130" is the Moscow Elecro-Lamp Plant (MELZ).

http://www.melz-evp.ru/about.html

http://rutubes.com/category/melz-tube-moscow-russia/

**Instead of**
"The signals were registered with a vertical resolution of 374 m by a photomultiplier tube (PMT) FEU-130 operating in the photon counting mode."

**we wrote**
"The signals were registered with a vertical resolution of 374 m by a photomultiplier tube (PMT) FEU-130 (USSR, Moscow Elecro-Lamp Plant) operating in the photon counting mode."
[**Page 3, lines 25 and 26, revised manuscript**]

**Comment:** P4, L4. This statement arouses my curiosity to know if this lidar made any observations of the Chelyabinsk bolide plume in 2013. It was observed by satellite aerosol profilers above 30 km.

Gorkavyi, N., D. F. Rault, P. A. Newman, A. M. da Silva, and A. E. Dudorov (2013), New stratospheric dust belt due to the Chelyabinsk bolide, Geophys. Res. Lett., 40, 4728–4733, doi:10.1002/grl.50788.

**Response:** We have rechecked our lidar measurement data (both scattering ratios and integrated stratospheric backscatter coefficients). The Chelyabinsk bolide plume was not detected over Tomsk in 2013 (See, please, red lines from 977 to 992 in the "10-day average values" file in **the Supplement** to check the integrated stratospheric backscatter coefficient data). So, the belt was not over Tomsk.

**Comment:** P4, L10. What does the pi character refer to? Please consider stating its meaning.

**Response:** The $\pi$ character denotes an angle of $\pi$ radian (180 degrees), i.e. the angle of the backscatter lidar signal propagation.

**We have added the following sentence to the revised manuscript:**
"$\pi$ denotes an angle of $\pi$ radian, i.e. the angle of the backscatter lidar signal propagation."
[**Page 4, lines 14 and 15, revised manuscript**]

**Comment:** P4, L20. This is understandable and defensible. But the median tropopause at Tomsk is such that a lot of the lowermost stratosphere is not sampled with the fixed lower limit of 15 km. For instance, many volcanic plumes just above the tropopause have occurred in recent years (Kasatochi is one example). And pyrocumulonimbus smoke plumes are routine in summer, but usually between the tropopause and ~15 km. It might be worthwhile to try other lower-boundary, for instance based on potential temperature or a tropopause-relative height offset.

**Response:** Yes, some stratospheric aerosol events could be missed due to the lower limit of 15 km for integrated stratospheric backscatter coefficient $B_\pi^a$ calculation. However, this limit definitely excludes any tropospheric aerosol events. Moreover, we usually determined the scattering ratio $R(H)$ starting from the lower limit of 12.5 km (Figs. 5, 7, 8, 12–15), or even from 10 km in the case of Eyjafjallajökull volcano (Fig. 10).

**Comment:** Figure 2. The error/uncertainty bars are not defined or even mentioned. Moreover, however they are defined, the majority of them extend farther than the winter/summer range. This suggests to me that this pattern of monthly averages is not particularly repeatable. Auth should consider discussion of the robustness of this pattern in light of the relatively large uncertainty. The feature that stands out here is the March average and uncertainty. Can the authors explain why this month's aerosol amount is so large and variable?

**Response:** Sect 3.1 was considerably rewritten. See, please, the colored version of our revised manuscript for details. For example, we have extended the analyzed period of the background aerosol loading variations over Tomsk up to 16 years (1999–2015) and averaged the $B_\pi^a$ values separately for the westerly and easterly phases of the quasi-biennial oscillations (QBO) characterized by zonal winds in the equatorial region at 30 mbar (Fig. 3, ex-Fig. 2). The monthly average $B_\pi^a$ data for March–June 2000 (after the Hekla eruption), August–November 2008 (after the Okmok and Kasatochi eruptions), August–October 2009 (after the Sarychev Peak eruption), and also April and August–October 2011 (after the Merapi, Grimsvötn, and Nabro eruptions) were not taken into account. The exclusion of these perturbed data allowed us to extend the analyzed period of the background aerosol loading variations and, therefore, to improve the statistical reliability of the $B_\pi^a$ data series. All error bars in Figs. 2 and 3 represent the standard (1–$\sigma$) deviation.

**Comment:** P6, L4. The minimum integrated backscatter coefficient value in Figure 1 and Trickl et al.'s background value are not equal, in contrast to the authors' claim. Figure 1's value is between 1e-4 and 2e-4. Trickl et al.'s 1979 background value is 5e-5, smaller by roughly a factor of two. Zuev et al.'s Pinatubo peak matches well with Trickl et al., but the comparable quiescent-period values are off by about a factor of 2. This sentence should be corrected. The differences in the background values should be acknowledged and discussed. Trickl integrated from a tropopause-relative altitude that probably includes more of the lowermost stratosphere than Zuev et al. yet their integral is much smaller. The wavelength difference is seemingly too small to account for the difference.

**Response:** Thank you for this comment. We agree that the background value of $B_\pi^a$ observed in Tomsk at $\lambda$ = 532 nm in 2004 was not equal to the Trickl et al.'s background value determined for Garmisch-Partenkirchen at $\lambda$ = 694 nm in 1979. We corrected this sentence. However, it is not reasonable to discuss and hardly possible to intercompare the 2004 Tomsk and 1979 Garmisch-Partenkirchen $B_\pi^a$ background values, because one atmospheric (air) circulation epoch was replaced by another one during 25 years (e.g., Chu, 2002; Chernavskaya et al., 2006). Therefore, the sources and type of the background aerosol were also changed.

**Instead of**

"The thin horizontal line in Fig. 1 indicates the minimum value of the annual average $B_\pi^a$ reached in 2004. This value equals to that determined in 1979 and considered as the background one (Trickl et al., 2013)."

**we wrote**

"Note that taking into account the spectral dependence of $B_\pi^a$, its minimum annual average value observed in Tomsk at $\lambda$ = 532 nm in 2004 was close to that determined for Garmisch-Partenkirchen at $\lambda$ = 694 nm in 1979 and considered as the background by Trickl et al. (2013)."
[**Page 7, line 20 and Page 8, lines 1 and 2, revised manuscript**]

Chu, P.-S.: Large-Scale Circulation Features Associated with Decadal Variations of Tropical Cyclone Activity over the Central North Pacific, Journal of Climate, 15, 2678–2689, doi:10.1175/1520-0442(2002)015<2678:LSCFAW>2.0.CO;2, 2002.

Chernavskaya, M. M., Kononova, N. K., and Val'Chuk, T. E.: Correlation between atmospheric circulation processes over the Northern Hemisphere and parameter of solar variability during 1899–2003, Adv. Space Res., 37, 1640–1645, doi:10.1016/j.asr.2005.06.022, 2006.

Trickl, T., Giehl, H., Jäger, H., and Vogelmann, H.: 35 yr of stratospheric aerosol measurements at Garmisch-Partenkirchen: from Fuego to Eyjafjallajökull, and beyond, Atmos. Chem. Phys., 13, 5205–5225, doi:10.5194/acp-13-5205-2013, 2013.

**Comment:** P7, L14-19. Discussion of the Brewer Dobson circulation and its impact on the extratropical SAL. Presumably the point here is that the pattern shown by Fig. 2 is consistent with the generalizations here such that a citation is needed for work(s) that show this intra-annual tendency.

**Response:** There is no need for a reference to any other work, because we can show the intra-annual tendency (an annual cycle in the mid-latitude SAL via the Brewer-Dobson circulation) based on our 16-year averaged values of the monthly average $B_\pi^a$ (Fig. 3, ex-Fig. 2). See, please, the rewritten Sect. 3.1 for details.

**Comment:** Figure 3. What is the averaging period of the data points?

**Response:** (Figure 3 is now Fig. 2). The averaging period for each **red** point (warm half-year) is from April to September of the corresponding year. The averaging period for each **blue** point (cold half-year) is from October of the current year to March of the next year. The "warm" and "cold" average points are assigned to 1 June of the current year and 1 January of the next year, respectively. Black and red vertical bars at the bottom of the figure indicate volcanic eruptions as in Fig. 1 (see also Table 1).

**Comment:** Since error bars were shown for Figure 2, they should also be shown here, and the implications discussed.

**Response:** Done. All error bars represent the standard deviation.

**Comment:** P8, L17-18. This statement does not reconcile with VEI=4. I recognize that VEI=4 is in the GVP database. So this sentence probably should call out the apparent discrepancy.

**Response:** Well, both VEI and the maximum plume altitude of the Eyjafjallajökull volcano eruption were taken from the GVP database, as it is noted in Table 1 caption. The VEI coefficient is responsible for the volcanic gas and ash volume, rather than for their rise altitude.

**Comment:** P10, L3. The trajectory analysis is largely unnecessary. There is no need to run trajectories for the 8 Aug observation; it's obvious that layer can't be from Kasatochi (and there is no other plausible source). For the September observation (and latter ones), the stratosphere at that time had Okmok and Kasatochi aerosols distributed all over. If the Tomsk back trajectory intersected with any of the Okmok layers prior to its pass over the Kasatochi volcano, that is a more plausible attribution than a 3.5 week trajectory connection.

**Response:** Well, we agree that the sentence "In the August of 2008, the detected aerosol layers were related only to the Okmok plume, but in the September of 2008, there was observed a superposition of plumes from both volcanoes." declares the obvious fact. The mentioned sentence was removed from the text. Regarding the usage of the trajectory analysis, we can say that all HYSPLIT trajectories were used only to illustrate and not to prove the results of our lidar observation. We noted in several places in our manuscript (including Sect. "Discussion and conclusion") that all the trajectories longer than 2 weeks can be considered only as probable ones. We also compared our vertical profiles of the scattering ratio $R(H)$ with the Minsk CIS-LiNet lidar station profiles (Zuev et al., 2009) and revealed the similar SAL perturbations over Minsk. As a whole, we considerably rewrote Sect 3.2.1. See, please, the colored revised version of our manuscript for details.

Zuev, V. V., Balin, Yu. S., Bukin, O. A., Burlakov, V. D., Dolgii, S. I., Kabashnikov, V. P., Nevzorov, A. V., Osipenko, F. P., Pavlov, A. N., Penner, I. E., Samoilova, S. V., Stolyarchuk, S. Yu., Chaikovskii, A. P., and Shmirko, K. A.: Results of joint observations of aerosol perturbations of the stratosphere at the CIS-LiNet network in 2008, Atmos. Ocean. Opt., 22, 295–301, 2009.

**Comment:** P10. The impression I got from reading this analysis was that Zuev et al. are attempting to draw general conclusions about the Kasatochi and Okmok plumes and the SAL. If they are instead limiting their assessment of the SAL to just where the Tomsk observations are, they should make that explicit. The more general SAL analysis of these plumes is already published but not cited in this manuscript. E.g. Bourassa et al., Anderssen et al., Kravitz et al. These make clear that the Kasatochi and Okmok plumes were observed at all altitudes from the tropopause to nearly 19 km.

Andersson et al. (2015), Significant radiative impact of volcanic aerosol in the lowermost stratosphere, DOI: 10.1038/ncomms8692.

Bourassa, A., D. Degenstein, B. Elash, and E. Llewellyn (2010), Evolution of the stratospheric aerosol enhancement following the eruptions of Okmok and Kasatochi: Odin-OSIRIS measurements, J. Geophys. Res., 115, D00L03, doi:10.1029/2009JD013274.

Kravitz, B., A. Robock, and A. Bourassa (2010), Negligible climatic effects from the 2008 Okmok and Kasatochi volcanic eruptions, J. Geophys. Res., 115, D00L05, doi:10.1029/2009JD013525.

**Response:** As we already pointed out above, we deeply rewrote Sect 3.2.1. to clarify the situation. First, our conclusion (based on the HYSPLIT trajectories), that the plumes from both Okmok and Kasatochi volcanoes reached altitudes of ≥16 km, is consistent with different satellite observation data, including the CALIOP data (Yang et al., 2010; Kristiansen et al., 2010; Prata et al., 2010). Second, the more general SAL analysis of these plumes was already published by Zuev et al. (2009) but not cited in the works published later by, e.g., Bourassa et al. (2010), Kravitz et al. (2010), and Anderssen et al. (2015). Nevertheless, we included Bourassa et al. (2010) and Andersson et al. (2015) to our manuscript as an example. See, please, the colored revised version of our manuscript for details.

Yang, K., Liu, X., Bhartia, P. K., Krotkov, N. A., Carn, S. A., Hughes, E. J., Krueger, A. J., Spurr, R. J. D., and Trahan, S. G.: Direct retrieval of sulfur dioxide amount and altitude from spaceborne hyperspectral UV measurements: Theory and application, J. Geophys. Res., 115, D00L09, doi:10.1029/2010JD013982, 2010.

Kristiansen, N. I., Stohl, A., Prata, A. J., Richter, A., Eckhardt, S., Seibert, P., Hoffmann, A., Ritter, C., Bitar, L., Duck, T. J., and Stebel, K.: Remote sensing and inverse transport modeling of the Kasatochi eruption sulfur dioxide cloud, J. Geophys. Res., 115, D00L16, doi:10.1029/2009JD013286, 2010.

Prata, A. J., Gangale, G., Clarisse, L., and Karagulian, F.: Ash and sulfur dioxide in the 2008 eruptions of Okmok and Kasatochi: Insights from high spectral resolution satellite measurements, J. Geophys. Res., 115, D00L18, doi:10.1029/2009JD013556, 2010.

**Comment:** P10, L12. My impression is that Zuev et al. are attempting to assess the accuracy of the injection height as reported by the GVP. The point made here, and below, regarding the accuracy of the Hmpa for the two eruptions, is of little consequence. I also get the impression that they are using the altitude of the back trajectory endpoint over the volcano as a point of comparison with Hmpa. If I have the wrong impression, perhaps other readers will be similarly affected. So I ask the authors to clarify the wording here. Otherwise, giving the precise altitude of a 3-4 week old trajectory much weight asks more of the trajectory model and the weather analyses than they can promise. Secondly, the Hmpa data in the GVP database is neither tightly constrained. Thirdly, it can be shown easily with CALIOP data that the Hmpa for these two eruptions is even farther off the mark than this analysis suggests. CALIOP data within a few days of each eruption shows that the injection altitude was at least 17 km (Okmok) and 18 km (Kasatochi). These data would offer a much more compelling analysis for this argument than the Tomsk data.

**Response:** We have rested on our experimental results obtained with the SLS lidar, not on the CALIOP data. We calculated a lot of trajectories and showed only several ones as an example. Note that our HYSPLIT trajectories in cases of Okmok and Kasatochi eruptions (started from Tomsk) definitely ended within their plumes over the volcanoes. Thus, these results of the trajectory analysis made are also in agreement with the CALIOP data (see, please, our response for your previous comment). We never claimed that HYSPLIT trajectories are able to determine the maximum plume altitudes of any volcanic eruptions. No essential conclusions were made on the basis of the HYSPLIT trajectories.

**Comment:** P11, L15. This is way too precise and exclusive. The Sarychev Peak eruption spanned several days, as shown in their table. The time of a 3+ week back trajectory is by its nature too uncertain to permit such a definitive connection.

**Response:** The trajectory (presented in Fig. 9 and started from one of the detected aerosol layers over Tomsk) shows a possible way how aerosol from the Sarychev Peak volcanic plume erupted in one of the eruption days (e.g. 15 June) could pass and be detected over Tomsk on 7 July. We consider that the trajectory also shows that the detected aerosol layer is not associated with the after-effect of the 2009 Redoubt eruption.

**Instead of**
"As seen in Fig. 9, this layer is associated with the backward trajectory passed over Sarychev Peak volcano at an altitude of ~13.8 km on 15 June at the moment of the eruption, 17:30 UTC."

**we wrote**
"This layer is seen in Fig. 9 to be: 1) associated with the backward trajectory passed over Sarychev Peak volcano at an altitude of ~13.8 km on 15 June at the moment of the eruption, 17:30 UTC; and 2) not associated with the after-effect of the Redoubt eruption."
[**Page 13, lines 13–16, revised manuscript**]

**Comment:** Figure 10. These strong stratospheric layers at 15 km and their source are not subjected to a back trajectory test. Given that they are a much shorter time post eruption than the prior examples, it would critical to show if they connect to the volcano. If Eyja did not inject material this far above the tropopause, where did this material come from?

**Response:** Now we present in Fig. 11 an ensemble of the air-mass backward trajectories started from altitudes (~11.1–14.6 km a.s.l.) of strong stratospheric layers over Tomsk (Fig. 10) on 21 April 2010 at 00:00 LT (20 April, 17:00 UTC). Only one trajectory started from an altitude of ~11.6 km a.s.l. directly passed over Eyjafjallajökull volcano. The other trajectories passed south of the volcano.

**Instead of**
"Figure 11 shows the HYSPLIT air-mass backward trajectory started from an altitude of the detected aerosol layers (~11.6 km a.s.l.) over Tomsk on 21 April at 00:00 LT (20 April, 17:00 UTC). The trajectory passed over Eyjafjallajökull volcano on one of the eruption days, 16 April at 13:00 UTC, at the altitude $H_{\text{traj.}}^{\text{back.}} \approx 10.7$ km that is clearly higher than $H_{\text{MPA}} \leq 9$ km. This inconsistency between the altitudes $H_{\text{traj.}}^{\text{back.}}$ and $H_{\text{MPA}}$ ($H_{\text{traj.}}^{\text{back.}}$ should normally be equal to or lower than $H_{\text{MPA}}$) is discussed in Sect. 4."

**we wrote**
"Figure 11 shows the HYSPLIT air-mass backward ensemble trajectories started from altitudes of the detected aerosol layers (~11.1–14.6 km a.s.l.) over Tomsk on 21 April at 00:00 LT (20 April, 17:00 UTC). Only one trajectory (started from an altitude of ~11.6 km) directly passed over Eyjafjallajökull volcano on one of the eruption days, 16 April at 13:00 UTC, at the altitude $H_{\text{traj.}}^{\text{back.}} \approx 10.7$ km that is clearly higher than $H_{\text{MPA}} \leq 9$ km. The inconsistency between the HYSPLIT $H_{\text{traj.}}^{\text{back.}}$ and GVP $H_{\text{MPA}}$ altitudes is discussed in Sect. 4. The other trajectories passed south of the volcano."
[**Page 15, lines 9–13, revised manuscript**]

**Comment:** P15, L13. Regarding the connection between the weak layer in Figure 12, localized reduced total ozone, and Merapi, a stronger candidate would be Arctic O3 depletion. See Manney et al., (Nature, 2011) "Unprecedented Arctic Ozone Loss in 2011" doi:10.1038/nature10556.

**Response:** We agree with Dr. Fromm that the degree of ozone depletion in April 2011 was greater than it could be due to the influence on ozone of volcanic aerosol plumes. As the purpose of our research was to study aerosol perturbations of the stratosphere over Tomsk and was not to study the ozone depletion, here and elsewhere all information and discussions about ozone depletion and mini-holes were removed from the manuscript, without prejudice to the

generality of the foregoing. We substituted Fig. 12 by a new one with additional perturbed scattering ratio profiles to make it clearer and informative.

**Instead of**
"As an example, Fig. 12 presents an aerosol layer observed over Tomsk at an altitude of ~18 km on 18 April 2011."

**we wrote**
"Figure 12 presents the observed after-effect of the Merapi eruption, i.e. several perturbed scattering ratio profiles retrieved from the SLS aerosol lidar measurements between 28 February and 18 April 2011."
[**Page 16, lines 11 and 12, revised manuscript**]

**Instead of**
"**Figure 12.** Detection of the Merapi volcanic plume in the stratosphere over Tomsk. The volcano erupted in Indonesia from 4 to 5 November 2010."

**we wrote**
"**Figure 12.** Perturbed scattering ratio profiles retrieved from the SLS aerosol lidar measurements in the winter-spring period of 2011."
[**Page 17, line 2, revised manuscript**]

The sentence "Note also that a significant decrease in the total ozone content was observed over Tomsk at the same period of time (April 2011), which is evidence of stratospheric ozone depletion caused by the Merapi aerosol plume (Zuev et al., 2016)." was removed from our manuscript.
[**Page 16, line 15, revised manuscript**]

The reference Zuev et al., 2016 was removed from our manuscript too.

Zuev, V. V., Zueva, N. E., Savelieva, E. S., Bazhenov, O. E., and Nevzorov, A. V.: On the role of the eruption of the Merapi volcano in an anomalous total ozone decrease over Tomsk in April 2011, Atmos. Ocean. Opt., 29, 298–303, 2016.

**Comment:** P16, L4. This conclusion regarding Nabro's injection height has been exhaustively disputed. There are no citations here of the papers demonstrating the classic, direct injection of Nabro to the stratosphere. http://science.sciencemag.org/content/339/6120/647.4

http://science.sciencemag.org/content/339/6120/647.3

The Bourassa et al. scenario was shown to be improbable by Fairlie et al.(2014). Clarisse et al. (2014), Fairlie et al. (2014), and Penning de Vries et al. (2014) showed, using a variety of satellite data, that direct injection above the tropopause was more consistent with these data than the indirect path via the Monsoon. Fromm et al. (2014) established a root cause for the misattribution to the Asian Monsoon pathway and made connections with prior papers on other volcanic stratospheric aerosol discrepancies. Hence the full weight of Nabro-related papers gives a very different perspective than what is documented here. I would ask the authors to more fully capture these various works in their presentation on Nabro.

http://www.atmos-chem-phys.net/14/7045/2014/

http://www.atmos-chem-phys.net/14/3095/2014/

http://www.atmos-chem-phys.net/14/8149/2014/

http://onlinelibrary.wiley.com/doi/10.1002/2014JD021507/full.

**Response:** We have to agree with you. Thank you for these references. We used them all.

We revised and partially corrected Sect. 3.3.2.

**Instead of**
"Grimsvötn volcano erupted ash clouds and gases directly into the stratosphere at an altitude of 20 km, whereas the Nabro volcanic plume did not exceed the local tropopause altitude. Nevertheless, Bourassa et al. (2012) and Robock (2015) showed that a considerable part of the Nabro volcanic aerosol and gases, erupted into the upper troposphere, was able to enter the mid-latitude stratosphere due to deep convection and vertical air transport associated with the strong Asian summer monsoon anticyclone."

**we wrote**
"According to the GVP data, Grimsvötn volcano erupted ash clouds and gases directly into the stratosphere at an altitude of 20 km, whereas the Nabro volcanic plume did not exceed the local tropopause altitude. Bourassa et al. (2012) showed that a considerable part of the Nabro volcanic aerosol and gases, erupted into the upper troposphere, was able to

enter the mid-latitude stratosphere due to deep convection and vertical air transport associated with the strong Asian summer monsoon anticyclone. On the other hand, Vernier et al. (2013), Fromm et al. (2013), Fairlie et al. (2014), Clarisse et al. (2014), and Penning de Vries et al. (2014) showed that the initial Nabro plume was directly injected into the lower stratosphere at altitudes up to 18 km (Fromm eat al., 2014)."
[**Page 17, lines 5–11, revised manuscript**]

Vernier, J.-P., Thomason, L. W., Fairlie, T. D., Minnis, P., Palikonda, R., and Bedka, K. M.: Comment on "Large volcanic aerosol load in the stratosphere linked to Asian monsoon transport", Science, 339, 647-d, doi:10.1126/science.1227817, 2013.

Fromm, M., Nedoluha, G., and Charvát, Z.: Comment on "Large volcanic aerosol load in the stratosphere linked to Asian monsoon transport", Science, 339, 647-c, doi:10.1126/science.1228605, 2013.

Fairlie, T. D., Vernier, J.-P., Natarajan, M., and Bedka, K. M.: Dispersion of the Nabro volcanic plume and its relation to the Asian summer monsoon, Atmos. Chem. Phys., 14, 7045–7057, doi:10.5194/acp-14-7045-2014, 2014.

Clarisse, L., Coheur, P.-F., Theys, N., Hurtmans, D., and Clerbaux, C.: The 2011 Nabro eruption, a SO2 plume height analysis using IASI measurements, Atmos. Chem. Phys., 14, 3095-3111, doi:10.5194/acp-14-3095-2014, 2014.

Penning de Vries, M. J. M., Dörner, S., Puķīte, J., Hörmann, C., Fromm, M. D., and Wagner, T.: Characterisation of a stratospheric sulfate plume from the Nabro volcano using a combination of passive satellite measurements in nadir and limb geometry, Atmos. Chem. Phys., 14, 8149-8163, doi:10.5194/acp-14-8149-2014, 2014.

Fromm, M., Kablick III, G., Nedoluha, G., Carboni, E., Grainger, R., Campbell, J., and Lewis, J.: Correcting the record of volcanic stratospheric aerosol impact: Nabro and Sarychev Peak, J. Geophys. Res., 119, 10343–10364, doi:10.1002/2014JD021507, 2014.

**Comment:** P16, L18. Often? I'm not aware of any literature showing any direct positive correlation between PSC formation and volcanic sulfur loading. Hence a citation is needed. Fromm et al. (2003) actually showed little (or even negative) correlation between PSC frequency and ambient aerosol loading. Hence it would be important for the authors to substantiate the claim they make here.

Fromm, M., J. Alfred, and M. Pitts, A unified, long-term, high-latitude stratospheric aerosol and cloud database using SAM II, SAGE II, and POAM II/III data: Algorithm description, database definition, and climatology, J. Geophys. Res., 108(D12), 4366, doi:10.1029/2002JD002772, 2003.

**Response:** The PSC formation depends on both low temperature (lower than –78 °C) and the presence of condensation nuclei. The volcanogenic aerosol also plays the role of condensation nuclei. The PSC formation is a local process and could not work on the global scale. Concerning the direct positive correlation between PSC formation and volcanic sulfur loading, see, e.g., the following reference:

Rose, W. I., Millard, G. A., Mather, T. A., Hunton, D. E., Anderson, B., Oppenheimer, C., Thornton, B. F., Gerlach, T. M., Viggiano, A. A., Kondo, Y., Miller, T. M., and Ballenthin, J. O.: Atmospheric chemistry of a 33–34 hour old volcanic cloud from Hekla Volcano (Iceland): Insights from direct sampling and the application of chemical box modeling, J. Geophys. Res., 111, D20206, doi:10.1029/2005JD006872, 2006.

We rewrote Sect. 3.4 and included Fromm et al., (2003) with appropriate comments to our manuscript.

**Instead of**
"Therefore, the injections of volcanogenic $H_2SO_4$ aerosols or/and $SO_2$ into the stratosphere often lead to PSC formation, if the air temperature $< -78$ °C."

**we wrote**
"Therefore, injections of volcanogenic $H_2SO_4$ aerosols or/and $SO_2$ into the stratosphere can lead to PSC formation, if the air temperature $< -78$ °C. The direct positive correlation between PSC formation and volcanogenic nitric and sulfur acid aerosols loading was shown, e.g., by Rose et al. (2006). However, it should be noted that, in contrast to Rose et al. (2006), Fromm et al. (2003) showed little (or even negative) correlation between PSC events and ambient aerosol loading."
[**Page 18, lines 8–11, revised manuscript**]

**Comment:** P17, L6. Not necessarily. In fact, spatial correlations between total-O3 minima (socalled ozone mini-holes) and PSCs are routinely attributable to synoptic-scale dynamics. See

Hood et al. and Teitelbaum et al. Exploring polar stratospheric cloud and ozone minihole formation: The primary importance of synoptic scale flow perturbations, Teitelbaum, H., M. Fromm, & M. Moustaoui, J. Geophys. Res., 106, 28173–28188, 2001.

Hood, L. L., B.E. Soukharev, M. Fromm, and J.P. McCormack, Origin of extreme ozone minima at middle to high northern latitudes, J. Geophys. Res., 106, 20925–20940, 2001.

**Response:** All information and discussions about ozone depletion were removed from Sect. 3.4. See, please, the colored version of our revised manuscript for details.

**Instead of**

"Hence, the stratospheric temperature over Tomsk can occasionally be cooled lower than –78 °C, when Tomsk is inside the polar vortex.  Thus, the detection of aerosol layers in the stratosphere at extremely low temperatures  can be indicative of the presence of PSCs."

**we wrote**

"Hence, the stratospheric temperature over Tomsk can occasionally be cooled lower than –78 °C, when Tomsk is inside the polar vortex. Thus, the detection of aerosol layers in the stratosphere at extremely low temperatures can be indicative of the presence of PSCs."
**[Page 18, lines 14–16, revised manuscript]**

The reference Solomon (1999) was removed from our manuscript.

Solomon, S.: Stratospheric ozone depletion: A review of concepts and history, Rev. Geophys., 37, 275–316, doi:10.1029/1999RG900008, 1999.

**Instead of**

"The first lidar PSC observations (over Tomsk) , were made at $\lambda$ = 1064 nm on January 1995 (Zuev and Smirnov, 1997)."

**we wrote**

"The first lidar PSC observations over Tomsk were made at $\lambda$ = 1064 nm in January 1995 (Zuev and Smirnov, 1997)."
**[Page 19, line 7, revised manuscript]**

**Comment:** P17, L17. This PSC is consistent with the findings of Fromm et al. (1999) who showed that the cold pool and PSC frequency in the northern winter of 1994/95 was located near the Tomsk longitude.

http://onlinelibrary.wiley.com/doi/10.1029/1999JD900273/epdf.

**Response:** Thank you for this reference. We added Fromm et al. (1999) with appropriate comments to our manuscript.

**Instead of**

"The stratospheric temperature was lower than –80 °C ."

**we wrote**

"The stratospheric temperature was lower than –80 °C. The cold pool presence and PSC events near the Tomsk longitude during the northern winter of 1994/95 were also reported by Fromm et al. (1999). The formation of these dense PSCs was caused by high concentrations of residual post-Pinatubo aerosols."
**[Page 19, lines 9–12, revised manuscript]**

Fromm, M. D., Bevilacqua, R. M., Hornstein, J., Shettle, E., Hoppel, K., and Lumpe, J. D.: An analysis of Polar Ozone and Aerosol Measurement (POAM) II Arctic polar stratospheric cloud observations, 1993–1996, J. Geophys. Res., 104, 24341–24357, doi:10.1029/1999JD900273, 1999.

**Comment:** P17, L19. Please see the prior comment regarding mini-holes.

**Response:** See, please, our **Response** to **Comment:** P17, L6.

**Comment:** P17, L20. This seems to be quite unlikely considering that PSCs are formed inside the vortex, which represents air isolated from the extratropics. Rabaul aerosols, introduced just a few months before the northern 1994/95 vortex season, would not likely have been meridionally transported that far north in time to be in place before the vortex formed and isolation was in place.

**Response:** Well, aerosol resulted from the 1994 Rabaul eruption could enter inside the polar vortex weakened and deformed due to a minor sudden stratospheric warming in January 1995. This is a subject for discussion and additional research. Anyway, as the main cause of PSC formation is the extremely low temperatures, we excluded the Rabaul volcanic aerosol discussion from this part of the text.

**Comment:** P18, L2. It seems highly unlikely (or at least hard to prove) to argue for a PSC in the northern polar vortex one season after a southern hemisphere volcanic eruption. It would be best to state this as speculative, if the statement is to remain.

**Response:** We agree with this comment.

**Instead of**
"Another event of PSCs over Tomsk was observed at λ = 532 nm on 27 January 2007 (Fig. 14) ."

**we wrote**
"Another event of PSCs over Tomsk was observed at λ = 532 nm on 27 January 2007 (Fig. 14)."
[**Page 19, line 13, revised manuscript**]

**Comment:** P18, L6. See the prior comment about mini-holes. At this time in Jan 2007, a localized o3 minimum was at Tomsk's longitude. The ozone signature was likely an artifact of local dynamics.

http://exp-studies.tor.ec.gc.ca/cgibin/selectMap?lang=e&type1=du&day1=27&month1=01&year1=2007&howmany1=1&interval1=1&intervalunit1=d&hem1=P18.

**Response:** We agree with the comment. The discussion about ozone depletion was removed from the text.

**Instead of**
". Thus, PSCs were detected at least twice (in 1995 and 2007) during 30 years of stratospheric aerosol lidar measurements in Tomsk."

**we wrote**
"High $R(H)$ values at altitudes in the range of 13 to 17 km were probably due to the winter aerosol supplying of the SAL from the stratospheric tropical aerosol reservoir enriched by the 2006 Rabaul eruption plume (Table 1, Fig. 14). Thus, PSCs were detected at least twice (in 1995 and 2007) during 30 years of stratospheric aerosol lidar measurements in Tomsk."
[**Page 19, lines 17–20, revised manuscript**]

The reference Zuev et al. (2008) was removed from our manuscript.

Zuev, V. V., Bazhenov, O. E., Burlakov, V. D., Grishaev, M. V., Dolgii, S. I., and Nevzorov, A. V.: On the effect of volcanic aerosol on variations of stratospheric ozone and NO2 according to measurements at the Siberian Lidar Station, Atmos. Ocean. Opt., 21, 825–831, 2008.

We also substituted Fig. 14 by a new one and corrected its caption.

**Instead of**
"**Figure 14.** Detection of the Rabaul volcanic plume together with PSCs, formed at extremely low temperatures (< –78 °C), in the stratosphere over Tomsk. Rabaul volcano erupted in Papua New Guinea on 7 October 2006. Temperature profiles were retrieved from radiosondes launched on 27 January 2007 in Kolpashevo (station 29231) at 00:00 UTC and in Novosibirsk (station 29634) at 12:00 UTC (WWW, 2007)."

**we wrote**
"**Figure 14.** Detection of PSCs formed at extremely low temperatures (< –78 °C) in the stratosphere over Tomsk. Temperature profiles were obtained from radiosondes launched on 27 January 2007 in Kolpashevo (station 29231) at 00:00 UTC and in Novosibirsk (station 29634) at 12:00 UTC (WWW, 2007). The dashed ellipse denotes the after-effect of the Rabaul volcanic eruption occurred in Papua New Guinea on 7 October 2006."
[**Page 19, lines 2–5, revised manuscript**]

**Comment:** P18, L26. By postulating that pyroCb smoke could increase "annual average" stratospheric aerosol, it would imply that prior research has come to similar conclusions. I'm not aware of any such finding. Please modify this statement appropriately.

**Response:** You are probably right. We agree that extensive forest (bush) fires (and pyroCb smoke) could hardly be expected to increase the **annual average** $B_\pi^a$ value. We removed discussion about forest extensive fires and pyro-cumulonimbus clouds from Sect. 3.5 (without prejudice to the foregoing) and retained it only in Sect. 4.

The following part of the Sect. 3.5 was removed:

"~~Extensive forest (bush) fires could be another cause of occasional increases of the annual average $B_\pi^a$ value. Combustion products (gases and aerosol particles) can reach the stratospheric altitudes via convective ascent within pyro cumulonimbus (pyroCb) clouds (see, e.g., Fromm et al., 2006). For example, the smoke plumes from the strong bush fire, occurred near the Australian city of Melbourne on 7 February 2009, were observed in the local stratosphere at an altitude of ~18 km (Siddaway and Petelina, 2011). In recent years, the number and intensity of massive forest fires have significantly increased (e.g., in the USA; Trickl et al., 2013) due to the climate change. The smoke-filled air masses frequently enter the stratosphere over the South of Western Siberia from North America, where extensive forest fires occur. Their smoke plumes are most likely to be detected as the SAL perturbations over Tomsk. However, more detailed information about the pyroCb events is required for their correct identification. Thus, both the Kelut plume and smoke plumes from massive forest fires in the USA and Canada, with equal probability, could be the cause of the increase of $B_\pi^a$ value in the stratosphere over Tomsk during the first quarter of 2015.~~"

**Comment:** P19, L13. "always" was not proved or demonstrated herein. Hence a citation is needed, or this point should be restated or removed.

**Response:** We agree. This "always" was removed from the text. Some of plumes from the tropical volcanoes eruptions can completely be within the Southern Hemisphere stratosphere. The tropical eruptions which plumes were within the Northern Hemisphere stratosphere and passed over Tomsk are presented in our manuscript.

**Instead of**

"Additional aerosol loading of the tropical reservoir  an increase in the annual average $B_\pi^a$ value in the Northern Hemisphere mid-latitude stratosphere via the meridional transport in the cold seasons (October to March)."

**we wrote**

"Additional aerosol loading of the tropical reservoir can usually lead to an increase in the annual average $B_\pi^a$ value in the Northern Hemisphere mid-latitude stratosphere via the meridional transport in the cold seasons (October to March; Hitchman et al., 1994)."
**[Page 21, lines 6–8, revised manuscript]**

Hitchman, M.H., McKay, M., Trepte, C.R.: A climatology of stratospheric aerosol, J. Geophys. Res., 99, 20689-20700, doi:10.1029/94JD01525,1994.

**Comment:** P19, L18. It's not clear what is meant here. All volcanic plumes are represented as an initial point source. Please clarify.

**Response:** OK. We rewrote and clarified the sentence.

**Instead of**

"On the other hand, by contrast to tropical volcanoes, the northern ones represent point sources of volcanic gas, aerosol, and ash plumes. Their corresponding air-mass trajectories can either pass over a lidar station or pass it by."

**we wrote**

"On the other hand, by contrast to tropical volcanoes, the narrow volcanic gas, aerosol, and ash plumes from northern volcanoes can either pass over a lidar station or pass it by."
[**Page 21, lines 11 and 12, revised manuscript**]

**Comment:** P20, L15. What is meant by "thermal speed?" Please clarify.

**Response:** This is the speed achieved due to heating of erupted mass in the "gas thrust" and "convective thrust" regions of an eruption column, not due to any potential (conservative) fields like magnetic, Coulomb or gravitational fields. The word "thermal" was removed from the text.

**Comment:** P20, L15. This is confusing. A plume is buoyant because it is less dense than the surrounding air. This makes it sound as if the surrounding umbrella region is less dense.

**Response:** We agree with this comment. The sentence was rewritten accordingly.

**Instead of**

"The most heated fraction of gas-vapor emissions from the "convective thrust region" has the high  speed and, therefore, can penetrate through the -density "umbrella region" of the eruption column and reach altitudes higher than $H_{MPA}$ (Raible et al., 2016)."

**we wrote**

"The most heated fraction of gas-vapor emissions from the "convective thrust region" has the highest speed and, therefore, can penetrate through the higher-density "umbrella region" of the eruption column and reach altitudes higher than $H_{MPA}$ due to the cumulative (jet) effect (Raible et al., 2016)."
**[Page 21, lines 25–27, revised manuscript]**

**And**

**Instead of**

"In addition to volcanoes, PSCs also represent a cause of significant SAL perturbations ."

**we wrote**

"In addition to volcanoes, PSCs also represent a cause of significant SAL perturbations."
[**Page 22, line 1, revised manuscript**]

**Comment:** P20, L24. "drifting" needs to be clarified. It implies to me that the ozone feature is drifting with the wind, whereas the papers previously offered in these comments show that the minihole is tied to dynamics and thus move with the speed of the synoptic-scale wave, not the wind.

**Response:** The sentences "The possibility of PSCs to form and be detected in the mid-latitudes is usually related with the presence of "mini ozone holes" drifting over lidar measurement points. As the lifetime of these "holes" is sufficiently short in the mid-latitudes, the PSC observations can be only occasional." were removed from Sect. 4.

**Comment:** P20, L29. The impact of pyroCb smoke on annual average stratospheric aerosol has not been shown in the literature to my knowledge. Please cite a paper or modify this statement accordingly.

**Response:** Yes, we agree. See, please, our **Response** to **Comment:** P18, L26 concerning the **annual average** $B_\pi^a$ value.

**Comment:** P20, L32. The measures in acres and square km are not equivalent. Please correct this.

**Comment:** P21, L2. What is the reason to call out the Happy Camp fire?

**Response:** Thank you. We corrected the mistake: 134 acres $\approx$ 0.543 km$^2$. However, we removed the information about the Happy Camp fire from the text of our manuscript.

**We rewrote a substantial part of Sect. 4 to make it clearer.**

**Instead of**

"Smoke plumes from strong forest (bush) fires can reach the stratospheric altitudes (Fromm et al., 2006; Siddaway and Petelina, 2011), spread out to great distances, and perturb the SAL state over different regions, including Tomsk. This is known to result in a measurable increase in the annual average $B_\pi^a$ value. Due to the climate warming, the number and intensity of massive forest fires have significantly increased in the last few years (Wotton et al., 2010). For example, about 137 strong forest fires were registered in the Northwest Territories of Canada in July 2014 (CBC News, 2014), and the Happy Camp Complex fire (41.80° N, 123.37° E) eventually consumed more than 134 acres (~0.543 km$^2$) of forests in California in August–October 2014. According to the California Department of Forestry and Fire Protection (CAL FIRE, http://www.ca.gov/) data, the Happy Camp Complex fire is in the list of the "Top 20 Largest California Wildfires". The smoke plumes from these mentioned massive forest fires could probably cause the increase in $B_\pi^a$ value in the stratosphere over Tomsk in January–March 2015. More detailed information about the pyroCb events such as precise time, place, and smoke plume altitude is required to correctly assign the pyroCb plumes to the corresponding aerosol layers over an observation point via the HYSPLIT model trajectory analysis. It is quite possible that some after-effects of strong forest fires occurred, e.g., in North America could be detected over Tomsk, but not identified during lidar observations in Tomsk (1986–2015)."

**we wrote**

"Extensive forest (bush) fires could be another cause of occasional increases of the $B_\pi^a$ value. Combustion products (gases and aerosol particles) can reach the stratospheric altitudes via convective ascent within pyro-cumulonimbus (pyroCb) clouds (see, e.g., Fromm et al., 2006). For example, the smoke plumes from the strong bush fire, occurred near the Australian city of Melbourne on 7 February 2009, were observed in the local stratosphere at an altitude of ~18 km (Siddaway and Petelina, 2011). Due to the climate warming, the number and intensity of massive forest fires have considerably increased in the last few years (Wotton et al., 2010). For example, about 137 strong forest fires were registered in the Northwest Territories of Canada in July 2014 (CBC News, 2014). The smoke-filled air masses frequently enter the stratosphere over the South of Western Siberia from North America, where extensive forest fires occur. Their smoke plumes are most likely to be detected as the SAL perturbations over Tomsk. However, more detailed information about the pyroCb events is required for their correct identification. It is quite possible that some after-effects of strong forest fires occurred, e.g., in North America could be detected over Tomsk, but not identified during lidar observations in Tomsk (1986–2015)."
[**Page 22, lines 5–14, revised manuscript**]

**Comment:** P21, L4. Regarding pyroCb smoke, I believe that there would be many occasions through the years for stratospheric observations at Tomsk. But of course the frequency of such observations would decrease rapidly with altitude above the tropopause. The highest smoke layers in the northern hemisphere to my knowledge are 19 km. They are much more likely to be at 12-15 km. Perhaps it would be to your advantage to directly employ the local tropopause height in the search for pyroCb smoke instead of a fixed altitude (e.g. 15 km) that is generally 4 km above the tropopause.

**Response:** We agree that the lower limit of 12 km for aerosol observations is better than of 15 km. Anyway, the perturbed scattering ratio profiles were also detected at altitudes higher than 15 km.

**Comment:** Technical Comments See the accompanying pdf of the manuscript. Please also note the supplement to this comment:

http://www.atmos-chem-phys-discuss.net/acp-2016-792/acp-2016-792-RC2-supplement.pdf.

**Response:** We corrected the text in accordance with Dr. Fromm's technical comments.

**Comment: in the Technical Comments**: Trickl did not conclude that the 14.3 km aerosol was from Eyja. They were more circumspect about all the aerosols above the tropopause because back trajectories did not come close to Iceland or pass near Iceland during a big eruption day.

**Response:** At least, Trickl noted in his article the following:

(Abstract) A key observation for judging the role of eruptions just reaching the tropopause region was that of the plume from the Icelandic volcano Eyjafjallajökull above Garmisch-Partenkirchen (April 2010) due to the proximity of that source. The top altitude of the ash above the volcano was reported just as 9.3 km, but the lidar measurements revealed enhanced stratospheric aerosol up to 14.3 km.)

(Sect. 4.2, page 5215) The upper boundary of these aerosol layers was roughly 12 km on 17 and 19 April, and 14.3 km on 20 April (Fig. 9), i.e., above the thermal tropopause that was 10.2 km on average on these days.

Sincerely,

Authors

---

## Author Comment (AC3) · 20 Jan 2017

**Manuscript Number: acp-2016-792**

Manuscript Type: Research article

**Title:** 30-year lidar observations of the stratospheric aerosol layer state over Tomsk (Western Siberia, Russia)

**List of corrections made according to **Referee 1** and **Dr. Fromm** comments**

**We substituted Figures 1, 2, 3, 5a, 11, 12, 13, and 14 by new ones and corrected their captions.**

**New Figures 5b, 6c, were added to the manuscript.**

**Page 1**

**Instead of**

"We also make an assumption that both the Kelut volcano plume (Indonesia, February 2014) and smoke plumes from massive forest fires occurred in Canada (137 fires in the Northwest Territories, July 2014) and the USA (the Happy Camp Complex fire in California, August–October 2014), with equal probability, could be the cause of the SAL perturbations over Tomsk during the first quarter of 2015."

**we wrote**

"We also make an assumption that the Kelut volcano eruption (Indonesia, February 2014) could be the cause of the SAL perturbations over Tomsk during the first quarter of 2015."

**[Page 1, lines 27 and 28, revised manuscript]**

**Page 2**

**Instead of**

"The volcanogenic aerosol perturbs the radiation-heat balance of the atmosphere, and thus, significantly affects the atmospheric dynamics (Timmreck, 2012; Driscoll et al., 2012)."

**we wrote**

"The volcanogenic aerosol perturbs the radiation-heat balance of the atmosphere, and thus, significantly affects the atmospheric dynamics and climate (Timmreck, 2012; Driscoll et al., 2012; Kremser et al., 2016)." [Page 2, lines 7 and 8, revised manuscript]

**Page 3**

**Instead of**

"...can be useful, e.g., in studying climate change."

**we wrote**

"...can be useful, e.g., in studying climate change (Mills et al., 2016)." [Page 3, lines 3 and 4, revised manuscript]

**The following sentence was added to the manuscript:**

"Note that the CIS-LiNet station located in Minsk, Belarus, is also integrated into the European Aerosol Research Lidar Network (EARLINET; Wandinger et al., 2016)." [Page 3, lines 9 and 10, revised manuscript]

**Instead of**

"...FEU-130 operating in the photon counting mode."

**we wrote**

"...FEU-130 (USSR, Moscow Elecro-Lamp Plant) operating in the photon counting mode." [Page 3, lines 25 and 26, revised manuscript]

**Page 4**

**The following sentence was added to the manuscript:**

"A more detailed technical description of the SLS aerosol channel and its data acquisition electronics can be found, e.g., in (Burlakov et al., 2010)."

[Page 4, lines 10 and 11, revised manuscript]

**The following sentence was added to the manuscript:**

" $\pi$  denotes an angle of  $\pi$  radian, i.e. the angle of the backscatter lidar signal propagation." [Page 4, lines 14 and 15, revised manuscript]

**Instead of**

"In our case, the tropopause altitude over Tomsk varies from ~11 to 13 km, depending on season, and therefore, we set  $H_1 = 15$  km."

**we wrote**

"Tomsk is located near the southern boundary of subarctic latitudes, where the tropopause altitude can significantly vary, e.g., due to migration of the Arctic stratospheric jet stream within the Tomsk region. Sometimes one can observe a double (or even multiple) tropopause. For this reason, we consciously removed the interval of the tropopause altitude variations to observe the stratospheric perturbations only. As the tropopause altitude over Tomsk varies from ~11 to 13 km, depending on season, we set  $H_1 = 15$  km."

**[Page 4, lines 23–26, revised manuscript]**

**Page 5**

**Instead of**

"Figure 1. ... The red bars correspond to tropical volcanic eruptions, whereas the black ones correspond to eruptions of extratropical volcanoes located in the Northern Hemisphere. PSC: polar stratospheric clouds."

**we wrote**

"Figure 1.... The red bars correspond to tropical volcanic eruptions, whereas the black ones correspond to eruptions of extratropical volcanoes located in the Northern Hemisphere. The thin horizontal line in Fig. 1 indicates the minimum value of the annual average  $B_{\pi}^{a}$  reached in 2004. PSC: polar stratospheric clouds."

**[Page 5, lines 16 and 17, revised manuscript]**

**Page 7**

**Figure 2 is now Figure 3 and, conversely, Figure 3 is now Figure 2 in the revised manuscript.**

**Instead of**

"Figure 1. ... The red bars correspond to tropical volcanic eruptions, whereas the black ones correspond to eruptions of extratropical volcanoes located in the Northern Hemisphere. PSC: polar stratospheric clouds."

**we wrote**

"Figure 2. Inter-annual variations of  $B_{\pi}^{a}$  values (in the stratosphere over Tomsk) separately averaged over the warm and cold half-years. The "warm" and "cold" average points are assigned to 1 June of the current year and 1 January of the next year, respectively. Black and red vertical bars at the bottom of the figure indicate volcanic eruptions as in Fig. 1 (see also Table 1). All error bars represent the standard deviation." [Page 7, lines 10–13, revised manuscript]

**Page 7, lines 10–13, revised manuscri**

**Instead of**

"...background level of  $B_{\pi}^{a}$  reached after 1999. Note that only..."

**we wrote**

"...background level of  $B_{\pi}^{a}$  reached after 1998. Only...."

**[Page 7, line 16, revised manuscript]**

**Instead of**

"The minimum annual average  $B_{\pi}^{a}$  values were reached in 2003–2004."

**we wrote**

"The minimum annual average  $B_{\pi}^{a} = 1.29 \times 10^{-4} \text{ sr}^{-1}$  was reached in 2004."

**[Page 7, line 18, revised manuscript]**

**Instead of**

"The thin horizontal line in Fig. 1 indicates the minimum value of the annual average  $B_{\pi}^{a}$  reached in 2004. This value equals to that determined in 1979 and considered as the background one (Trickl et al., 2013)."

**we wrote**

"Note that taking into account the spectral dependence of  $B_{\pi}^{a}$ , its minimum annual average value observed in Tomsk at  $\lambda = 532$  nm in 2004 was close to that determined for Garmisch-Partenkirchen at  $\lambda = 694$  nm in 1979 and considered as the background by Trickl et al. (2013)."

**[Page 7, line 20 and Page 8, lines 1 and 2, revised manuscript]**

**Pages 8 and 9**

We considerably rewrote Sect 3.1. See, please, the colored version of our revised manuscript for details.

**Pages 10 and 11**

**Section 3.2.1 was deeply revised and rewritten. Figures 5b and 6c were added.**

**Instead of**

[revised manuscript text omitted]

---

## Author Comment (AC4) · 20 Jan 2017

The file "Data for Figure 1.opj" can be opened with the program Origin (e.g., Origin 8.5 or older versions).

Please also note the supplement to this comment:
http://www.atmos-chem-phys-discuss.net/acp-2016-792/acp-2016-792-AC4-supplement.zip

---

## Referee Report (RR1)

**Manuscript Number:** acp-2016-792
**Manuscript Type:** research article
**Title:** 30-year lidar observations of the stratospheric aerosol layer state over Tomsk (Western Siberia, Russia)

We thank Referee 1 for valuable comments, questions and suggestions which will allow us to improve our manuscript.

**Point-by-point response to Referee 1**

**2 Lidar instruments and methods**

**Comment:** I would like to see a short description of the lidar photon-counting data acquisition electronics. Is it from a commercial vendor or built just for the lidar?

**Our response:** The SLS aerosol channel data acquisition electronics we used is of in-house design and manufacture. A more detailed technical description of the SLS aerosol channel and its data acquisition electronics can be found, e.g., in (Burlakov et al., 2010). We will include the reference in the final version of our manuscript.

Burlakov, V. D., Dolgii, S. I., and Nevzorov, A. V.: A three-frequency Lidar for sensing microstructure characteristics of stratospheric aerosols, Instrum. Exp. Tech., 53, 890–894, doi:10.1134/S0020441210060230, 2010.

**We added the following sentence to the revised manuscript:**

"A more detailed technical description of the SLS aerosol channel and its data acquisition electronics can be found, e.g., in (Burlakov et al., 2010)."
[**Page 4, lines 10 and 11, revised manuscript**]

**Comment:** Is signal induced noise, and counting saturation taken into account?

**Response:** Yes. We take into account both photomultiplier tube (PMT) afterpulses and photon-counting saturation. Furthermore, current pulses from a PMT are fed to a broadband amplifier and a differential amplitude discriminator. The latter allows controlling the lower and upper discrimination thresholds of dark-current pulses of a particular PMT specimen under the conditions of real background illumination, i.e., selection of the optimal discrimination thresholds for increasing the signal-to-noise ratio (see, please, Sect. "Technical description of the lidar" in the mentioned paper Burlakov et al., 2010).

**Comment:** I would like to see more about the normalization. Is only a single altitude used? Is it 30 km? Many of the Scattering Ratio profiles shown in the paper haven't decreased to 1.0 at 30 km, so higher altitude data must have been used. Is there an objective way to do this?

**Response:** The SLS aerosol channel makes it possible to receive almost undisturbed backscattered signals from altitudes of ~ 40–45 km. At higher altitudes, the signal-to-noise ratio is too low. Therefore, altitudes of ~30–35 km, where the stratosphere is considered to be aerosol-free, were used as the calibration altitudes.

**Comment:** Choosing H1 (lower attitude of the SAL) as 15 km seems reasonable, but other lidar groups have used the actual tropopause or tropopause + 1 km. Choosing this altitude can be complicated since there can be multiple tropopauses sometimes. It can also be complicated when there has been an eruption since the upper troposphere can have much more aerosol. But perhaps you can comment on how much of a difference it would make to lower the H1 altitude during background conditions.

**Response:** Tomsk is located near the southern boundary of subarctic latitudes, where the tropopause altitude can vary significantly (from ~11 to 13 km, depending on season), e.g., due to migration of the Arctic stratospheric jet stream within our (Tomsk) region. Sometimes one can observe a double (or even multiple) tropopause. Therefore, we consciously removed the interval of the tropopause altitude variations to observe the stratospheric perturbations only. Probably, some information about the lowermost stratospheric aerosol perturbations could be missed.

**Instead of**
"In our case, the tropopause altitude over Tomsk varies from ~11 to 13 km, depending on season, and therefore, we set $H_1 = 15$ km."

"Tomsk is located near the southern boundary of subarctic latitudes, where the tropopause altitude can significantly vary, e.g., due to migration of the Arctic stratospheric jet stream within the Tomsk region. Sometimes one can observe a double (or even multiple) tropopause. For this reason, we consciously removed the interval of the tropopause altitude variations to observe the stratospheric perturbations only. As the tropopause altitude over Tomsk varies from ~11 to 13 km, depending on season, we set $H_1 = 15$ km."
[**Page 4, lines 23–26, revised manuscript**]

**3 Results of the SAL lidar observations over Tomsk**

**3.1 Time series of the integrated stratospheric backscatter coefficient (1986–2015)**

**Comment**: In Table 1 the maximum plume height is listed. How are these measured? The initial plume heights are not very accurate if done by naked-eye observations. Are these measured later with lidars?

**Response**: The maximum plume altitudes (MPAs) presented in Table 1 were taken from the Smithsonian Institution Global Volcanism Program. The Smithsonian MPA values were determined from the pooled analysis of visual and radar observations. When it was possible, the MPAs could also be determined with space-borne lidars more accurately compared to the mentioned observation methods. Considering in Sect. 3.5 the 2014 Kelut eruption as an example, we discussed the difference between MPA values determined via the space-borne lidar CALIOP and visual/radar observations.

**Comment**: There has been an ongoing discussion in the community about whether there is an annual cycle in SAL. Your Figure 2 shows a winter/summer ratio of about 1.35. Figure 3 is similar. This would be influenced by the choice of the H1 altitude. It would be interesting to calculate a ratio and error bar as your best estimate of an annual cycle. In Figure 2 what do the error bars represent, one sigma of the spread of the data?

**Response**: First of all, Figure 2 is now Figure 3 and, conversely, Figure 3 is now Figure 2 in the revised manuscript. We have extended the analyzed period of the background aerosol loading variations over Tomsk up to 16 years (1999–2015), instead of the period 2000–2006 in the old version of our manuscript. The monthly average $B_\pi^a$ data for March–June 2000 (after the Hekla eruption), August–November 2008 (after the Okmok and Kasatochi eruptions), August–October 2009 (after the Sarychev Peak eruption), and also April and August–October 2011 (after the Merapi, Grimsvötn, and Nabro eruptions) were not taken into account. The exclusion of these perturbed data allowed us to extend the analyzed period of the background aerosol loading variations and, therefore, to improve the statistical reliability of the $B_\pi^a$ data series. The $B_\pi^a$ values were averaged separately for the westerly and easterly phases of the quasi-biennial oscillations (QBO) characterized by zonal winds in the equatorial region at 30 mbar (Fig. 3, ex-Fig. 2). Based on these 16-year averaged values of the monthly average $B_\pi^a$, we show that an annual cycle in SAL exists. Moreover, based on the inter-annual $B_\pi^a$ variations (Fig. 2, ex-Fig. 3) separately averaged over the warm (April to September) and cold (October to March) half-years, we also show that aerosol loading of the mid-latitude stratosphere is maximal in the cold half-year, when the meridional air mass transport dominates (especially during the westerly phase of the QBO), and it is minimal in the warm half-year, when the zonal transport dominates. All error bars in Figs. 2 and 3 represent the standard (1–σ) deviation. As a whole, we considerably rewrote Sect 3.1. See, please, the colored version of our revised manuscript for details.

**3.2.1 Okmok and Kasatochi**

**Comment**: In Figure 5 there is a peak on September 4, 2000 at 27-30 km. Is that real?

**Response**: Yes, you are right. There are two small peaks of the scattering ratio profile (4 September 2000) at altitudes of 27 to 30 km in Figure 5. However, we believe that these peaks are not related to any aerosol events in the stratosphere and, therefore, they represent measurement uncertainties due to the low signal-to-noise ratio.

**Comment**: The altitude axis might be clearer with 1 km tic marks instead of 1.5 km.

**Response**: We agree. Done.

**4 Discussion and conclusion**

**Comment**: "The Happy Camp Complex fire consumed more than 134 acres (~543 km2)...". Acres are much smaller than km2, something is wrong with the areas.

**Response:** Thank you. We corrected the mistake: 134 acres $\approx 0.543$ km$^2$. However, we removed the information about the Happy Camp fire from the text of our manuscript.

**Comment:** Figures 6 (a), 6 (b), 9, and 11. I am amazed that all four trajectories almost exactly cross over the volcanoes after so many days. Did you find an optimum altitude or something that gave you the best trajectory? Were the trajectories sensitive to the initial conditions?

**Response:** We usually calculate a set of the HYSPLIT backward trajectories which start from different aerosol layers detected in the stratosphere over Tomsk after a volcano eruption (See, please, Fig. 11 in our revised manuscript). A signal integration period of 20–30 min yields uncertainties in the start time and altitude (i.e. in initial conditions) of a backward trajectory. Therefore, we usually present the best trajectory to illustrate a possible way of how aerosol from an erupted volcanic plume could pass and be detected over Tomsk.

Sincerely,

Authors

**Manuscript Number:** acp-2016-792
**Manuscript Type:** Research article
**Title:** 30-year lidar observations of the stratospheric aerosol layer state over Tomsk (Western Siberia, Russia)

**Point-by-point response to Dr. Fromm**

**General comments**

**Comment:** This manuscript gives a wide-ranging analysis of the aerosol lidar data collected at a western Siberia location. The long temporal span of this data set, for a location far removed from other ground-based lidars, is a welcome addition to the globe's sparse stratospheric aerosol archive. Considering that these data are presumably still being collected adds value for continued monitoring of the upper troposphere and lower stratosphere. Consequently this is a potentially critical work that may offer significant value to our understanding. The authors' approach is similar to many prior works that examine a particular lidar's data set in the context of other data sets and meteorological analyses. They present some individual lidar profiles, long-term time series, and annual cycle analyses. This is a very appropriate study for Atmospheric Chemistry and Physics.

**Our response:** We thank Dr. Fromm for his great interest in our work and comprehensive criticism, comments, questions, and suggestions which will definitely allow us to improve our manuscript.

**Comment:** Along with these assets, the paper has several major weaknesses. My assessment is that the weaknesses substantially limit the value of the manuscript in its present form. For the paper to merit publication, attention to these major concerns must be paid. One major concern is a significant absence of recognition of work that has direct bearing on the analyses and conclusions presented herein. The effect goes beyond that of showing insufficient background work. It extends to the realization that critical scientific aspects of stratospheric aerosol and cloud have not been taken into account in the interpretation of the Tomsk lidar data. Detailed examples are given below. Another major concern is that the Tomsk data may be biased with respect to other similar lidar data sets, as called out specifically in a comment below.

**Response:** We have tried to take into account all weaknesses specified by Dr. Fromm.

**Comment:** The manuscript offers no direct or even indirect comparisons of the Tomsk lidar data with any other aerosol data sets; hence it is not possible to assess the accuracy of the aerosol data in the full range of stratospheric aerosol loading. Another concern is that the authors utilize a local measurement data set of total ozone abundance, but without any qualification. The reader is left with great uncertainty as to the robustness of the results and certain interpretations the authors give for certain phenomena.

**Response:** Such data comparisons exist and were made with, e.g., data of the Minsk lidar station (Belarus) which is simultaneously incorporated in 1) the CIS-LiNet together with the Tomsk lidar station (Chaykovskii et al., 2005; Zuev et al., 2009); and 2) the EARLINET (Wandinger et al., 2016). Moreover, the Tomsk station measurement data were widely used, e.g., by Ridley et al. (2014).

Chaykovskii, A. P., Ivanov, A. P., Balin, Yu. S., El'nikov, A. V., Tulinov, G. F., Plusnin, I. I., Bukin, O. A., and Chen, B. B.: CIS-LiNet lidar network for monitoring aerosol and ozone: methodology and instrumentation, Atmos. Ocean. Opt., 18, 958–964, 2005.

Zuev, V. V., Balin, Yu. S., Bukin, O. A., Burlakov, V. D., Dolgii, S. I., Kabashnikov, V. P., Nevzorov, A. V., Osipenko, F. P., Pavlov, A. N., Penner, I. E., Samoilova, S. V., Stolyarchuk, S. Yu., Chaikovskii, A. P., and Shmirko, K. A.: Results of joint observations of aerosol perturbations of the stratosphere at the CIS-LiNet network in 2008, Atmos. Ocean. Opt., 22, 295–301, 2009.

Wandinger, U., Freudenthaler, V., Baars, H., Amodeo, A., Engelmann, R., Mattis, I., Groß, S., Pappalardo, G., Giunta, A., D'Amico, G., Chaikovsky, A., Osipenko, F., Slesar, A., Nicolae, D., Belegante, L., Talianu, C., Serikov, I., Linné, H., Jansen, F., Apituley, A., Wilson, K. M., de Graaf, M., Trickl, T., Giehl, H., Adam, M., Comerón, A., Muñoz-Porcar, C., Rocadenbosch, F., Sicard, M., Tomás, S., Lange, D., Kumar, D., Pujadas, M., Molero, F., Fernández, A. J., Alados-Arboledas, L., Bravo-Aranda, J. A., Navas-Guzmán, F., Guerrero-Rascado, J. L., Granados-Muñoz, M. J., Preißler, J., Wagner, F., Gausa, M., Grigorov, I., Stoyanov, D., Iarlori, M., Rizi, V., Spinelli, N., Boselli, A., Wang, X., Lo Feudo, T., Perrone, M. R., De Tomasi, F., and Burlizzi, P.: EARLINET instrument intercomparison campaigns: overview on strategy and results, Atmos. Meas. Tech., 9, 1001-1023, doi:10.5194/amt-9-1001-2016, 2016.

Ridley, D. A., et al. (2014), Total volcanic stratospheric aerosol optical depths and implications for global climate change, Geophys. Res. Lett., 41, 7763–7769, doi:10.1002/2014GL061541.

**Specific comments**

**Comment:** Somewhere in this paper the recent review paper by Kremser et al. should be mentioned and cited.

Kremser, S., et al. (2016), Stratospheric aerosol – Observations, processes, and impact on climate, Rev. Geophys., 54, doi:10.1002/2015RG000511.

**Response:** OK. We added this reference to our manuscript.

**Instead of**
"The volcanogenic aerosol perturbs the radiation-heat balance of the atmosphere, and thus, significantly affects the atmospheric dynamics (Timmreck, 2012; Driscoll et al., 2012)."

**we wrote**
"The volcanogenic aerosol perturbs the radiation-heat balance of the atmosphere, and thus, significantly affects the atmospheric dynamics and climate (Timmreck, 2012; Driscoll et al., 2012; Kremser et al., 2016)."
[**Page 2, lines 7 and 8, revised manuscript**]

Kremser, S., Thomason, L. W., von Hobe, M., Hermann, M., Deshler, T., Timmreck, C., Toohey, M., Stenke, A., Schwarz, J. P., Weigel, R., Fueglistaler, S., Prata, F. J., Vernier, J.-P., Schlager, H., Barnes, J. E., Antuña-Marrero, J.-C., Fairlie, D., Palm, M., Mahieu, E., Notholt, J., Rex, M., Bingen, C., Vanhellemont, F., Bourassa, A., Plane, J. M. C., Klocke, D., Carn, S. A., Clarisse, L., Trickl, T., Neely, R., James, A. D., Rieger, L., Wilson, J. C., and Meland, B.: Stratospheric aerosol – Observations, processes, and impact on climate, Rev. Geophys., 54, 278–335, doi:10.1002/2015RG000511, 2016.

**Comment:** There is no citation of Vernier et al (GRL, 2011), Ridley et al. (GRL, 2014), Santer et al. (Nature Geo., 2014) and some other very relevant recent papers on volcanoes and recent stratospheric aerosol layer trends.

**Response:** Thank you for these references. However, these and other very relevant recent papers were considered in Kremser et al. (2016), which we already cited.

**Comment:** P2, L10. Consider citing

Jäger, H. and Wege, K.: Stratospheric Ozone Depletion at Northern Midlatitudes after Major Volcanic Eruptions, J. Atmos. Chem., 10, 273-287, 1990.

**Response:** Thank you for the reference, but the data and results of this paper were mentioned in Trickl et al. (2013), which we also cited in our manuscript.

**Comment:** P2, L5. "vulcanian" refers to a specific style of eruption, the characteristics of which are not associated with stratospheric injection. Is the term even needed to make the point of this sentence?

**Response:** According to the Smithsonian Institution Global Volcanism Program, gas and ash plumes from some volcanic eruptions (with VEI ≥ 3) of the Vulcanian type can directly reach the stratospheric altitudes:

https://web.archive.org/web/20111110173623/http://www.volcano.si.edu/world/eruptioncriteria.cfm

(See there, please, Section "VEI (Volcanic Explosivity Index)" and Table containing the Eruption Type and Cloud Column Height).

**Comment:** P3, L19. How about measurement frequency? What was the lidar operation frequency and regularity? It would be important for the reader to know what is typical for the number of profiles that go into the 10-day average.

**Response:** The lidar operation regularity depends on the weather conditions, of course. The absence of clouds is required for stratospheric aerosol measurements. As seen in Fig. 1, the measurement frequency allows us to retrieve the 10-day average values of the integrated stratospheric backscatter coefficient, when it is possible. See, please, also the "10-day average values" file in **the Supplement** containing data for Fig. 1.

**Comment:** P3, L23. Consider giving the name of the PMT manufacturer.

**Response:** The manufacturer of PMTs "FEU-130" is the Moscow Elecro-Lamp Plant (MELZ).

http://www.melz-evp.ru/about.html

**Instead of**

"The signals were registered with a vertical resolution of 374 m by a photomultiplier tube (PMT) FEU-130 operating in the photon counting mode."

**we wrote**

"The signals were registered with a vertical resolution of 374 m by a photomultiplier tube (PMT) FEU-130 (USSR, Moscow Elecro-Lamp Plant) operating in the photon counting mode."
[**Page 3, lines 25 and 26, revised manuscript**]

**Comment:** P4, L4. This statement arouses my curiosity to know if this lidar made any observations of the Chelyabinsk bolide plume in 2013. It was observed by satellite aerosol profilers above 30 km.

Gorkavyi, N., D. F. Rault, P. A. Newman, A. M. da Silva, and A. E. Dudorov (2013), New stratospheric dust belt due to the Chelyabinsk bolide, Geophys. Res. Lett., 40, 4728–4733, doi:10.1002/grl.50788.

**Response:** We have rechecked our lidar measurement data (both scattering ratios and integrated stratospheric backscatter coefficients). The Chelyabinsk bolide plume was not detected over Tomsk in 2013 (See, please, red lines from 977 to 992 in the "10-day average values" file in **the Supplement** to check the integrated stratospheric backscatter coefficient data). So, the belt was not over Tomsk.

**Comment:** P4, L10. What does the pi character refer to? Please consider stating its meaning.

**Response:** The $\pi$ character denotes an angle of $\pi$ radian (180 degrees), i.e. the angle of the backscatter lidar signal propagation.

**We have added the following sentence to the revised manuscript:**
"$\pi$ denotes an angle of $\pi$ radian, i.e. the angle of the backscatter lidar signal propagation."
[**Page 4, lines 14 and 15, revised manuscript**]

**Comment:** P4, L20. This is understandable and defensible. But the median tropopause at Tomsk is such that a lot of the lowermost stratosphere is not sampled with the fixed lower limit of 15 km. For instance, many volcanic plumes just above the tropopause have occurred in recent years (Kasatochi is one example). And pyrocumulonimbus smoke plumes are routine in summer, but usually between the tropopause and ~15 km. It might be worthwhile to try other lower-boundary, for instance based on potential temperature or a tropopause-relative height offset.

**Response:** Yes, some stratospheric aerosol events could be missed due to the lower limit of 15 km for integrated stratospheric backscatter coefficient $B_\pi^a$ calculation. However, this limit definitely excludes any tropospheric aerosol events. Moreover, we usually determined the scattering ratio $R(H)$ starting from the lower limit of 12.5 km (Figs. 5, 7, 8, 12–15), or even from 10 km in the case of Eyjafjallajökull volcano (Fig. 10).

**Comment:** Figure 2. The error/uncertainty bars are not defined or even mentioned. Moreover, however they are defined, the majority of them extend farther than the winter/summer range. This suggests to me that this pattern of monthly averages is not particularly repeatable. Auth should consider discussion of the robustness of this pattern in light of the relatively large uncertainty. The feature that stands out here is the March average and uncertainty. Can the authors explain why this month's aerosol amount is so large and variable?

**Response:** Sect 3.1 was considerably rewritten. See, please, the colored version of our revised manuscript for details. For example, we have extended the analyzed period of the background aerosol loading variations over Tomsk up to 16 years (1999–2015) and averaged the $B_\pi^a$ values separately for the westerly and easterly phases of the quasi-biennial oscillations (QBO) characterized by zonal winds in the equatorial region at 30 mbar (Fig. 3, ex-Fig. 2). The monthly average $B_\pi^a$ data for March–June 2000 (after the Hekla eruption), August–November 2008 (after the Okmok and Kasatochi eruptions), August–October 2009 (after the Sarychev Peak eruption), and also April and August–October 2011 (after the Merapi, Grimsvötn, and Nabro eruptions) were not taken into account. The exclusion of these perturbed data allowed us to extend the analyzed period of the background aerosol loading variations and, therefore, to improve the statistical reliability of the $B_\pi^a$ data series. All error bars in Figs. 2 and 3 represent the standard (1–σ) deviation.

**Comment:** P6, L4. The minimum integrated backscatter coefficient value in Figure 1 and Trickl et al.'s background value are not equal, in contrast to the authors' claim. Figure 1's value is between 1e-4 and 2e-4. Trickl et al.'s 1979 background value is 5e-5, smaller by roughly a factor of two. Zuev et al.'s Pinatubo peak matches well with Trickl et al., but the comparable quiescent-period values are off by about a factor of 2. This sentence should be corrected. The differences in the background values should be acknowledged and discussed. Trickl integrated from a tropopause-relative altitude that probably includes more of the lowermost stratosphere than Zuev et al. yet their integral is much smaller. The wavelength difference is seemingly too small to account for the difference.

**Response:** Thank you for this comment. We agree that the background value of $B_\pi^a$ observed in Tomsk at $\lambda = 532$ nm in 2004 was not equal to the Trickl et al.'s background value determined for Garmisch-Partenkirchen at $\lambda = 694$ nm in 1979. We corrected this sentence. However, it is not reasonable to discuss and hardly possible to intercompare the 2004 Tomsk and 1979 Garmisch-Partenkirchen $B_\pi^a$ background values, because one atmospheric (air) circulation epoch was replaced by another one during 25 years (e.g., Chu, 2002; Chernavskaya et al., 2006). Therefore, the sources and type of the background aerosol were also changed.

**Instead of**

"The thin horizontal line in Fig. 1 indicates the minimum value of the annual average $B_\pi^a$ reached in 2004. This value equals to that determined in 1979 and considered as the background one (Trickl et al., 2013)."

**we wrote**

"Note that taking into account the spectral dependence of $B_\pi^a$, its minimum annual average value observed in Tomsk at $\lambda = 532$ nm in 2004 was close to that determined for Garmisch-Partenkirchen at $\lambda = 694$ nm in 1979 and considered as the background by Trickl et al. (2013)."

[**Page 7, line 20 and Page 8, lines 1 and 2, revised manuscript**]

Chu, P.-S.: Large-Scale Circulation Features Associated with Decadal Variations of Tropical Cyclone Activity over the Central North Pacific, Journal of Climate, 15, 2678–2689, doi:10.1175/1520-0442(2002)015<2678:LSCFAW>2.0.CO;2, 2002.

Chernavskaya, M. M., Kononova, N. K., and Val'Chuk, T. E.: Correlation between atmospheric circulation processes over the Northern Hemisphere and parameter of solar variability during 1899–2003, Adv. Space Res., 37, 1640–1645, doi:10.1016/j.asr.2005.06.022, 2006.

Trickl, T., Giehl, H., Jäger, H., and Vogelmann, H.: 35 yr of stratospheric aerosol measurements at Garmisch-Partenkirchen: from Fuego to Eyjafjallajökull, and beyond, Atmos. Chem. Phys., 13, 5205–5225, doi:10.5194/acp-13-5205-2013, 2013.

**Comment:** P7, L14-19. Discussion of the Brewer Dobson circulation and its impact on the extratropical SAL. Presumably the point here is that the pattern shown by Fig. 2 is consistent with the generalizations here such that a citation is needed for work(s) that show this intra-annual tendency.

**Response:** There is no need for a reference to any other work, because we can show the intra-annual tendency (an annual cycle in the mid-latitude SAL via the Brewer-Dobson circulation) based on our 16-year averaged values of the monthly average $B_\pi^a$ (Fig. 3, ex-Fig. 2). See, please, the rewritten Sect. 3.1 for details.

**Comment:** Figure 3. What is the averaging period of the data points?

**Response:** (Figure 3 is now Fig. 2). The averaging period for each **red** point (warm half-year) is from April to September of the corresponding year. The averaging period for each **blue** point (cold half-year) is from October of the current year to March of the next year. The "warm" and "cold" average points are assigned to 1 June of the current year and 1 January of the next year, respectively. Black and red vertical bars at the bottom of the figure indicate volcanic eruptions as in Fig. 1 (see also Table 1).

**Comment:** Since error bars were shown for Figure 2, they should also be shown here, and the implications discussed.

**Response:** Done. All error bars represent the standard deviation.

**Comment:** P8, L17-18. This statement does not reconcile with VEI=4. I recognize that VEI=4 is in the GVP database. So this sentence probably should call out the apparent discrepancy.

**Response:** Well, both VEI and the maximum plume altitude of the Eyjafjallajökull volcano eruption were taken from the GVP database, as it is noted in Table 1 caption. The VEI coefficient is responsible for the volcanic gas and ash volume, rather than for their rise altitude.

**Comment:** P10, L3. The trajectory analysis is largely unnecessary. There is no need to run trajectories for the 8 Aug observation; it's obvious that layer can't be from Kasatochi (and there is no other plausible source). For the September observation (and latter ones), the stratosphere at that time had Okmok and Kasatochi aerosols distributed all over. If the Tomsk back trajectory intersected with any of the Okmok layers prior to its pass over the Kasatochi volcano, that is a more plausible attribution than a 3.5 week trajectory connection.

**Response:** Well, we agree that the sentence "In the August of 2008, the detected aerosol layers were related only to the Okmok plume, but in the September of 2008, there was observed a superposition of plumes from both volcanoes." declares the obvious fact. The mentioned sentence was removed from the text. Regarding the usage of the trajectory analysis, we can say that all HYSPLIT trajectories were used only to illustrate and not to prove the results of our lidar observation. We noted in several places in our manuscript (including Sect. "Discussion and conclusion") that all the trajectories longer than 2 weeks can be considered only as probable ones. We also compared our vertical profiles of the scattering ratio $R(H)$ with the Minsk CIS-LiNet lidar station profiles (Zuev et al., 2009) and revealed the similar SAL perturbations over Minsk. As a whole, we considerably rewrote Sect 3.2.1. See, please, the colored revised version of our manuscript for details.

Zuev, V. V., Balin, Yu. S., Bukin, O. A., Burlakov, V. D., Dolgii, S. I., Kabashnikov, V. P., Nevzorov, A. V., Osipenko, F. P., Pavlov, A. N., Penner, I. E., Samoilova, S. V., Stolyarchuk, S. Yu., Chaikovskii, A. P., and Shmirko, K. A.: Results of joint observations of aerosol perturbations of the stratosphere at the CIS-LiNet network in 2008, Atmos. Ocean. Opt., 22, 295–301, 2009.

**Comment:** P10. The impression I got from reading this analysis was that Zuev et al. are attempting to draw general conclusions about the Kasatochi and Okmok plumes and the SAL. If they are instead limiting their assessment of the SAL to just where the Tomsk observations are, they should make that explicit. The more general SAL analysis of these plumes is already published but not cited in this manuscript. E.g. Bourassa et al., Anderssen et al., Kravitz et al. These make clear that the Kasatochi and Okmok plumes were observed at all altitudes from the tropopause to nearly 19 km.

Andersson et al. (2015), Significant radiative impact of volcanic aerosol in the lowermost stratosphere, DOI: 10.1038/ncomms8692.

Bourassa, A., D. Degenstein, B. Elash, and E. Llewellyn (2010), Evolution of the stratospheric aerosol enhancement following the eruptions of Okmok and Kasatochi: Odin-OSIRIS measurements, J. Geophys. Res., 115, D00L03, doi:10.1029/2009JD013274.

Kravitz, B., A. Robock, and A. Bourassa (2010), Negligible climatic effects from the 2008 Okmok and Kasatochi volcanic eruptions, J. Geophys. Res., 115, D00L05, doi:10.1029/2009JD013525.

**Response:** As we already pointed out above, we deeply rewrote Sect 3.2.1. to clarify the situation. First, our conclusion (based on the HYSPLIT trajectories), that the plumes from both Okmok and Kasatochi volcanoes reached altitudes of ≥16 km, is consistent with different satellite observation data, including the CALIOP data (Yang et al., 2010; Kristiansen et al., 2010; Prata et al., 2010). Second, the more general SAL analysis of these plumes was already published by Zuev et al. (2009) but not cited in the works published later by, e.g., Bourassa et al. (2010), Kravitz et al. (2010), and Anderssen et al. (2015). Nevertheless, we included Bourassa et al. (2010) and Andersson et al. (2015) to our manuscript as an example. See, please, the colored revised version of our manuscript for details.

Yang, K., Liu, X., Bhartia, P. K., Krotkov, N. A., Carn, S. A., Hughes, E. J., Krueger, A. J., Spurr, R. J. D., and Trahan, S. G.: Direct retrieval of sulfur dioxide amount and altitude from spaceborne hyperspectral UV measurements: Theory and application, J. Geophys. Res., 115, D00L09, doi:10.1029/2010JD013982, 2010.

Kristiansen, N. I., Stohl, A., Prata, A. J., Richter, A., Eckhardt, S., Seibert, P., Hoffmann, A., Ritter, C., Bitar, L., Duck, T. J., and Stebel, K.: Remote sensing and inverse transport modeling of the Kasatochi eruption sulfur dioxide cloud, J. Geophys. Res., 115, D00L16, doi:10.1029/2009JD013286, 2010.

Prata, A. J., Gangale, G., Clarisse, L., and Karagulian, F.: Ash and sulfur dioxide in the 2008 eruptions of Okmok and Kasatochi: Insights from high spectral resolution satellite measurements, J. Geophys. Res., 115, D00L18, doi:10.1029/2009JD013556, 2010.

**Comment:** P10, L12. My impression is that Zuev et al. are attempting to assess the accuracy of the injection height as reported by the GVP. The point made here, and below, regarding the accuracy of the Hmpa for the two eruptions, is of little consequence. I also get the impression that they are using the altitude of the back trajectory endpoint over the volcano as a point of comparison with Hmpa. If I have the wrong impression, perhaps other readers will be similarly affected. So I ask the authors to clarify the wording here. Otherwise, giving the precise altitude of a 3-4 week old trajectory much weight asks more of the trajectory model and the weather analyses than they can promise. Secondly, the Hmpa data in the GVP database is neither tightly constrained. Thirdly, it can be shown easily with CALIOP data that the Hmpa for these two eruptions is even farther off the mark than this analysis suggests. CALIOP data within a few days of each eruption shows that the injection altitude was at least 17 km (Okmok) and 18 km (Kasatochi). These data would offer a much more compelling analysis for this argument than the Tomsk data.

**Response:** We have rested on our experimental results obtained with the SLS lidar, not on the CALIOP data. We calculated a lot of trajectories and showed only several ones as an example. Note that our HYSPLIT trajectories in cases of Okmok and Kasatochi eruptions (started from Tomsk) definitely ended within their plumes over the volcanoes. Thus, these results of the trajectory analysis made are also in agreement with the CALIOP data (see, please, our response for your previous comment). We never claimed that HYSPLIT trajectories are able to determine the maximum plume altitudes of any volcanic eruptions. No essential conclusions were made on the basis of the HYSPLIT trajectories.

**Comment:** P11, L15. This is way too precise and exclusive. The Sarychev Peak eruption spanned several days, as shown in their table. The time of a 3+ week back trajectory is by its nature too uncertain to permit such a definitive connection.

**Response:** The trajectory (presented in Fig. 9 and started from one of the detected aerosol layers over Tomsk) shows a possible way how aerosol from the Sarychev Peak volcanic plume erupted in one of the eruption days (e.g. 15 June) could pass and be detected over Tomsk on 7 July. We consider that the trajectory also shows that the detected aerosol layer is not associated with the after-effect of the 2009 Redoubt eruption.

**Instead of**
"As seen in Fig. 9, this layer is associated with the backward trajectory passed over Sarychev Peak volcano at an altitude of ~13.8 km on 15 June at the moment of the eruption, 17:30 UTC."

**we wrote**
"This layer is seen in Fig. 9 to be: 1) associated with the backward trajectory passed over Sarychev Peak volcano at an altitude of ~13.8 km on 15 June at the moment of the eruption, 17:30 UTC; and 2) not associated with the after-effect of the Redoubt eruption."
[**Page 13, lines 13–16, revised manuscript**]

**Comment:** Figure 10. These strong stratospheric layers at 15 km and their source are not subjected to a back trajectory test. Given that they are a much shorter time post eruption than the prior examples, it would critical to show if they connect to the volcano. If Eyja did not inject material this far above the tropopause, where did this material come from?

**Response:** Now we present in Fig. 11 an ensemble of the air-mass backward trajectories started from altitudes (~11.1–14.6 km a.s.l.) of strong stratospheric layers over Tomsk (Fig. 10) on 21 April 2010 at 00:00 LT (20 April, 17:00 UTC). Only one trajectory started from an altitude of ~11.6 km a.s.l. directly passed over Eyjafjallajökull volcano. The other trajectories passed south of the volcano.

**Instead of**
"Figure 11 shows the HYSPLIT air-mass backward trajectory started from an altitude of the detected aerosol layers (~11.6 km a.s.l.) over Tomsk on 21 April at 00:00 LT (20 April, 17:00 UTC). The trajectory passed over Eyjafjallajökull volcano on one of the eruption days, 16 April at 13:00 UTC, at the altitude $H_{\text{traj.}}^{\text{back.}} \approx 10.7$ km that is clearly higher than $H_{\text{MPA}} \leq 9$ km. This inconsistency between the altitudes $H_{\text{traj.}}^{\text{back.}}$ and $H_{\text{MPA}}$ ($H_{\text{traj.}}^{\text{back.}}$ should normally be equal to or lower than $H_{\text{MPA}}$) is discussed in Sect. 4."

**we wrote**
"Figure 11 shows the HYSPLIT air-mass backward ensemble trajectories started from altitudes of the detected aerosol layers (~11.1–14.6 km a.s.l.) over Tomsk on 21 April at 00:00 LT (20 April, 17:00 UTC). Only one trajectory (started from an altitude of ~11.6 km) directly passed over Eyjafjallajökull volcano on one of the eruption days, 16 April at 13:00 UTC, at the altitude $H_{\text{traj.}}^{\text{back.}} \approx 10.7$ km that is clearly higher than $H_{\text{MPA}} \leq 9$ km. The inconsistency between the HYSPLIT $H_{\text{traj.}}^{\text{back.}}$ and GVP $H_{\text{MPA}}$ altitudes is discussed in Sect. 4. The other trajectories passed south of the volcano."
[**Page 15, lines 9–13, revised manuscript**]

**Comment:** P15, L13. Regarding the connection between the weak layer in Figure 12, localized reduced total ozone, and Merapi, a stronger candidate would be Arctic O3 depletion. See Manney et al., (Nature, 2011) "Unprecedented Arctic Ozone Loss in 2011" doi:10.1038/nature10556.

**Response:** We agree with Dr. Fromm that the degree of ozone depletion in April 2011 was greater than it could be due to the influence on ozone of volcanic aerosol plumes. As the purpose of our research was to study aerosol perturbations of the stratosphere over Tomsk and was not to study the ozone depletion, here and elsewhere all information and discussions about ozone depletion and mini-holes were removed from the manuscript, without prejudice to the

generality of the foregoing. We substituted Fig. 12 by a new one with additional perturbed scattering ratio profiles to make it clearer and informative.

**Instead of**

"As an example, Fig. 12 presents an aerosol layer observed over Tomsk at an altitude of ~18 km on 18 April 2011."

**we wrote**

"Figure 12 presents the observed after-effect of the Merapi eruption, i.e. several perturbed scattering ratio profiles retrieved from the SLS aerosol lidar measurements between 28 February and 18 April 2011."
[**Page 16, lines 11 and 12, revised manuscript**]

**Instead of**

"**Figure 12.** Detection of the Merapi volcanic plume in the stratosphere over Tomsk. The volcano erupted in Indonesia from 4 to 5 November 2010."

**we wrote**

"**Figure 12.** Perturbed scattering ratio profiles retrieved from the SLS aerosol lidar measurements in the winter-spring period of 2011."
[**Page 17, line 2, revised manuscript**]

The sentence "Note also that a significant decrease in the total ozone content was observed over Tomsk at the same period of time (April 2011), which is evidence of stratospheric ozone depletion caused by the Merapi aerosol plume (Zuev et al., 2016)." was removed from our manuscript.
[**Page 16, line 15, revised manuscript**]

The reference Zuev et al., 2016 was removed from our manuscript too.

Zuev, V. V., Zueva, N. E., Savelieva, E. S., Bazhenov, O. E., and Nevzorov, A. V.: On the role of the eruption of the Merapi volcano in an anomalous total ozone decrease over Tomsk in April 2011, Atmos. Ocean. Opt., 29, 298–303, 2016.

**Comment:** P16, L4. This conclusion regarding Nabro's injection height has been exhaustively disputed. There are no citations here of the papers demonstrating the classic, direct injection of Nabro to the stratosphere. http://science.sciencemag.org/content/339/6120/647.4

http://science.sciencemag.org/content/339/6120/647.3

The Bourassa et al. scenario was shown to be improbable by Fairlie et al.(2014). Clarisse et al. (2014), Fairlie et al. (2014), and Penning de Vries et al. (2014) showed, using a variety of satellite data, that direct injection above the tropopause was more consistent with these data than the indirect path via the Monsoon. Fromm et al. (2014) established a root cause for the misattribution to the Asian Monsoon pathway and made connections with prior papers on other volcanic stratospheric aerosol discrepancies. Hence the full weight of Nabro-related papers gives a very different perspective than what is documented here. I would ask the authors to more fully capture these various works in their presentation on Nabro.

http://www.atmos-chem-phys.net/14/7045/2014/

http://www.atmos-chem-phys.net/14/3095/2014/

http://www.atmos-chem-phys.net/14/8149/2014/

http://onlinelibrary.wiley.com/doi/10.1002/2014JD021507/full.

**Response:** We have to agree with you. Thank you for these references. We used them all.

We revised and partially corrected Sect. 3.3.2.

**Instead of**

"Grimsvötn volcano erupted ash clouds and gases directly into the stratosphere at an altitude of 20 km, whereas the Nabro volcanic plume did not exceed the local tropopause altitude. Nevertheless, Bourassa et al. (2012) and Robock (2015) showed that a considerable part of the Nabro volcanic aerosol and gases, erupted into the upper troposphere, was able to enter the mid-latitude stratosphere due to deep convection and vertical air transport associated with the strong Asian summer monsoon anticyclone."

**we wrote**

"According to the GVP data, Grimsvötn volcano erupted ash clouds and gases directly into the stratosphere at an altitude of 20 km, whereas the Nabro volcanic plume did not exceed the local tropopause altitude. Bourassa et al. (2012) showed that a considerable part of the Nabro volcanic aerosol and gases, erupted into the upper troposphere, was able to

enter the mid-latitude stratosphere due to deep convection and vertical air transport associated with the strong Asian summer monsoon anticyclone. On the other hand, Vernier et al. (2013), Fromm et al. (2013), Fairlie et al. (2014), Clarisse et al. (2014), and Penning de Vries et al. (2014) showed that the initial Nabro plume was directly injected into the lower stratosphere at altitudes up to 18 km (Fromm eat al., 2014)."
[**Page 17, lines 5–11, revised manuscript**]

Vernier, J.-P., Thomason, L. W., Fairlie, T. D., Minnis, P., Palikonda, R., and Bedka, K. M.: Comment on "Large volcanic aerosol load in the stratosphere linked to Asian monsoon transport", Science, 339, 647-d, doi:10.1126/science.1227817, 2013.

Fromm, M., Nedoluha, G., and Charvát, Z.: Comment on "Large volcanic aerosol load in the stratosphere linked to Asian monsoon transport", Science, 339, 647-c, doi:10.1126/science.1228605, 2013.

Fairlie, T. D., Vernier, J.-P., Natarajan, M., and Bedka, K. M.: Dispersion of the Nabro volcanic plume and its relation to the Asian summer monsoon, Atmos. Chem. Phys., 14, 7045–7057, doi:10.5194/acp-14-7045-2014, 2014.

Clarisse, L., Coheur, P.-F., Theys, N., Hurtmans, D., and Clerbaux, C.: The 2011 Nabro eruption, a SO2 plume height analysis using IASI measurements, Atmos. Chem. Phys., 14, 3095-3111, doi:10.5194/acp-14-3095-2014, 2014.

Penning de Vries, M. J. M., Dörner, S., Puķīte, J., Hörmann, C., Fromm, M. D., and Wagner, T.: Characterisation of a stratospheric sulfate plume from the Nabro volcano using a combination of passive satellite measurements in nadir and limb geometry, Atmos. Chem. Phys., 14, 8149-8163, doi:10.5194/acp-14-8149-2014, 2014.

Fromm, M., Kablick III, G., Nedoluha, G., Carboni, E., Grainger, R., Campbell, J., and Lewis, J.: Correcting the record of volcanic stratospheric aerosol impact: Nabro and Sarychev Peak, J. Geophys. Res., 119, 10343–10364, doi:10.1002/2014JD021507, 2014.

**Comment:** P16, L18. Often? I'm not aware of any literature showing any direct positive correlation between PSC formation and volcanic sulfur loading. Hence a citation is needed. Fromm et al. (2003) actually showed little (or even negative) correlation between PSC frequency and ambient aerosol loading. Hence it would be important for the authors to substantiate the claim they make here.

Fromm, M., J. Alfred, and M. Pitts, A unified, long-term, high-latitude stratospheric aerosol and cloud database using SAM II, SAGE II, and POAM II/III data: Algorithm description, database definition, and climatology, J. Geophys. Res., 108(D12), 4366, doi:10.1029/2002JD002772, 2003.

**Response:** The PSC formation depends on both low temperature (lower than –78 °C) and the presence of condensation nuclei. The volcanogenic aerosol also plays the role of condensation nuclei. The PSC formation is a local process and could not work on the global scale. Concerning the direct positive correlation between PSC formation and volcanic sulfur loading, see, e.g., the following reference:

Rose, W. I., Millard, G. A., Mather, T. A., Hunton, D. E., Anderson, B., Oppenheimer, C., Thornton, B. F., Gerlach, T. M., Viggiano, A. A., Kondo, Y., Miller, T. M., and Ballenthin, J. O.: Atmospheric chemistry of a 33–34 hour old volcanic cloud from Hekla Volcano (Iceland): Insights from direct sampling and the application of chemical box modeling, J. Geophys. Res., 111, D20206, doi:10.1029/2005JD006872, 2006.

We rewrote Sect. 3.4 and included Fromm et al., (2003) with appropriate comments to our manuscript.

**Instead of**
"Therefore, the injections of volcanogenic $H_2SO_4$ aerosols or/and $SO_2$ into the stratosphere often lead to PSC formation, if the air temperature $< -78$ °C."

**we wrote**
"Therefore, injections of volcanogenic $H_2SO_4$ aerosols or/and $SO_2$ into the stratosphere can lead to PSC formation, if the air temperature $< -78$ °C. The direct positive correlation between PSC formation and volcanogenic nitric and sulfur acid aerosols loading was shown, e.g., by Rose et al. (2006). However, it should be noted that, in contrast to Rose et al. (2006), Fromm et al. (2003) showed little (or even negative) correlation between PSC events and ambient aerosol loading."
[**Page 18, lines 8–11, revised manuscript**]

**Comment:** P17, L6. Not necessarily. In fact, spatial correlations between total-O3 minima (socalled ozone mini-holes) and PSCs are routinely attributable to synoptic-scale dynamics. See

Hood et al. and Teitelbaum et al. Exploring polar stratospheric cloud and ozone minihole formation: The primary importance of synoptic scale flow perturbations, Teitelbaum, H., M. Fromm, & M. Moustaoui, J. Geophys. Res., 106, 28173–28188, 2001.

Hood, L. L., B.E. Soukharev, M. Fromm, and J.P. McCormack, Origin of extreme ozone minima at middle to high northern latitudes, J. Geophys. Res., 106, 20925–20940, 2001.

**Response:** All information and discussions about ozone depletion were removed from Sect. 3.4. See, please, the colored version of our revised manuscript for details.

**Instead of**

"Hence, the stratospheric temperature over Tomsk can occasionally be cooled lower than –78 °C, when Tomsk is inside the polar vortex.  Thus, the detection of aerosol layers in the stratosphere at extremely low temperatures  can be indicative of the presence of PSCs."

**we wrote**

"Hence, the stratospheric temperature over Tomsk can occasionally be cooled lower than –78 °C, when Tomsk is inside the polar vortex. Thus, the detection of aerosol layers in the stratosphere at extremely low temperatures can be indicative of the presence of PSCs."

**[Page 18, lines 14–16, revised manuscript]**

The reference Solomon (1999) was removed from our manuscript.

Solomon, S.: Stratospheric ozone depletion: A review of concepts and history, Rev. Geophys., 37, 275–316, doi:10.1029/1999RG900008, 1999.

**Instead of**

"The first lidar PSC observations (over Tomsk) , were made at λ = 1064 nm on January 1995 (Zuev and Smirnov, 1997)."

**we wrote**

"The first lidar PSC observations over Tomsk were made at λ = 1064 nm in January 1995 (Zuev and Smirnov, 1997)."

**[Page 19, line 7, revised manuscript]**

**Comment:** P17, L17. This PSC is consistent with the findings of Fromm et al. (1999) who showed that the cold pool and PSC frequency in the northern winter of 1994/95 was located near the Tomsk longitude.

http://onlinelibrary.wiley.com/doi/10.1029/1999JD900273/epdf.

**Response:** Thank you for this reference. We added Fromm et al. (1999) with appropriate comments to our manuscript.

**Instead of**

"The stratospheric temperature was lower than –80 °C ."

**we wrote**

"The stratospheric temperature was lower than –80 °C. The cold pool presence and PSC events near the Tomsk longitude during the northern winter of 1994/95 were also reported by Fromm et al. (1999). The formation of these dense PSCs was caused by high concentrations of residual post-Pinatubo aerosols."

**[Page 19, lines 9–12, revised manuscript]**

Fromm, M. D., Bevilacqua, R. M., Hornstein, J., Shettle, E., Hoppel, K., and Lumpe, J. D.: An analysis of Polar Ozone and Aerosol Measurement (POAM) II Arctic polar stratospheric cloud observations, 1993–1996, J. Geophys. Res., 104, 24341–24357, doi:10.1029/1999JD900273, 1999.

**Comment:** P17, L19. Please see the prior comment regarding mini-holes.

**Response:** See, please, our **Response** to **Comment:** P17, L6.

**Comment:** P17, L20. This seems to be quite unlikely considering that PSCs are formed inside the vortex, which represents air isolated from the extratropics. Rabaul aerosols, introduced just a few months before the northern 1994/95 vortex season, would not likely have been meridionally transported that far north in time to be in place before the vortex formed and isolation was in place.

**Response:** Well, aerosol resulted from the 1994 Rabaul eruption could enter inside the polar vortex weakened and deformed due to a minor sudden stratospheric warming in January 1995. This is a subject for discussion and additional research. Anyway, as the main cause of PSC formation is the extremely low temperatures, we excluded the Rabaul volcanic aerosol discussion from this part of the text.

**Comment:** P18, L2. It seems highly unlikely (or at least hard to prove) to argue for a PSC in the northern polar vortex one season after a southern hemisphere volcanic eruption. It would be best to state this as speculative, if the statement is to remain.

**Response:** We agree with this comment.

**Instead of**
"Another event of PSCs over Tomsk was observed at λ = 532 nm on 27 January 2007 (Fig. 14) as an after-effect of the Rabaul volcano eruption occurred on 7 October 2006 (Table 1)."

**we wrote**
"Another event of PSCs over Tomsk was observed at λ = 532 nm on 27 January 2007 (Fig. 14)."
[**Page 19, line 13, revised manuscript**]

**Comment:** P18, L6. See the prior comment about mini-holes. At this time in Jan 2007, a localized o3 minimum was at Tomsk's longitude. The ozone signature was likely an artifact of local dynamics.

http://exp-studies.tor.ec.gc.ca/cgibin/selectMap?lang=e&type1=du&day1=27&month1=01&year1=2007&howmany1=1&interval1=1&intervalunit1=d&hem1=P18.

**Response:** We agree with the comment. The discussion about ozone depletion was removed from the text.

**Instead of**
"Moreover, the stratospheric ozone was considerably depleted at that time and the total ozone content was 30 percent of the norm (Zuev et al., 2008). Thus, PSCs were detected at least twice (in 1995 and 2007) during 30 years of stratospheric aerosol lidar measurements in Tomsk."

**we wrote**
"High $R(H)$ values at altitudes in the range of 13 to 17 km were probably due to the winter aerosol supplying of the SAL from the stratospheric tropical aerosol reservoir enriched by the 2006 Rabaul eruption plume (Table 1, Fig. 14). Thus, PSCs were detected at least twice (in 1995 and 2007) during 30 years of stratospheric aerosol lidar measurements in Tomsk."
[**Page 19, lines 17–20, revised manuscript**]

The reference Zuev et al. (2008) was removed from our manuscript.

Zuev, V. V., Bazhenov, O. E., Burlakov, V. D., Grishaev, M. V., Dolgii, S. I., and Nevzorov, A. V.: On the effect of volcanic aerosol on variations of stratospheric ozone and NO2 according to measurements at the Siberian Lidar Station, Atmos. Ocean. Opt., 21, 825–831, 2008.

We also substituted Fig. 14 by a new one and corrected its caption.

**Instead of**
"**Figure 14.** Detection of the Rabaul volcanic plume together with PSCs, formed at extremely low temperatures (< –78 °C), in the stratosphere over Tomsk. Rabaul volcano erupted in Papua New Guinea on 7 October 2006. Temperature profiles were retrieved from radiosondes launched on 27 January 2007 in Kolpashevo (station 29231) at 00:00 UTC and in Novosibirsk (station 29634) at 12:00 UTC (WWW, 2007)."

**we wrote**
"**Figure 14.** Detection of PSCs formed at extremely low temperatures (< –78 °C) in the stratosphere over Tomsk. Temperature profiles were obtained from radiosondes launched on 27 January 2007 in Kolpashevo (station 29231) at 00:00 UTC and in Novosibirsk (station 29634) at 12:00 UTC (WWW, 2007). The dashed ellipse denotes the after-effect of the Rabaul volcanic eruption occurred in Papua New Guinea on 7 October 2006."
[**Page 19, lines 2–5, revised manuscript**]

**Comment:** P18, L26. By postulating that pyroCb smoke could increase "annual average" stratospheric aerosol, it would imply that prior research has come to similar conclusions. I'm not aware of any such finding. Please modify this statement appropriately.

**Response:** You are probably right. We agree that extensive forest (bush) fires (and pyroCb smoke) could hardly be expected to increase the **annual average** $B_\pi^a$ value. We removed discussion about forest extensive fires and pyro-cumulonimbus clouds from Sect. 3.5 (without prejudice to the foregoing) and retained it only in Sect. 4.

The following part of the Sect. 3.5 was removed:

"Extensive forest (bush) fires could be another cause of occasional increases of the annual average $B_\pi^a$ value. Combustion products (gases and aerosol particles) can reach the stratospheric altitudes via convective ascent within pyro cumulonimbus (pyroCb) clouds (see, e.g., Fromm et al., 2006). For example, the smoke plumes from the strong bush fire, occurred near the Australian city of Melbourne on 7 February 2009, were observed in the local stratosphere at an altitude of ~18 km (Siddaway and Petelina, 2011). In recent years, the number and intensity of massive forest fires have significantly increased (e.g., in the USA; Trickl et al., 2013) due to the climate change. The smoke-filled air masses frequently enter the stratosphere over the South of Western Siberia from North America, where extensive forest fires occur. Their smoke plumes are most likely to be detected as the SAL perturbations over Tomsk. However, more detailed information about the pyroCb events is required for their correct identification. Thus, both the Kelut plume and smoke plumes from massive forest fires in the USA and Canada, with equal probability, could be the cause of the increase of $B_\pi^a$ value in the stratosphere over Tomsk during the first quarter of 2015."

**Comment:** P19, L13. "always" was not proved or demonstrated herein. Hence a citation is needed, or this point should be restated or removed.

**Response:** We agree. This "always" was removed from the text. Some of plumes from the tropical volcanoes eruptions can completely be within the Southern Hemisphere stratosphere. The tropical eruptions which plumes were within the Northern Hemisphere stratosphere and passed over Tomsk are presented in our manuscript.

**Instead of**

"Additional aerosol loading of the tropical reservoir always leads to an increase in the annual average $B_\pi^a$ value in the Northern Hemisphere mid-latitude stratosphere via the meridional transport in the cold seasons (October to March)."

**we wrote**

"Additional aerosol loading of the tropical reservoir can usually lead to an increase in the annual average $B_\pi^a$ value in the Northern Hemisphere mid-latitude stratosphere via the meridional transport in the cold seasons (October to March; Hitchman et al., 1994)."
**[Page 21, lines 6–8, revised manuscript]**

Hitchman, M.H., McKay, M., Trepte, C.R.: A climatology of stratospheric aerosol, J. Geophys. Res., 99, 20689-20700, doi:10.1029/94JD01525,1994.

**Comment:** P19, L18. It's not clear what is meant here. All volcanic plumes are represented as an initial point source. Please clarify.

**Response:** OK. We rewrote and clarified the sentence.

**Instead of**

"On the other hand, by contrast to tropical volcanoes, the northern ones represent point sources of volcanic gas, aerosol, and ash plumes. Their corresponding air-mass trajectories can either pass over a lidar station or pass it by."

**we wrote**

"On the other hand, by contrast to tropical volcanoes, the narrow volcanic gas, aerosol, and ash plumes from northern volcanoes can either pass over a lidar station or pass it by."
[**Page 21, lines 11 and 12, revised manuscript**]

**Comment:** P20, L15. What is meant by "thermal speed?" Please clarify.

**Response:** This is the speed achieved due to heating of erupted mass in the "gas thrust" and "convective thrust" regions of an eruption column, not due to any potential (conservative) fields like magnetic, Coulomb or gravitational fields. The word "thermal" was removed from the text.

**Comment:** P20, L15. This is confusing. A plume is buoyant because it is less dense than the surrounding air. This makes it sound as if the surrounding umbrella region is less dense.

**Response:** We agree with this comment. The sentence was rewritten accordingly.

**Instead of**

"The most heated fraction of gas-vapor emissions from the "convective thrust region" has the high  speed and, therefore, can penetrate through the -density "umbrella region" of the eruption column and reach altitudes higher than $H_{\text{MPA}}$ (Raible et al., 2016)."

**we wrote**

"The most heated fraction of gas-vapor emissions from the "convective thrust region" has the highest speed and, therefore, can penetrate through the higher-density "umbrella region" of the eruption column and reach altitudes higher than $H_{\text{MPA}}$ due to the cumulative (jet) effect (Raible et al., 2016)."
**[Page 21, lines 25–27, revised manuscript]**

**And**

**Instead of**

"In addition to volcanoes, PSCs also represent a cause of significant SAL perturbations ."

**we wrote**

"In addition to volcanoes, PSCs also represent a cause of significant SAL perturbations."
[**Page 22, line 1, revised manuscript**]

**Comment:** P20, L24. "drifting" needs to be clarified. It implies to me that the ozone feature is drifting with the wind, whereas the papers previously offered in these comments show that the minihole is tied to dynamics and thus move with the speed of the synoptic-scale wave, not the wind.

**Response:** The sentences "The possibility of PSCs to form and be detected in the mid-latitudes is usually related with the presence of "mini ozone holes" drifting over lidar measurement points. As the lifetime of these "holes" is sufficiently short in the mid-latitudes, the PSC observations can be only occasional." were removed from Sect. 4.

**Comment:** P20, L29. The impact of pyroCb smoke on annual average stratospheric aerosol has not been shown in the literature to my knowledge. Please cite a paper or modify this statement accordingly.

**Response:** Yes, we agree. See, please, our **Response** to **Comment:** P18, L26 concerning the **annual average** $B_\pi^a$ value.

**Comment:** P20, L32. The measures in acres and square km are not equivalent. Please correct this.

**Comment:** P21, L2. What is the reason to call out the Happy Camp fire?

**Response:** Thank you. We corrected the mistake: 134 acres $\approx$ 0.543 km$^2$. However, we removed the information about the Happy Camp fire from the text of our manuscript.

**We rewrote a substantial part of Sect. 4 to make it clearer.**

**Instead of**

"Smoke plumes from strong forest (bush) fires can reach the stratospheric altitudes (Fromm et al., 2006; Siddaway and Petelina, 2011), spread out to great distances, and perturb the SAL state over different regions, including Tomsk. This is known to result in a measurable increase in the annual average $B_\pi^a$ value. Due to the climate warming, the number and intensity of massive forest fires have significantly increased in the last few years (Wotton et al., 2010). For example, about 137 strong forest fires were registered in the Northwest Territories of Canada in July 2014 (CBC News, 2014), and the Happy Camp Complex fire (41.80° N, 123.37° E) eventually consumed more than 134 acres (~0.543 km$^2$) of forests in California in August–October 2014. According to the California Department of Forestry and Fire Protection (CAL FIRE, http://www.ca.gov/) data, the Happy Camp Complex fire is in the list of the "Top 20 Largest California Wildfires". The smoke plumes from these mentioned massive forest fires could probably cause the increase in $B_\pi^a$ value in the stratosphere over Tomsk in January–March 2015. More detailed information about the pyroCb events such as precise time, place, and smoke plume altitude is required to correctly assign the pyroCb plumes to the corresponding aerosol layers over an observation point via the HYSPLIT model trajectory analysis. It is quite possible that some after-effects of strong forest fires occurred, e.g., in North America could be detected over Tomsk, but not identified during lidar observations in Tomsk (1986–2015)."

**we wrote**

"Extensive forest (bush) fires could be another cause of occasional increases of the $B_\pi^a$ value. Combustion products (gases and aerosol particles) can reach the stratospheric altitudes via convective ascent within pyro-cumulonimbus (pyroCb) clouds (see, e.g., Fromm et al., 2006). For example, the smoke plumes from the strong bush fire, occurred near the Australian city of Melbourne on 7 February 2009, were observed in the local stratosphere at an altitude of ~18 km (Siddaway and Petelina, 2011). Due to the climate warming, the number and intensity of massive forest fires have considerably increased in the last few years (Wotton et al., 2010). For example, about 137 strong forest fires were registered in the Northwest Territories of Canada in July 2014 (CBC News, 2014). The smoke-filled air masses frequently enter the stratosphere over the South of Western Siberia from North America, where extensive forest fires occur. Their smoke plumes are most likely to be detected as the SAL perturbations over Tomsk. However, more detailed information about the pyroCb events is required for their correct identification. It is quite possible that some after-effects of strong forest fires occurred, e.g., in North America could be detected over Tomsk, but not identified during lidar observations in Tomsk (1986–2015)."
[**Page 22, lines 5–14, revised manuscript**]

**Comment:** P21, L4. Regarding pyroCb smoke, I believe that there would be many occasions through the years for stratospheric observations at Tomsk. But of course the frequency of such observations would decrease rapidly with altitude above the tropopause. The highest smoke layers in the northern hemisphere to my knowledge are 19 km. They are much more likely to be at 12-15 km. Perhaps it would be to your advantage to directly employ the local tropopause height in the search for pyroCb smoke instead of a fixed altitude (e.g. 15 km) that is generally 4 km above the tropopause.

**Response:** We agree that the lower limit of 12 km for aerosol observations is better than of 15 km. Anyway, the perturbed scattering ratio profiles were also detected at altitudes higher than 15 km.

**Comment:** Technical Comments See the accompanying pdf of the manuscript. Please also note the supplement to this comment:

http://www.atmos-chem-phys-discuss.net/acp-2016-792/acp-2016-792-RC2-supplement.pdf.

**Response:** We corrected the text in accordance with Dr. Fromm's technical comments.

**Comment: in the Technical Comments**: Trickl did not conclude that the 14.3 km aerosol was from Eyja. They were more circumspect about all the aerosols above the tropopause because back trajectories did not come close to Iceland or pass near Iceland during a big eruption day.

**Response:** At least, Trickl noted in his article the following:

(Abstract) A key observation for judging the role of eruptions just reaching the tropopause region was that of the plume from the Icelandic volcano Eyjafjallajökull above Garmisch-Partenkirchen (April 2010) due to the proximity of that source. The top altitude of the ash above the volcano was reported just as 9.3 km, but the lidar measurements revealed enhanced stratospheric aerosol up to 14.3 km.)

(Sect. 4.2, page 5215) The upper boundary of these aerosol layers was roughly 12 km on 17 and 19 April, and 14.3 km on 20 April (Fig. 9), i.e., above the thermal tropopause that was 10.2 km on average on these days.

Sincerely,

Authors

**Manuscript Number:** acp-2016-792
**Manuscript Type:** Research article
**Title:** 30-year lidar observations of the stratospheric aerosol layer state over Tomsk (Western Siberia, Russia)

**List of corrections made according to Referee 1 and Dr. Fromm comments**

**We substituted Figures 1, 2, 3, 5a, 11, 12, 13, and 14 by new ones and corrected their captions.**

**New Figures 5b, 6c, were added to the manuscript.**

**Page 1**

**Instead of**
"We also make an assumption that both the Kelut volcano plume (Indonesia, February 2014) and smoke plumes from massive forest fires occurred in Canada (137 fires in the Northwest Territories, July 2014) and the USA (the Happy Camp Complex fire in California, August–October 2014), with equal probability, could be the cause of the SAL perturbations over Tomsk during the first quarter of 2015."

**we wrote**
"We also make an assumption that the Kelut volcano eruption (Indonesia, February 2014) could be the cause of the SAL perturbations over Tomsk during the first quarter of 2015."
[**Page 1, lines 27 and 28, revised manuscript**]

**Page 2**

**Instead of**
"The volcanogenic aerosol perturbs the radiation-heat balance of the atmosphere, and thus, significantly affects the atmospheric dynamics (Timmreck, 2012; Driscoll et al., 2012)."

**we wrote**
"The volcanogenic aerosol perturbs the radiation-heat balance of the atmosphere, and thus, significantly affects the atmospheric dynamics and climate (Timmreck, 2012; Driscoll et al., 2012; Kremser et al., 2016)."
[**Page 2, lines 7 and 8, revised manuscript**]

**Page 3**

**Instead of**
"…can be useful, e.g., in studying climate change."

**we wrote**
"…can be useful, e.g., in studying climate change (Mills et al., 2016)."
[**Page 3, lines 3 and 4, revised manuscript**]

**The following sentence was added to the manuscript:**
"Note that the CIS-LiNet station located in Minsk, Belarus, is also integrated into the European Aerosol Research Lidar Network (EARLINET; Wandinger et al., 2016)."
[**Page 3, lines 9 and 10, revised manuscript**]

**Instead of**
"…FEU-130 operating in the photon counting mode."

**we wrote**
"…FEU-130 (USSR, Moscow Elecro-Lamp Plant) operating in the photon counting mode."
[**Page 3, lines 25 and 26, revised manuscript**]

**Page 4**

**The following sentence was added to the manuscript:**
"A more detailed technical description of the SLS aerosol channel and its data acquisition electronics can be found, e.g., in (Burlakov et al., 2010)."
[**Page 4, lines 10 and 11, revised manuscript**]

**The following sentence was added to the manuscript:**
"π denotes an angle of π radian, i.e. the angle of the backscatter lidar signal propagation."
[**Page 4, lines 14 and 15, revised manuscript**]

**Instead of**
"In our case, the tropopause altitude over Tomsk varies from ~11 to 13 km, depending on season, and therefore, we set $H_1 = 15$ km."

**we wrote**
"Tomsk is located near the southern boundary of subarctic latitudes, where the tropopause altitude can significantly vary, e.g., due to migration of the Arctic stratospheric jet stream within the Tomsk region. Sometimes one can observe a double (or even multiple) tropopause. For this reason, we consciously removed the interval of the tropopause altitude variations to observe the stratospheric perturbations only. As the tropopause altitude over Tomsk varies from ~11 to 13 km, depending on season, we set $H_1 = 15$ km."
[**Page 4, lines 23–26, revised manuscript**]

**Page 5**

**Instead of**
"**Figure 1.** … The red bars correspond to tropical volcanic eruptions, whereas the black ones correspond to eruptions of extratropical volcanoes located in the Northern Hemisphere. PSC: polar stratospheric clouds."

**we wrote**
"**Figure 1.** … The red bars correspond to tropical volcanic eruptions, whereas the black ones correspond to eruptions of extratropical volcanoes located in the Northern Hemisphere. The thin horizontal line in Fig. 1 indicates the minimum value of the annual average $B_\pi^a$ reached in 2004. PSC: polar stratospheric clouds."
[**Page 5, lines 16 and 17, revised manuscript**]

**Page 7**

**Figure 2 is now Figure 3 and, conversely, Figure 3 is now Figure 2 in the revised manuscript.**

**Instead of**
"**Figure 1.** … The red bars correspond to tropical volcanic eruptions, whereas the black ones correspond to eruptions of extratropical volcanoes located in the Northern Hemisphere. PSC: polar stratospheric clouds."

**we wrote**
"**Figure 2.** Inter-annual variations of $B_\pi^a$ values (in the stratosphere over Tomsk) separately averaged over the warm and cold half-years. The "warm" and "cold" average points are assigned to 1 June of the current year and 1 January of the next year, respectively. Black and red vertical bars at the bottom of the figure indicate volcanic eruptions as in Fig. 1 (see also Table 1). All error bars represent the standard deviation."
[**Page 7, lines 10–13, revised manuscript**]

**Instead of**
"…background level of $B_\pi^a$ reached after 1999. Note that only…"
**we wrote**
"…background level of $B_\pi^a$ reached after 1998. Only…."
[**Page 7, line 16, revised manuscript**]

**Instead of**
"The minimum annual average $B_\pi^a$ values were reached in 2003–2004."
**we wrote**
"The minimum annual average $B_\pi^a = 1.29 \times 10^{-4}$ sr$^{-1}$ was reached in 2004."
[**Page 7, line 18, revised manuscript**]

**Instead of**
"The thin horizontal line in Fig. 1 indicates the minimum value of the annual average $B_\pi^a$ reached in 2004. This value equals to that determined in 1979 and considered as the background one (Trickl et al., 2013)."

"Note that taking into account the spectral dependence of $B_\pi^a$, its minimum annual average value observed in Tomsk at $\lambda$ = 532 nm in 2004 was close to that determined for Garmisch-Partenkirchen at $\lambda$ = 694 nm in 1979 and considered as the background by Trickl et al. (2013)."
[**Page 7, line 20 and Page 8, lines 1 and 2, revised manuscript**]

**Pages 8 and 9**

We considerably rewrote Sect. 3.1. See, please, the colored version of our revised manuscript for details.

**Pages 10 and 11**

Section 3.2.1 was deeply revised and rewritten. Figures 5b and 6c were added.

**Instead of**

[revised manuscript text omitted]

---

## Author Response (AR2)

**Manuscript Number:** acp-2016-792
**Manuscript Type:** Research article
**Title:** 30-year lidar observations of the stratospheric aerosol layer state over Tomsk (Western Siberia, Russia)

**Author's response to the technical corrections suggested by Dr. Fromm**

**\*\*\*A comment from Referee 1:** I would like to see more about the normalization. Is only a single altitude used? Is it 30 km? Many of the Scattering Ratio profiles shown in the paper haven't decreased to 1.0 at 30 km, so higher altitude data must have been used. Is there an objective way to do this?

**Our response:** The SLS aerosol channel makes it possible to receive almost undisturbed backscattered signals from altitudes of ~40–45 km. At higher altitudes, the signal-to-noise ratio is too low. Therefore, altitudes of ~30–35 km, where the stratosphere is considered to be aerosol-free, were used as the calibration altitudes.\*\*\*

**The comment from Dr. Fromm (concerning our response to the Referee 1 comment):** It appears the reviewer was asking for a change to the paper. The authors should indicate if they made any change to the paper.

**Our response:** Answering **the Referee 1 comment** given above, we did not make any changes in our manuscript. However, we have decided to add some sentences to clarify this point in the text.

**Instead of**
"The detected lidar signals were calibrated by normalizing them to the molecular backscatter signal from aerosol-free altitudes above the SAL, i.e. $H_0 \geq 30$ km ($H_0$ is called the calibration altitude)."

**We wrote**
"The SLS aerosol channel makes it possible to receive almost undisturbed backscattered signals from altitudes of ~40–45 km. At higher altitudes, the signal-to-noise ratio is too low. Therefore, altitudes of ~30–35 km, where the stratosphere is considered to be aerosol-free, were used as the calibration altitudes $H_0$. Thus, the detected lidar signals were calibrated by normalizing them to the molecular backscatter signal from altitudes $H_0 \geq 30$ km."
[**Page 4, lines 15–19, revised manuscript**]

**Comment:** Perhaps "Contemporaneous" instead of "Simultaneous" would be more appropriate.

**Response:** We agree with this suggestion and made the corresponding correction in the text.

**Instead of**
"Simultaneous stratospheric aerosol observations at the Minsk CIS-LiNet station at $\lambda = 532$ nm revealed the similar SAL perturbations over Minsk from July to October (Fig. 5b; Zuev et al., 2009)."

**We wrote**
"Contemporaneous stratospheric aerosol observations at the Minsk CIS-LiNet station at $\lambda = 532$ nm revealed the similar SAL perturbations over Minsk from July to October (Fig. 5b; Zuev et al., 2009)."
[**Page 4, lines 15–19, revised manuscript**]

**Comment:** Suggestion: make abscissa range same for both, to make them easier to compare quickly.

**Response:** We agree. We substituted Fig. 5 by a new one with required correction. **See, please, Fig. 5 on Page 11, lines 18 and 19, revised manuscript.**

**Comment:** Perhaps replace "moment" with "time".

**Response:** OK. We made the required correction.

**Instead of**

[revised manuscript text omitted]